# Drivers of nitrogen and phosphorus dynamics in a groundwater-fed urban catchment revealed by high frequency monitoring

Liang Yu[1, 2], Joachim C. Rozemeijer[3], Hans Peter Broers[4], Boris M. van Breukelen[5], Jack J. Middelburg[6], Maarten Ouboter[2], and Ype van der Velde[1]

[1]Faculty of Science, Vrije University Amsterdam, 1181HV, Amsterdam, the Netherlands
[2]Waternet Water Authority, 1096 AC, Amsterdam, the Netherlands
[3]Deltares, 3508 TC, Utrecht, the Netherlands
[4]TNO Geological Survey of the Netherlands, 3584 CB, Utrecht, the Netherlands
[5]Department of Water Management, Faculty of Civil Engineering and Geosciences, Delft University of Technology, Stevinweg 1, 2628 CN, Delft, the Netherlands
[6]Department of Earth Sciences, Faculty of Geosciences, Utrecht University, P.O. Box 80 021, 3508 TA, Utrecht, the Netherlands

*Correspondence to*: Liang Yu (xiaobaidrawing@gmail.com)

**Abstract.** Eutrophication of water bodies has been a problem causing severe degradation of water quality in cities. To gain mechanistic understanding of the temporal dynamics of nitrogen (N) and phosphorus (P) in a groundwater fed low-lying urban polder, we applied high frequency monitoring in Geuzenveld, a polder in the city of Amsterdam. The high frequency monitoring equipment was installed at the pumping station where water leaves the polder. From March 2016 to June 2017, total phosphorus (TP), ammonium ($NH_4$), turbidity, electrical conductivity (EC), and water temperature were measured at intervals smaller than 20 minutes. This paper discussed the results at three time scales: annual scale, rain event scale, and single pumping event scale. Mixing of upwelling groundwater (main source of N and P) and runoff from precipitation on pavements and roofs was the dominant hydrological process governing the temporal pattern of the EC, while N and P fluxes from the polder were also regulated by primary production and iron transformations. In our groundwater-seepage controlled catchment, $NH_4$ appeared to be the dominant form of N with surface water concentrations in the range of 2-6 mg N/L, which stems from production in an organic-rich subsurface. The concentrations of $NH_4$ in the surface water were governed by the mixing process in autumn and winter and were reduced down to 0.1 mg N/L during the algae growing season in spring. The depletion of dissolved $NH_4$ in spring suggests uptake by primary producers, consistent with high concentrations of chlorophyll-a, $O_2$, and suspended solids during this period. Total P and turbidity were high during winter (range 0.5-2.5 mg P/L and 200-1800 FNU, respectively) due to the release of P and reduced iron from anoxic sediment to the water column, where $Fe^{2+}$ was rapidly oxidised and precipitated as iron oxides which contributed to turbidity. In the other seasons, P is retained in the sediment by sorption to precipitated iron oxides. Nitrogen is exported from the polder to the receiving waters throughout the whole year, mostly in the form of $NH_4$, but in the form of organic N in spring. P leaves the polder mainly during winter, primarily associated with $Fe(OH)_3$ colloids and as dissolved P. Based on this new understanding of the dynamics of N and P in this low lying urban catchment, we suggested management strategies that may effectively control and reduce eutrophication in urban polders and receiving downstream waters.

Keywords: Nitrogen and phosphorus dynamic, high frequency monitoring, benthic algae, iron chemistry, Amsterdam, groundwater seepage

## 1 Introduction

Eutrophication is one of the most notorious phenomena of water quality impairment in cities, caused by excess inputs of N and P. The identified sources of nutrients are from wastewater treatment plants, storm runoff, overflow of sewage systems, manure and fertilizer application in urban green areas and atmospheric deposition (Walsh et al., 2005; Kabenge et al., 2016; Toor et al., 2017; Yang & Toor, 2018; Putt et al., 2019). Recently, groundwater has been identified as another important source

of N and P in cities situated in low-lying deltas, where dissolved $NH_4$ and $PO_4$ in groundwater seep up into urban surface water (Yu et al, 2018 & 2019). The upwelling nutrients in groundwater, originating from the organic rich delta subsurface, reaching the surface water of cities and are transferred to downstream waters and eventually reach the coastal zones, where they may induce harmful algal blooms or cause hypoxia along coastlines (He and Xu, 2015; Beusen et al., 2016; Le Moal et al., 2019). Hence, it is of pivotal importance to understand N and P dynamics in the urban freshwater bodies in order to mitigate the input of nutrients into the oceans (e.g. Nyenje, et al., 2010; Toor et al., Paerl et al., 2016; 2017; Le Moal et al., 2019).

Nutrients dynamics are governed by biological, chemical and physical processes and their interactions. Assimilation by primary producers is a major biological factor regulating N and P concentrations in the aquatic environment. Aquatic micro- and macro-organisms assimilate P as $PO_4$ and N mainly in fixed forms such as nitrate ($NO_3$) and ammonium ($NH_4$), but for some specific organisms also in the form of $N_2$. In estuaries, $NH_4$ is the preferred N-form for~~by~~ microbes ~~in some cases like in estuaries~~ (Middelburg and Nieuwenhuize, 2000), but the uptake rate for both $NH_4$ and $NO_3$ can achieve maximum rates under sustained exposure of $NH_4$ or $NO_3$ (Bunch and Bernot, 2012). Moreover, the nitrogen species are also involved in redox transformations (Soetaert and Herman, 1995). Under anaerobic conditions, $NO_3$ can be reduced to $NH_4$, in particular with high organic matter contents~~, or~~ It may also be denitrified to $N_2$ and $N_2O$ under such condition (Mulder et al., 1995), the latter is a climate-active gas. Under aerobic conditions, $NH_4$ can be oxidized to $NO_3$ through nitrification by nitrifying microbes, which is an $O_2$ consuming and acid generating process. Nitrification even occurs under cold conditions (below 10 °C) (Painter, 1970; Wilczak et al., 1996; Cavaliere and Baulch, 2019).

The mixing of water from different flow routes is an important hydrological process that controls nutrient dynamics (Rozemeijer and Broers, 2007; Rozemeijer et al., 2010a; Van der Grift et al., 2014; Yu et al., 2019). As nutrient concentrations and speciation differ among different flow routes (Wriedt et al., 2007; Rozemeijer et al., 2010a; Yu et al., 2019; Yang and Toor, 2019), the mixing process results in dilution or enrichment of nutrients in surface water bodies during precipitation events (Wang et al., 2016).

Retention is another factor that determines nutrient concentrations and transport (McGlathery et al., 2001; Zhu et al., 2004; Henry and Fisher, 2003), especially for phosphorus most of which is retained in inland water bodies sediment (Audet et al., 2019). The retained P are either being permanently buried in the sediment or temporarily stored and acting later on as internal nutrient source (Kleeberg et al., 2007; Filippelli, 2008; Zhang et al., 2018). Multiple researchers have highlighted the influence of iron chemistry on the dynamics of P in pH neutral environments (Chen et al., 2018; Van der Grift et al., 2018). This is especially relevant when iron-rich groundwater interacts with surface water (Griffioen, 2006; Rozemeijer et al., 2010a; Van der Grift, 2014; Yu et al., 2019), in which P is immobilized by the formation of iron(oxy)hydroxides during groundwater aeration. However, changes in chemistry or temperature may lead to the release of P and reduced iron. For instance, under anaerobic conditions, Fe and P can be mobilized by sulfate reduction, but this can be counteracted by the presence of $NO_3$ as electron acceptor (Smolders et al., 2006).

Most studies of eutrophication are based on discrete sampling events which can give a general pattern of nutrient dynamics, but can easily miss important nutrient transport and processing phenomena (Rozemeijer et al., 2010; Rode et al., 2016; Toor et al., 2017). The countermeasures to control eutrophication have been hampered because of limited knowledge of N and P dynamics, for instance their response to changing weather conditions and land use (van Geer et al., 2016). In the past few years, the development of new sensors and sampling technologies allow us to obtain data with substantially shorter intervals. In this paper, the high frequency monitoring technology is referred to as an automatic monitoring program with sampling and analyzing frequencies that are sufficient for obtaining detailed water quality variation information. High frequency technology has proved to be a way to understand nutrient dynamics (Rode et al., 2016; Van Geer et al., 2016; Bieroza et al., 2018). Due to the abundant information offered by this technology, combined methodologies have been developed to quantitatively understand the in stream hydrochemistry of nutrients (Miller et al., 2016, Van der Grift et al., 2016, Duncan et al., 2017).

In our previous study on the water quality of Amsterdam (Yu et al, 2019), the transport routes of N and P from groundwater to surface water through seepage and drains were identified. In addition, spatial and temporal concentration patterns from discrete sampling campaigns showed a clear dilution pattern of other water quality parameters such as EC. However, the temporal patterns of N and P were still poorly understood, probably due to their reactive nature and more complex biogeochemistry. In order to obtain insight into the controlling mechanisms of N and P transport and fate in urban delta catchments affected by groundwater, we performed a year-round high-resolution N and P concentration monitoring campaign. A deep understanding of the water quality dynamic drivers would be a great asset for controlling eutrophication and improving aquatic ecological status (Fletcher et al., 2015; Díaz et al., 2016; Eggimann et al., 2017; Nizzoli et al., 2020). We conducted a one-year high frequency monitoring campaign in 2016-2017. Measured parameters were EC, $NH_4$, TP, turbidity and water temperature. The temporal patterns of these parameters were studied at three time scales: the annual scale, rain event scale, and pumping event scale.

## 2 Methods

### 2.1 Study site

The Geuzenveld study site is part of an urban lowland polder catchment, which is characterized by groundwater seepage that constantly determines the surface water quality, being the main source of solutes in the water system. The groundwater seepage is a continuous source of slightly brackish, anoxic, and iron and nutrient rich slightly brackish waters. Yu et al. (2019) presented the results of a 10 year monitoring program describing the main processes determining the water quality in the catchments, which is dominated by mixing of runoff water and seepage water. A high-frequency monitoring campaign was set-up to further unravel the temporal patterns ofon the nutrient N and P, of which N is typically present in the form of $NH_4$ from groundwater. Geuzenveld is a newly built urban polder on the west of the city of Amsterdam (Fig.1). Since the 1990s, when it was converted from agricultural to urban land, it has developed into a highly paved area. Similar to other new neighborhoods, Geuzenveld is equipped with a separated drainage system. A rain harvesting system was installed on all the buildings and houses in the polder, leading rain water from the roof and the street directly to the ditches, which results in fast and large amounts of runoff during storm events. Geuzenveld is a groundwater fed catchment due to the constantly higher groundwater head (-2.5 ~ -3 m NAP, NAP: Normalized Amsterdam Peil) in the main aquifer relative to the surface water level in the polder ditches (~ -4.25 m NAP). (Fig.2). To keep the foundations of the building dry, there is a groundwater drainage system placed under an artificial sandy layer, right on top of a natural clay layer. The drain elevations range from -4.84 to -4.61 m NAP , which is below the phreatic groundwater level throughout the year, making sure that groundwater seepage either discharges through the drains or the ditches.

The water system of Geuzenveld is connected to the secondary water channel to its east, then connected to the adjacent primary channel, called boezem, the Boezem Haarlemmerweg. The boezem water level is -2.10 m NAP. It is much higher than the target surface water level of Geuzenveld, -4.25 m NAP. The surface water level in polder Geuzenveld is controlled by a pump station, which is the main outlet of this polder, situated in the northeastern corner.

There are two pumps (Pump 1 and Pump 2) in the pumping station, and they have different start and end pumping threshold points (Table 1).

**Table 1 Pumping scheme of polder Geuzenveld**

| Time | Settings | Pump 1 | Pump 2 |
|------|----------|--------|--------|
| 05:00:00-19:00:00 | Start point (m NAP) | -4.20 | -4.16 |
| | End point (m NAP) | -4.26 | -4.24 |
| 19:00:00-05:00:00 | Start point (m NAP) | -4.23 | -4.18 |

| | End point (m NAP) | -4.31 | -4.29 |
|---|---|---|---|

The two pumps are activated when the surface water level exceeds the triggering level which are furthermore separated as day and night triggering levels (Table 1). The capacity of each pump is 3.6 m$^3$ per minute. Most of the time, only one of the two pumps works and the surface water level is maintained between -4.31 m NAP and -4.23 m NAP, which are the night inactive and active pumping levels respectively. Normally, the surface water level drops immediately when the pump(s) start(s) working. Once the pump(s) stop(s), the surface water level will steadily rise due to the continuous inflow of groundwater seepage. During rainfall events, the surface water level rises faster (Fig.2A).

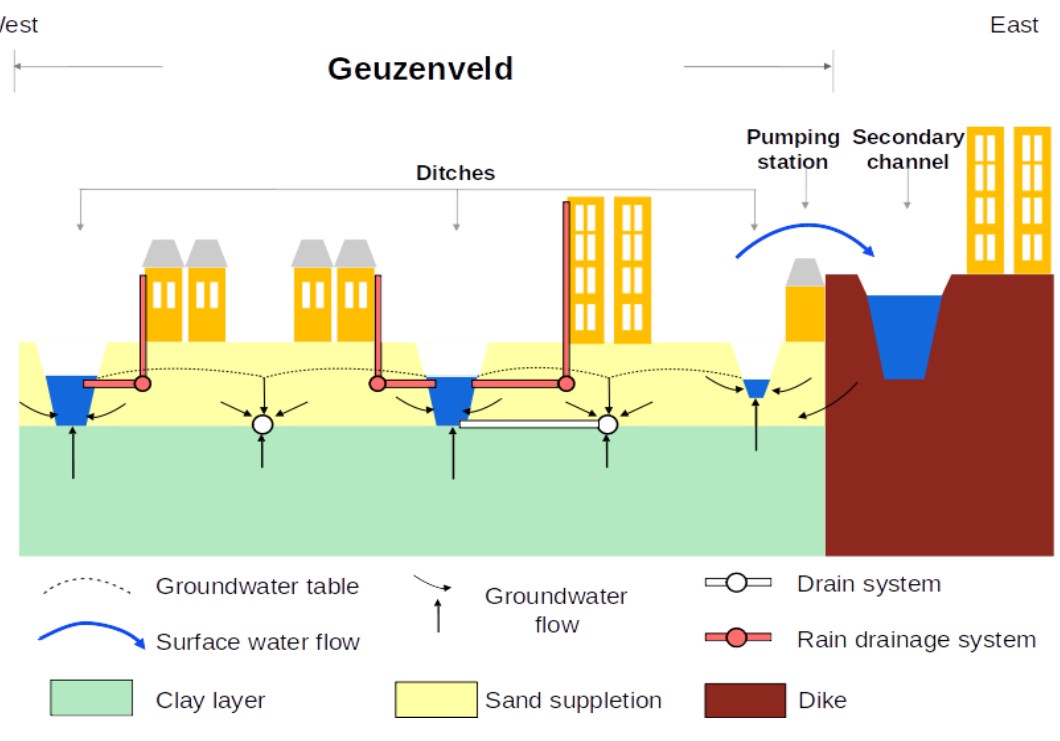

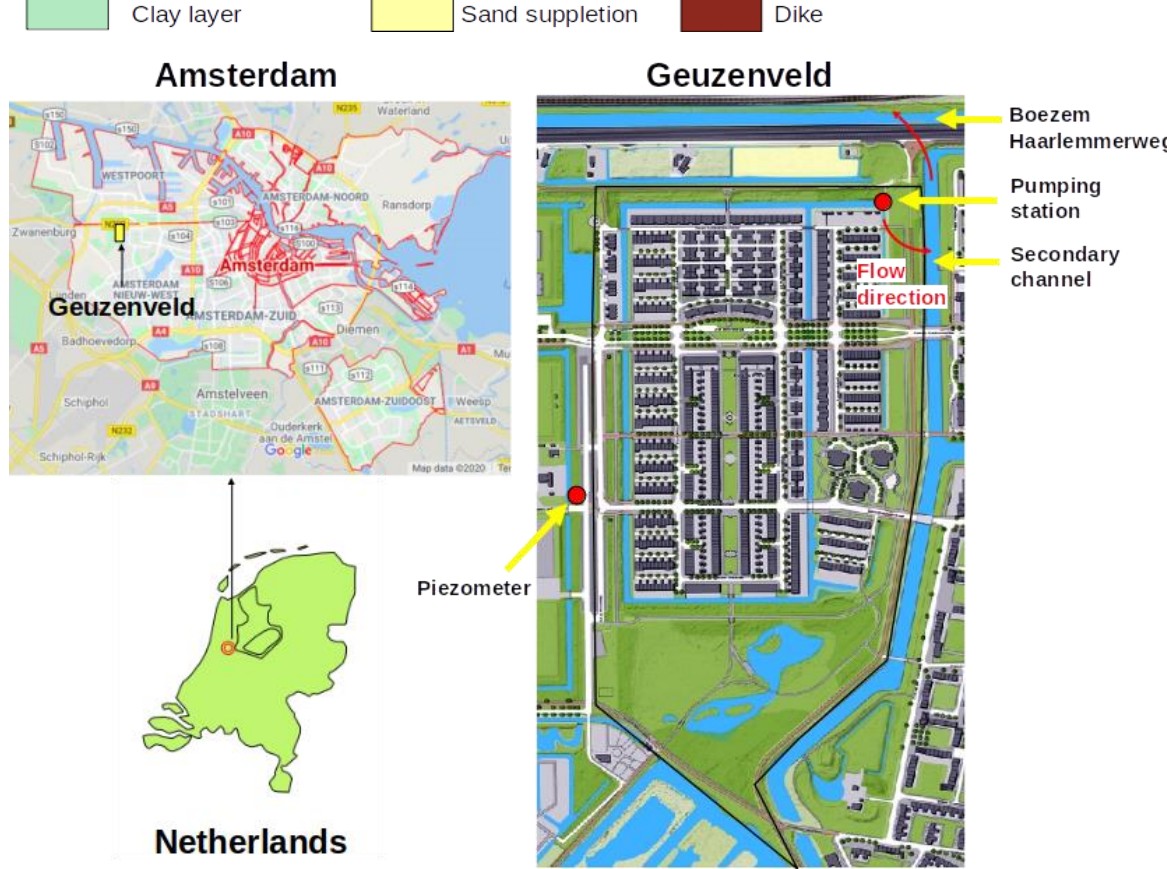

 **Figure 1 Location of polder Geuzenveld (source: © Google Maps ) and its landscape cross section and rain water and**

**groundwater drainage system**
**2.2 Monitoring network setup**
**2.2.1 High frequency monitoring**
A high frequency monitoring network was built on a temporary floating platform in front of the pumping station. The water
flowed around and underneath this platform to the pumping station when the pumps started working. One year time series of
NH$_4$-N (mg L$^{-1}$), TP (and ortho-P) (mg L$^{-1}$), turbidity (Formazin Nephelometric Unit, FNU), electrical conductivity (EC, µS/cm)
and water temperature (°C) were collected by the following equipment: a Sigmatax sampler combined with a Phosphax sigma
auto analyser for total phosphorus (TP), Amtax for NH$_4$-N combined with a Filtrax automatic sampler, a Solitax-tline sc for
turbidity (manufactured by: Hach Lange GmbH Düsseldorf, Germany), and CTD-Diver for electrical conductivity (EC) and
water temperature (manufactured by: Van Essen Instruments, Delft, The Netherlands). The monitoring frequencies were set
to 20 mins, 10 mins, 5 mins, 5 mins and 5 mins interval for TP, NH$_4$-N, turbidity, EC and water temperature, respectively.
The Phosphax sigma is an analogue analyser for the high precision determination of total phosphorus concentration in
accordance with EN 1189 Phosphormolybdenum Blue method. Samples are automatically taken through a Sigmatax sampling
probe and include suspended solids. Subsequently, the sample is ultrasonic homogenized before delivery to the Phosphax
sigma. It is digested by the sulphuric acid-persulphate method (APHAAWWA-WPCF, 1989), and analysed with a LED
photometer (at 880 nm) (Hach, user manual of Phosphax sigma, 2016).
Samples for NH$_4$ are prepared by a filtration system, Filtrax. It continuously extracts samples through two ultra-filtration
membranes (0.15 µm) plates. Particles get dispersed by a continuous aeration system near the surface of the membranes (The
aeration caused severe build-up iron precipitants on the plates). The samples are then delivered to Amtax sc for analysis. The
ammonium in the sample is first converted to gaseous ammonia. Only the NH$_3$ gas passes through the gas-permeable membrane
of the electrode and is detected. This method guarantees a wide measuring range and is less sensitive to other compounds
compared to methods that make use of an ion-selective electrode (ISE). The Amtax sc in our study was calibrated automatically
at 22:00 every 24 hours before September 2016, every 48 hours thereafter. Maintenance work was conducted weekly as the
tubes were easily blocked by iron precipitates (Hach, user manual of Amtax sc, 2013).
The Solitax-tline sc sensor is a turbidity sensor with dual-beam optics and added backscatter. The measuring principle is based
on a combined infrared absorption scattered light technique that measures the lowest turbidity values in accordance with DIN
EN 27027 just as precisely and continuously as high sludge contents. Using this method, the light scattered sideways by the
turbidity particles is measured over an angle of 90° (Hach, User manual of Solitax sc, 2009).
The monitoring period of NH$_4$ and turbidity is from 2016-05-10 to 2017-06-16. Time series of phosphorus were obtained from
2016-05-23 to 2017-06-16. Electrical conductivity and temperature data are from 2016-06-10 to 2017-06-15. The NO$_3$ analyser,
Nitratax, time series consistently showed an artificial drift and proved to be unreliable in our field setting, possibly due to
biofilm accumulation in combination with iron oxides precipitation (see discussion). All the equipment outputs were integrated
into one wireless station. The monitoring station was shut down several times by lightning, so an electricity restart program
was also applied in this network. It worked for all equipment except for the Phosphax, which had to be restarted manually after
a black out.
Precipitation (hourly) and Evapotranspiration (daily) data were downloaded from the Schiphol KNMI station which is about
2 km away from Geuzenveld. Hourly pumping activity and surface water level data were obtained from Waternet, the water
authority of Amsterdam.

### 2.2.2 Low frequency monitoring

Since 2006, Waternet has monitored the water quality with a frequency of 12 times per year by sampling at the pumping station of Geuzenveld. Between 2016 and 2017, the sampling frequency was increased to twice per month. We selected the following parameters from the routine monitoring campaign: (1) EC, $NH_4$-N and TP to fill in the gaps in the continuous time series, and to verify and monitor the potential drift and offset of the high frequency data and (2) pH, $O_2$, $HCO_3$, $NO_3$, TN, Kjeldahl-N, suspended solids (detail of methods are described by Yu et al, 2019), chlorophyll-a, and transparency for further understanding the biogeochemical processes. Organic-N was estimated by subtracting $NH_4$-N from Kjeldahl-N.

Bi-weekly total iron in the water column was analysed separately using ICP-AES (inductively coupled plasma-atomic emission spectrometry). Total Fe was analysed from samples to which $HgCl_2$ was added for preservation and that were stored in a dark and cool environment. To release all Fe that may have sorbed or precipitated during storage, we added 1 or 0.5 ml HCl in the water samples to dissolve eventual flocks. Then the samples were homogenized in an ultrasonic bath for 24h, mixed again to break down all the flocks. For extraction of all the Fe, we transferred 10 mL of the homogenized sample into a Teflon bottle, added 3.2 mL HCl : $HNO_3$ 3:1 , and stored in a stove at 90 °C for 24 hours. The final solutions were analysed by ICP-AES. Blanks were included and treated identical to samples.

### 2.3 Data processing and analysis

A correlation analysis between the high frequency and discrete monitoring data was applied to illustrate the reliability of the high frequency time series. Furthermore, the time series data were analysed at 3 time scales: annual scale, rainfall events (several days) and single pumping events (several hours). The relationships among the monitored parameters was explored by testing their correlations at each time scale. At the annual scale, a correlation analysis was applied to the complete time period and the wet and dry periods (definition in section 3.1.1). To discern the hydrological and chemical/biological attributes to the observed dynamics, a linear mixing model was introduced at the annual scale, assuming precipitation and groundwater seepage are the only water inputs, pumping and evapotranspiration are the only outputs, and pumping activity is the only way solutes leave the water system. In this model, we assumed a constant seepage rate. Accordingly, surface water level was calculated from:

$$\frac{dV}{dt} = \big(P(t) + S - E(t)\big) * A_{polder} - Pump(t) \tag{1}$$

$$L(t) = V(t)/A_{ditch} \tag{2}$$

*V* is total water volume in the ditches, *P* is precipitation, *S* is a constant seepage, *E* is potential evapotranspiration, *$A_{polder}$* is area of the polder, *Pump(t)* is water volume being pumped out with maximum capacity 216 $m^3$ $h^{-1}$, *$A_{ditch}$* the area of the ditches in the polder. *L* is surface water level in the ditches. Water level *L* determines the activation of pumping activity. Once *L(t)* exceeds the upper ranges of water level (start point, section 2.1), the pumps will start to pump until *L* goes below the stopping end (section 2.1) in the pumping scheme. Given the year-round seepage conditions throughout the polder, combined with an artificially drained subsurface, we assumed that the potential evapotranspiration was close to the actual evapotranspiration as no water shortages occur in our situation. In this study, we used the difference between groundwater head in the first aquifer and the surface water level (Figure 2A) to estimate a range of the seepage. The actual number of 2 mm per day was chosen based on the behavior of the mixing model and calibrated using the measured surface water levels (Figure S1).

A complete mixing of solutes was assumed in the model, which means that seepage, ditch water and precipitation mix instantaneously when they enter the surface water. A delay from precipitation to run-off/drainage and to ditches was not specifically considered.


$$\frac{d(VC)}{dt} = S * A_{polder} * C_{gw} + P(t) * A_{polder} * C_P - Pump(t) * C(t) \tag{3}$$


$V$ is the ditch water volume given by equation (1), $C(t)$ is solute concentration at time t, $C_{gw}$ is the average groundwater
concentration, $C_p$ is the average concentration in runoff.
In our study area, the EC is a useful water quality parameter for describing the mixing processes between groundwater and
runoff water, as the EC represents the end members of the mixing: groundwater with an high EC (1750 µS/cm) and runoff
water (100 µS/cm) with a low EC (see also Yu et al., 2019). Moreover, we assume that EC behaving as a conservative tracer
as the EC is highly correlated with the Cl concentration ($R^2 = 0.71$, $p$-value $< 0.05$) and the temporal patterns of EC and Cl are
very similar (see supplement Figure S2). In the model, seepage rate was adapted to get the best fit between the modeled and
the measured EC. The calibrated seepage rate was 2.0 mm d$^{-1}$. Compared to EC, nutrients are highly reactive solutes and thus
can vary a lot along their flow routes due to biogeochemical processes. The model provided a tool to simulate concentration
dynamics under the assumption that EC, $NH_4$ and TP were conservative. The simulated concentrations of EC, $NH_4$-N and TP
were plotted together with the high frequency measured time series. A comparison between the modeled and the measured
results was performed by using correlation analysis.
The average concentration of EC in groundwater was set equal to the average of the sampling survey, which was 1750 µS/cm
(including both deep and shallow groundwater, Yu et al., 2019). For the $NH_4$ and TP concentration data, we chose the
measurement from a drain sampling point (Drain 3, Yu et al., 2019) in the middle of the polder as the non-disturbed
groundwater collected by the drains in this area of the polder. They were 8.1 mg N L$^{-1}$ and 1.6 mg L$^{-1}$ respectively. The starting
(01-06-2015) concentrations were 1200 µS/cm, 4 mg L$^{-1}$, and 2 mg L$^{-1}$ for EC, $NH_4$, and TP respectively. The model was not
sensitive to the selected end-member values.
The time series data were further analysed at shorter scales: rain event scale and pumping event scale. Four rain events were
selected according to the dilution extent of EC, defined as an EC value reduced by over 35%, they were: 10-06-2016 ~ 15-07-
2016, 15-08-2016 ~ 26-09-2016, 10-11-2016 ~ 05-01-2017, and 20-02-2017 ~ 10-04-2017. These four events covered both
EC dilution during rainfall and the recovery afterwards in different seasons. We selected 4 representative pumping events to
present the response of EC, $NH_4$, TP, and turbidity to the pumping activities. Those events were in 15-07-2016 ~ 17-07-2016,
27-10-2016 ~ 29-10-2016, 20-12-2016 ~ 22-12-2016, and 05-05-2017 ~ 07-05-2017. Correlation analysis was as well applied
to each event at the corresponding two time scales, averaging over whole days for precipitation events and over hours for
pumping events. Data processing and analyzing were performed using Rstudio (R version 4.0.2) and time series package "xts".
**3. Results**
**3.1 Annual pattern of meteorological, hydrological, and water quality time series**
**3.1.1 Meteorological and hydrological conditions in polder Geuzenveld**
To explain the time series (Fig. 2), we distinguish between dry/wet periods and dry/wet seasons. The wet and dry periods (days
to weeks) are represented by a water surplus (light blue color in Fig.2B, daily evapotranspiration < daily precipitation) or a
water deficiency (dark blue in Fig.2B, daily evapotranspiration > daily precipitation). We defined the wet and dry seasons
based on water surplus and deficit. The average net rainfall (the water surplus/deficit in Figure 2) is 1.4 mm/d for the period
of 01-10-2016~15-03-2017, and -0.8 mm/d for the rest. Subsequently, we statistically analysed the difference between these
two periods for multiple parameters. Table 2 shows the mean of each parameter for the wet and dry seasons and their
significance test results. The wet and dry seasons means are significantly different for all parameters, but the EC.

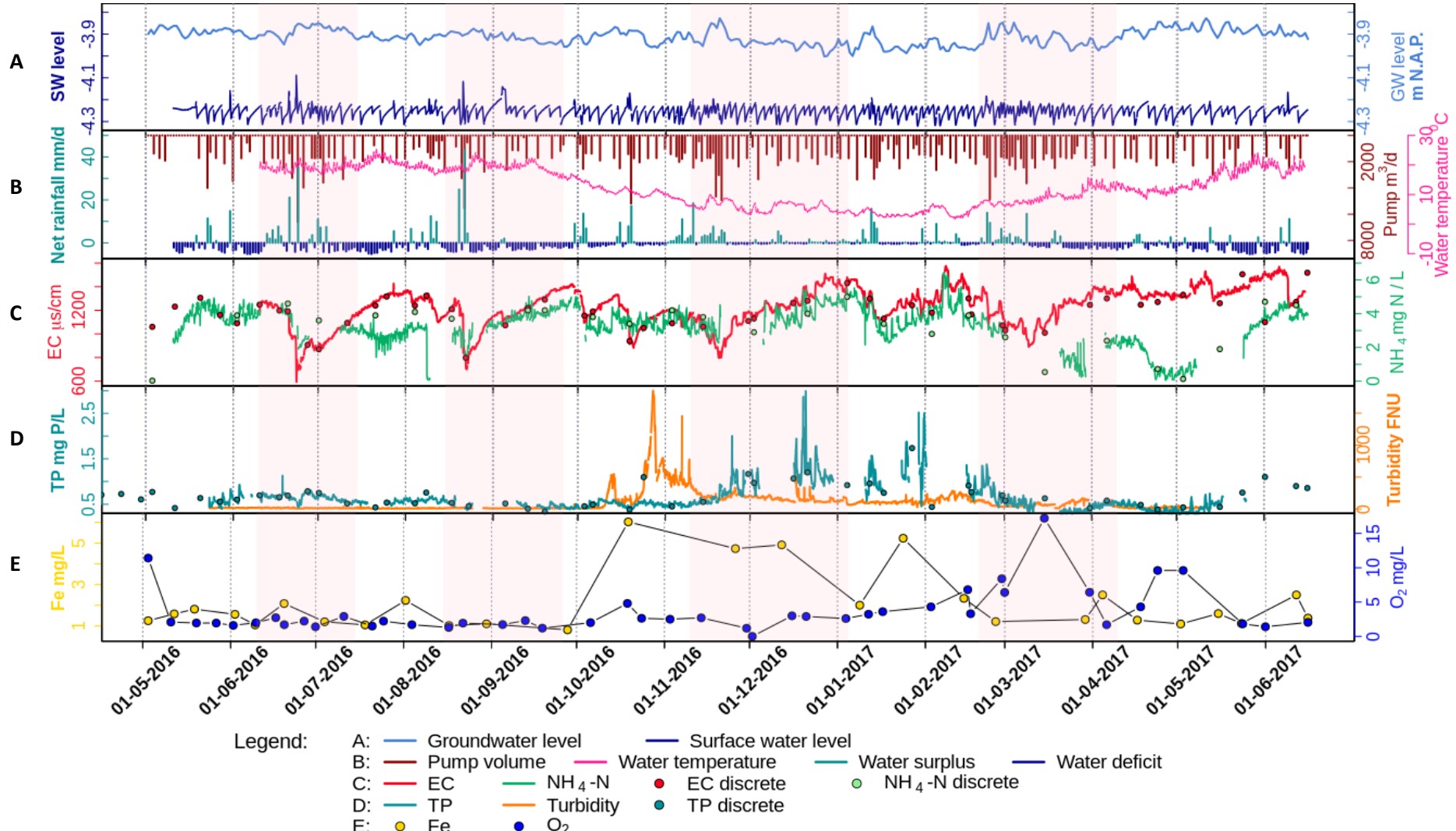


**Figure 2** Time series of (A) surface water level (SW level) and groundwater level (m NAP), (B) net rainfall (daily water surplus (+) (lightblue) and deficit (-) (darkblue), mm d$^{-1}$)

and daily pumping volume (Pump m$^3$ d$^{-1}$), (C) hourly time series of EC (µS/cm) and NH$_4$ -N (mg N L$^{-1}$ ), and (D) hourly TP, turbidity, (E) discrete samples of Fe (total iron in

water column) and O$_2$ concentrations (mg L$^{-1}$ ). The dots in (C) and (D) are the corresponding discrete sampling data, which are plotted to show their close match to the

continuous time series data, as well as to fill in the gaps. All data were monitored at the pump station. The transparent pink blocks are the selected rain events for further

analysis in section 3.3. See Table S1-S3 for the correlation tests performed on the dataset.

**Table 2 The mean of each parameter, and the significance for the wet and dry seasons**

|  | Net rainfall* mm/d | Pump volume* m³/d | Water temperature* °C | EC µs/cm | NH$_4$* mg N/L | TP * mg P/L | Turbidity* FNU | Fe* mg/L | O$_2$* mg/L |
|---|---|---|---|---|---|---|---|---|---|
| Wet | 1.4 | 1050 | 6.7 | 1212 | 3.7 | 0.8 | 197 | 3.4 | 4.3 |
| dry | -0.8 | 712 | 17 | 1252 | 3.0 | 0.5 | 15 | 1.5 | 3.3 |

* $p < 0.05$[1]

Over the whole monitoring period, the water temperature ranged between 2 to 26 °C. From June to mid-September 2016, the
temperature remained above 18 °C, then declined to become lower than 10 °C at the end of October. The following four months
(November to February) were the coldest. Especially in January and February 2017, during which the water temperature
dropped to below 3 °C. By the end of February temperatures started to rise again to reach 10 °C by the end of March 2017.
The surface water level in Geuzenveld has been maintained between -4.31 and -4.1 m NAP, strictly regulated by pumping
(Fig.2A). After the pumps stopped, the surface water level recovered faster during the wet season (between October 2016 and
March 2017) than during the dry season. Similarly, the shallow groundwater level positively corresponded to the precipitation
and negatively to the daily accumulative pumping volume. The phreatic groundwater level in Fig.2A (light blue) was from
one of the piezometers, which lies right outside of the polder (Figure 1, 52°22'46.0"N 4°47'15.6"E). In contrast to the constant
surface water levels (Fig.2A, dark blue), the shallow groundwater had relatively low levels in the wet season compared to the
dry season. This is related to the water level regulation of the boezem Haarlemmerweg with higher levels in summer than in
winter (https://www.rijnland.net/actueel/water-en-weer/waterpeil). Phreatic water levels were consistently 20-40 cm higher
than the surface water level in the polder, which confirms the continuous groundwater seepage into the surface water system.
**3.1.2 Annual water quality patterns**
The Pearson's coefficients of determination (R²)~~coefficients of determination (R² "Pearson" method used)~~ between the high
frequency data and the routine discrete sampling data from the water authority are 0.88 for EC ($p$-value < 0.05), 0.92 for NH$_4$
($p$-value < 0.05), and 0.97 for TP ($p$-value < 0.05). The scatter plots between the high and low frequency measurements are
shown in Figure S7.
During a rainfall event, rain and runoff from pavements and roofs, which were collected by a separate drainage system, directly
fed the surface water (Fig.1). Distinct rainfall events cause a strong dilution pattern of both EC and NH$_4$ (in Fig.2C). The EC
ranged from 600 to 1500 µS/cm. In general, during rainfall events, the EC declined because of dilution, while, after the events,
EC gradually rose back up to around 1500 µS/cm. The duration of this process, i.e. *recovery time,* was longer in the wet season
than in the dry season. A similar pattern of dilution and recovery is also visible for NH$_4$, especially for the period August 2016
– March 2017, where NH$_4$ shows a very similar response as EC (Table S2, wet season, $R^2 = 0.73$ ), although with somewhat
larger day to day fluctuations. However, a contrasting pattern without NH$_4$ recovery occurred twice: from the middle of June
to the end of August 2016 and from the middle of March to the middle of May 2017. During these periods, concentrations of
NH$_4$ were considerably lower and deviated from the slope of the EC pattern. NH$_4$ decreased from around 4 mg L$^{-1}$ to around
2 mg L$^{-1}$ between the middle of June to the end of August 2016, but the continuous NH$_4$ measurements are not supported by
the discrete samples which follow the EC pattern more closely. During the second period from March to the middle of May
the deviation from the recovery pattern is more pronounced, and NH$_4$ concentrations dropped to almost 0 mg L$^{-1}$ and started
recovering from the beginning of May. This pattern is fully supported by the available discrete samples. During the same

---

[1] Wilcoxon rank-sum test. The tests were performed in Rstudio (version 3.6.1), wilcox.text() in package "stats".

period in 2016 the high-frequency monitoring had not yet started, a single $NH_4$ discrete measurement is available for the $2^{nd}$
of May, that seems to reveal a similar pattern in the spring of 2016.
Both TP and turbidity showed contrasting patterns during the wet and dry seasons (Fig. 2D). Turbidity stayed below 60 FNU
during the dry season until October and rapidly increased after a first rain event to 500 FNU (more details refer to Figure S3
in supplementary information). A drop to about 200 FNU occurred right after this first peak, which seemed to correspond to
excessive precipitation and a large pumping volume (Fig.2B). Soon after, turbidity went up again and peaked at 1800 FNU.
Turbidity leveled off towards values around 200 FNU for the rest of the wet season and dropped below 60 FNU from April
2017 onwards.
TP concentrations were significantly higher during the period between 15-11-2016 and 01-03-2017 than the rest of the time
(p-value < 0.001, Figure S5), during which TP fluctuated around 0.5 mg $L^{-1}$, but always below 1 mg $L^{-1}$. During the wet season
with the low temperatures (Table S2, $R^2$ = -0.68), TP almost constantly stayed above 1 mg $L^{-1}$ and even reached values of
about 3 mg $L^{-1}$ in December. Although there were large gaps in the TP time series during this period, the high TP concentrations
appear to have been diluted by rain events, for example the event at around January $10^{th}$, 2017. Most discrete samples
measurements of TP matched well with values from the high frequency time series (Fig.2D, Table S1, $R^2$ = 0.88).
Total-Fe concentrations were most of time lower than 2 mg $L^{-1}$ (Fig. 2E), but for the wet season concentrations were higher
and reached up to about 6 mg $L^{-1}$. The initiation of Fe increases at the beginning of the wet season coincided with that of
turbidity (Fig.2D and Table S2, $R^2$ = 0.72). Upon the increasing temperature in March 2017, total Fe concentrations dropped
back to below 2 mg $L^{-1}$ (a negative correlation between temperature and Fe is shown in Table S1). Dissolved $O_2$ concentrations
were generally low in the water column; i.e. usually below 5 mg $L^{-1}$. Concentrations of over 3 mg $L^{-1}$ were only found in
March, April and May.
**3.2 Model of water quality time series based on water balance**
A simple fixed-end-member mixing model was used to reconstruct the conservative mixing of EC, $NH_4$, and TP. The simulated
and the measured EC, $NH_4$, and TP are plotted in Figure 3. The correlations between the modeled and measured results are
shown in the supplementary information (Table S4-S6). Potential processes that might deprive or enrich nutrients relative to
the conservative mixing process along the flow routes were inferred from the discrepancies between the modeled and the
measured data. Figure 3(A) and Table S5 show that the predicted and observed EC dynamics agree reasonably well from May
to November $20^{th}$, 2016 ($R^2$ = 0.91). After that, the conservative mixing approach underestimated the EC but the main dynamics
and the amplitudes were still reproduced (Table S6, $R^2$ = 0.82); as groundwater is the only contributor to the high EC due to
the seepage of quite mineralized, slightly brackish water, the model must underestimate the seepage flux from November $20^{th}$,
2016 on. Overall, the observed dynamics of EC are consistent with mixing of high EC seepage water with low EC runoff water
(coefficient of determination between the modeled and measured EC is 0.65 over the complete period, Table S4).
The dynamics of measured $NH_4$ concentrations show close resemblance to the model results, especially during the wet season
(01-10-2016~15-03-2017). Clearly, $NH_4$ is diluted during the rain events and a gradual increase of $NH_4$ starts after each rain
event during the wet season showing slopes that resemble the model reconstruction. Over the whole period, measured $NH_4$
concentrations were overestimated by the model, indicating that some $NH_4$ is probably lost due to non-conservative processes.
This is especially true for the spring season of 2017, where $NH_4$ concentrations must be controlled by additional processes.
Concentrations of TP are generally far below the conservative model reconstruction, except between the end of November and
the beginning of March. During this particular period the minimum measured TP concentrations are captured nicely by the
conservative model, ~~however~~ but the distinct peaks up to 3 mg $L^{-1}$ are not ~~are not captured by the model and must have different~~
~~physical or chemical processes determining them~~.

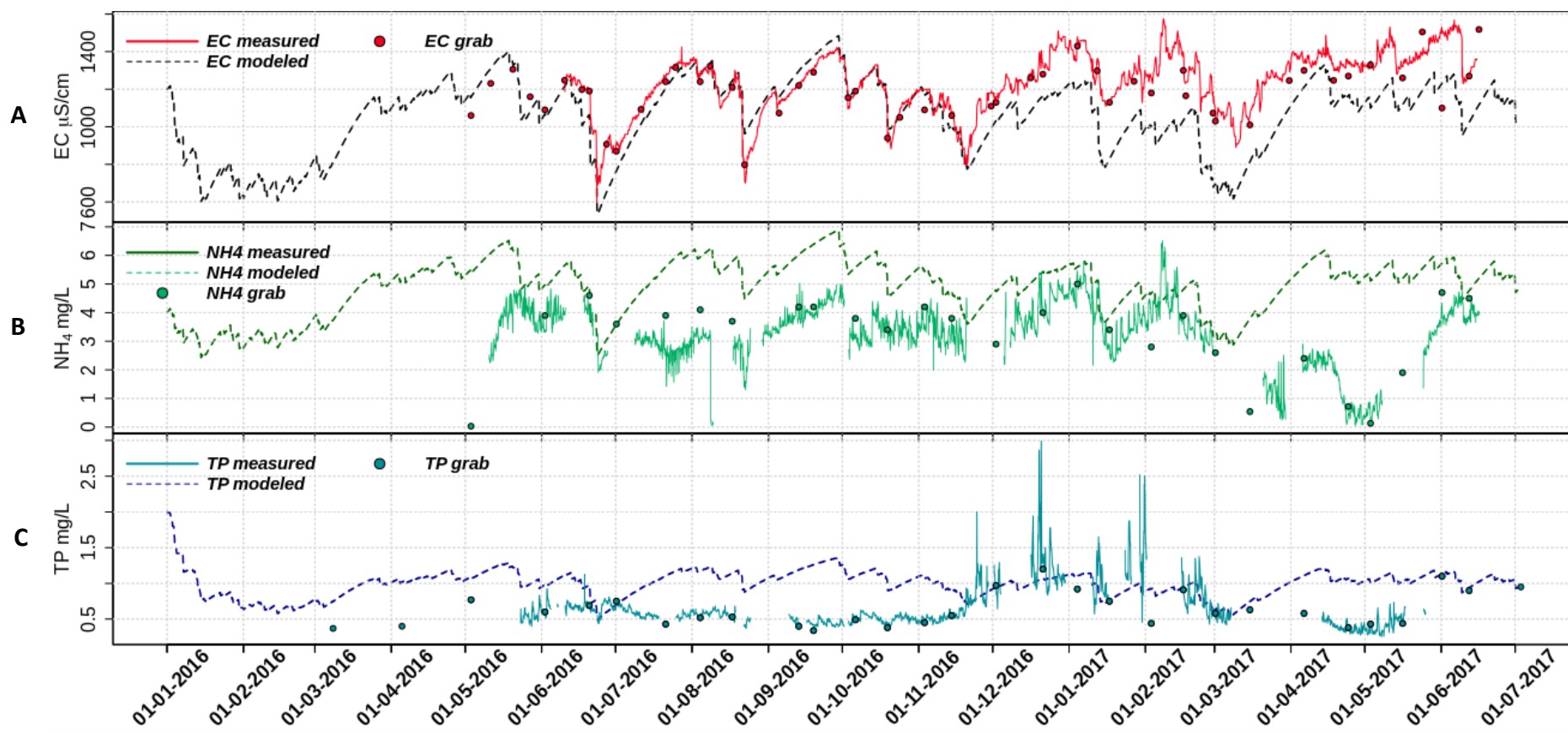

**Figure 3 Plots of fixed-end-member mixing model predicted (A) EC, (B) NH₄ and (C) TP with their measured time series data and the discrete sampling results. See Table S4-**
**S6 for the correlation tests performed on the dataset.**

### 3.3 Water quality responses to single events analysis

To elucidate the response pattern of water quality to precipitation and pumping activity, we selected four major events (Fig. 2 (4 pink shades) and Figure 4) and four pumping events (Figure 5). The former events were chosen according to their clear dilution pattern of EC (Fig.4), while the latter were pumping events without occurrence of rainfall (Fig.5). All seasons were covered, including some of the wet and dry periods.

### 3.3.1 Rainfall events

EC and $NH_4$ showed clear dilution and recovery patterns during all events, while the pattern was not clear for TP and turbidity (Fig.4). The extent of dilution of EC appears to depend on the precipitation intensity. Rainfall during the recovery period determined how long it took to recover back to the highest level. The short but intensive rainfall during dry season events 1 and 2 reduced EC rapidly from around 1300 to around 700 µS/cm, while the recovery took about 1 month. Events 3 and 4 had less rainfall and dilution of EC was less (from about 1300 to about 800 µS/cm) and recovery took more than one and a half month in event 3, during which multiple small events occurred. The dilution patterns of the $NH_4$ in events 1 and 2 were similar to those of EC ($R^2 = 0.86$ and 0.83, respectively, Table S7 & S8) and show resemblance for event 3 ($R^2 = 0.75$, Table S10). Moreover, a direct negative correlation between $NH_4$ and rain intensity supports this dilution effect for event 2. Due to the data gaps of $NH_4$ in  event 4 we cannot completely describe the pattern of $NH_4$ for this one, but it corresponds with that start of reduced $NH_4$ which was described in sections 3.1 and 3.2.

The response of TP was generally not related to the intensity of rainfall and pumping, except for event 3 during the wet period. Dilution effects, as were observed for $NH_4$, were not observed for TP for events 1, 2 and 4. During the wet season event 3, TP concentrations show negative correlations with precipitation and pumping intensity ($R^2 = -0.79$ and -0.59, respectively, Table S9) and correspond with decreasing turbidity. Event 4 marks the transition between the wet and dry season and the drop in TP coincides with the drop in $NH_4$, independently from individual rain storms during the dry season.

During the dry season (with event 1 and 2 included) turbidity always stayed below 50 FNU. Turbidity sometimes showed single peaks which are likely related to disturbances of the floating platform by wind and should probably be treated as false signals. Turbidity is more variable and has higher variance for wet season events 3 and 4, which corresponds with the findings of the annual scale analysis (section 3.1.2). During event 3, turbidity varied between 100 and 500 FNU. Although clear relations exist between Fe, TP and turbidity, all higher during the wet season (Figure 2, Table S2), these are not clearly reflected at the scale of individual precipitation events. Simultaneous peaks of TP and turbidity occur that are not easily related to the weather conditions in November and December but TP and turbidity show contrasting signals at the start of the event. The turbidity clearly decreases during rain storm event 3 and at the start of event 4. This change is not reflected by the correlation at the total event scale (Tables S9 and S10) but obvious when studying only the time scale of the decreasing limb of the EC dilution. Event 4 coincides with the transition to the spring season in 2017, showing decreasing EC, TP and turbidity in the last rains of the wet season and a strong decrease of $NH_4$ and increase of turbidity when conditions dried up and temperatures rose.

### 3.3.2 Pumping events and day and night pattern

The selected pumping events covered four seasons: summer (2016 July, event 1), autumn (2016 October, late autumn, event 2), winter (2016 December, event 3) and spring (2017 May, event 4) (Fig.5). While the effects of pumping on EC are rather small, TP, $NH_4$ and turbidity are all affected by pumping. The effects of pumping appear to be different for events in different seasons; turbidity for example increases during pumping in July and December but decreases in May. The increase during the December pumping is especially marked ($R^2$ Pumping intensity versus Turbidity = 0.77, Table S13). TP decreases during pumping in July ($R^2 = -0.67$) and October and increases in May ($R^2 = 0.6$). Event 2 seems to have started a major drop in turbidity (more than 1000 FNU) that continued some time after pumping.


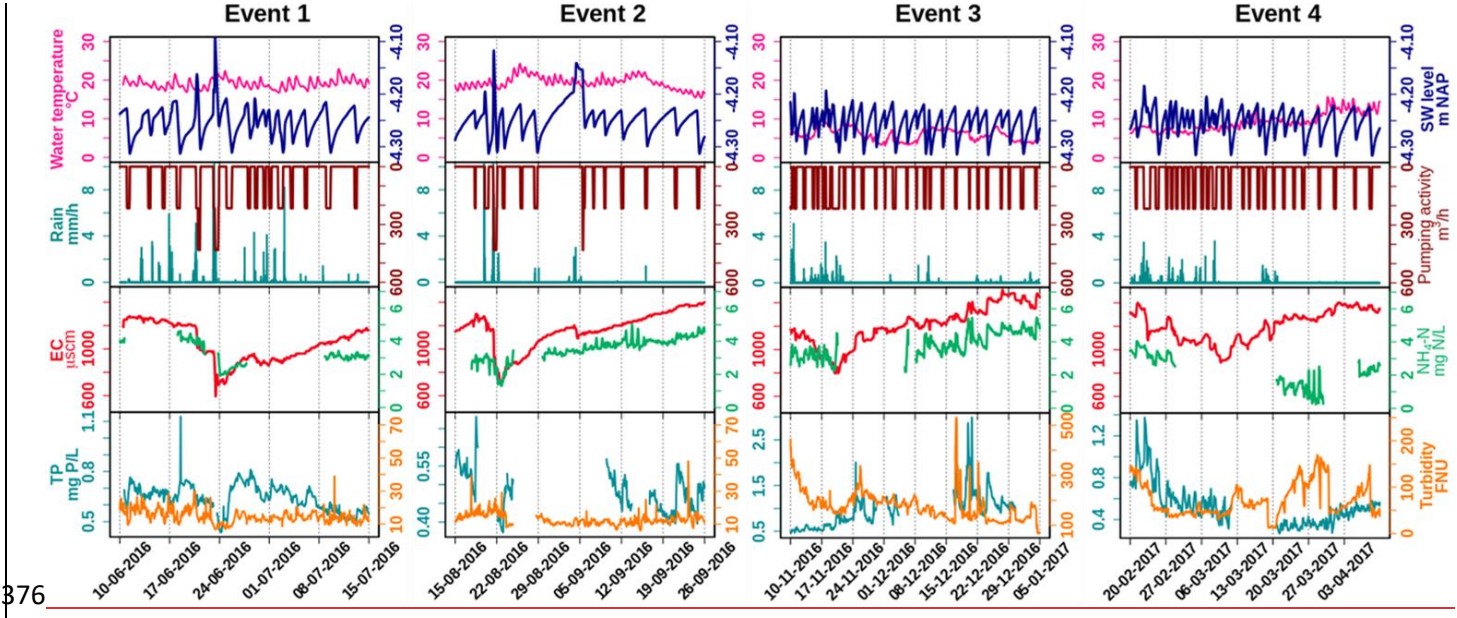


**Figure 4 Selected events showing dilution and peaks of water quality parameters, with hourly precipitation (mm/h)**
**and hourly pumping activity (m³/h). Note that between events different scales of TP and turbidity were used to reveal**
**the dynamics.** See Table S7-S10 for the correlation tests performed on the dataset.

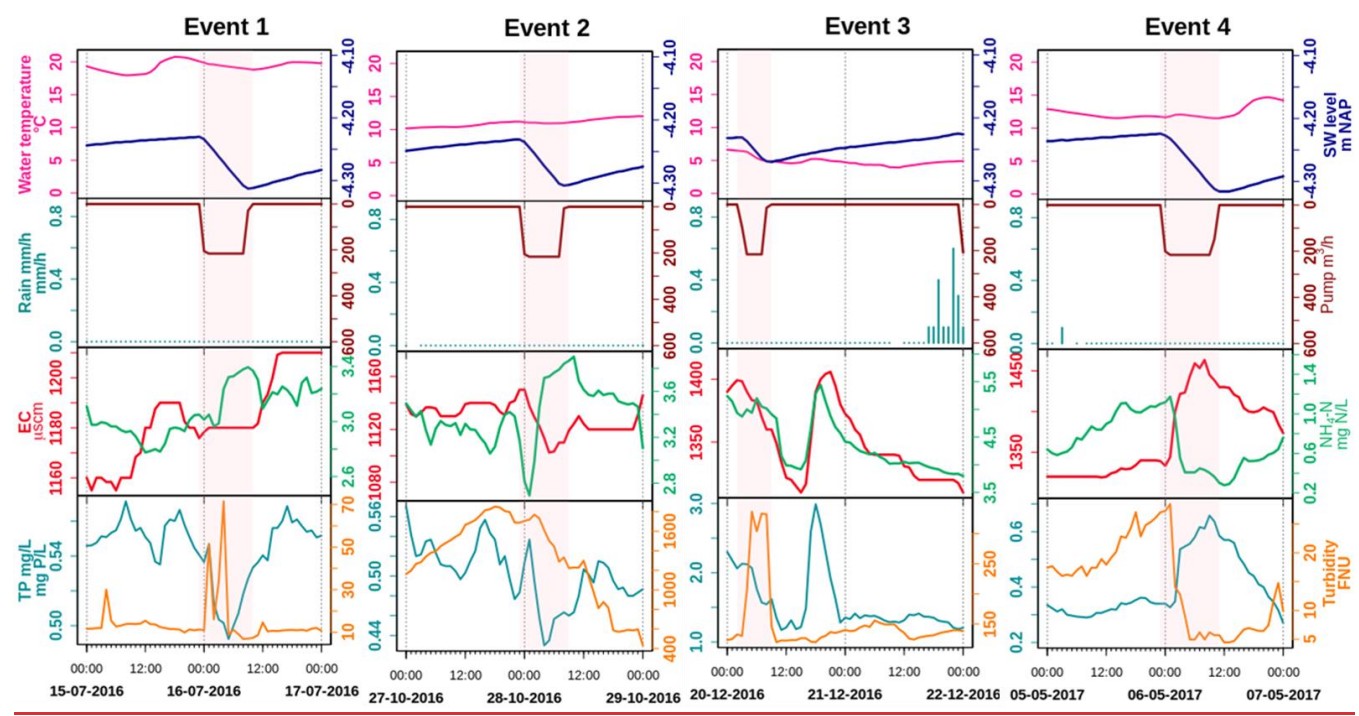

**Figure 5 Pumping and pumping effect patterns on water quality, blue blocks represent the pumping duration.** See
Table S11-S14 for the correlation tests performed on the dataset.
**4. Discussion**
This study aimed at understanding the dynamics of N and P fluxes from the low-lying urban polder of Geuzenveld to
downstream surface waters in order to eventually support water managers to mitigate eutrophication. Based on our high-
resolution water quality measurements, we found that the surface-water chemistry at the polder outlet pumping station is
governed by a complex combination of hydrological mixing and biogeochemical processing. In the following discussion, we
start with the presentation of the relatively straightforward dilution behavior of EC, followed by adding the impact of primary

production (i.e. algae growth) for understanding the NH$_4$ concentration patterns, and benthic primary producer and iron chemistry for understanding the turbidity and TP concentration patterns.

**4.1 Hydrological mixing between groundwater and rainfall**

In a highly manipulated low-lying urban catchment like Geuzenveld, mixing between rainwater and groundwater in the ditches is fast due to the high fraction of impervious area and the installation of both a rainwater and a groundwater drainage system that transport these contrasting water types efficiently to the ditches (Yu et al., 2019; Walsh et al., 2005). Runoff in Geuzenveld has EC of about 166 µS/cm (Yu et al., 2019), which is lower than the groundwater EC (1746 µS/cm on average). As a relatively conservative water quality parameter (Figure S2), mixing between rainwater and groundwater should be the main process for EC. This presumption is supported by the agreement between the modelled and the measured EC dynamics for the period between May to November 2016. Precipitation events diluted the EC values at the pumping station, and the magnitude of dilution depended on the intensity of precipitation; heavy rainfall resulted in low EC values (Fig.2D and Fig.4). In periods with the absence of rainfall, the EC values follow a recovery curve that resembles a linearly mixed reservoir with concentrations increasing to values that approach the EC of the continuous groundwater supply of around 1500 µS/cm. After November 2016, ~~the conservative mixing approach underestimated the~~ EC was underestimated by the model.~~but the main dynamics were still reproduced and the amplitude of the EC dynamics remains similar to the model results, except for the short period Nov 20th to Dec 1st, 2016. Starting around~~ The sudden increase of the measured EC around Nov 20th~~, the EC started to increase relative to the dry season before. It~~ coincides with an intensive pumping event after the first intensive rainfall ~~event~~ that happened after a prolonged period of cumulative water deficit. This may be related with a first flush from the drain system that starts to be activated more strongly, thus removing clogged material and lowering the overall resistance of the drain system for shallow and deep groundwater inflow (van der Velde et al., 2010). It suggests that this triggered the inflow of somewhat more mineralized groundwater relative to the period before, creating a shift in the EC towards ~250 µS/cm higher values that continued during the remainder of the monitoring campaign. It appeared that it raised the EC, but did not change its ~~the~~ amplitude or dynamics ~~of the EC~~ during the remainder of that period (Fig. 2 and 3, Table S6). The elevated EC may alternatively due to the application of road salts in winter which starts from November.~~An alternative reason for the higher EC starting from November, 2016 on, would be the application of road salts during the winter period. Although freezing conditions occurred from November onwards,~~ But we did not find any evidence for the prolonged effects of road salts, as the chloride concentrations in the grab samples only showed two higher measurements, one in December 2016 and one in January 2017 (see Supplement, Figure S2.)~~. So, overall, the observed dynamics of EC are consistent with mixing of high EC seepage water with low EC runoff water.~~

The mixing process can explain part of the dynamics of NH$_4$ and TP in the wet season, but insufficient for explaining the dynamics during the dry season due to the presence of biological and chemical processes. ~~During winter, mixing can also explain the dynamics of NH$_4$ and TP (Fig.3).~~ Compared with groundwater, which carries around 8 mg L$^{-1}$ NH$_4$ and 1.6 mg L$^{-1}$ TP, rain and runoff have much lower nutrient concentrations, which makes groundwater the main nutrients source (Yu et al., 2019). Nutrients derived from groundwater mix with rainwater in the ditches through direct seepage and the efficient groundwater drainage systems. Clearly, NH$_4$ is diluted during the rain events and a gradual increase of NH$_4$ starts after each rain event during the wet season showing slopes that resemble the model reconstruction. ~~Over the whole period, measured NH$_4$ concentrations are~~The ~~overestimated~~ overestimation of the modeled NH$_4$ in general~~by the model,~~ indicat~~es~~ing that some NH$_4$ ~~is a~~ probabl~~e~~y lost to transformation processes~~. This is~~, especially ~~true~~ in the spring ~~season~~ of 2017~~, where NH$_4$ concentrations must be controlled by other processes~~. Concentrations of TP are also generally far below the conservative model reconstruction~~, except between the end of November and the beginning of March. During this particular period the minimum measured TP concentrations are captured nicely by the conservative model, however~~. The distinct peaks up to 3 mg L$^{-1}$ are not captured by the model and must ~~have~~ be determined by different physical or chemical processes~~determining them~~. ~~While the~~

## 4.2 Primary production and nutrients

$NH_4$ dynamics during winter can be explained by mixing. However, biological processes are overruling the mixing process
during spring and summer. It resulted in lower measured $NH_4$ concentrations than modeled during this period. Studies have
shown that benthic and planktonic primary producers (e.g. phytoplankton) assimilate nutrients and are an important factor
controlling nutrient dynamics in rivers, lakes, and streams (Hansson, 1988; Jäger et al., 2017). In polder Geuzenveld, the
biological nutrient uptake is not only reflected in the time series data (Fig.2 and 3, Table S3) but is also evident in the monthly
measurements from the water authority for the period 2007-2018, as summarized in Figure 6 and Table S15-S19.

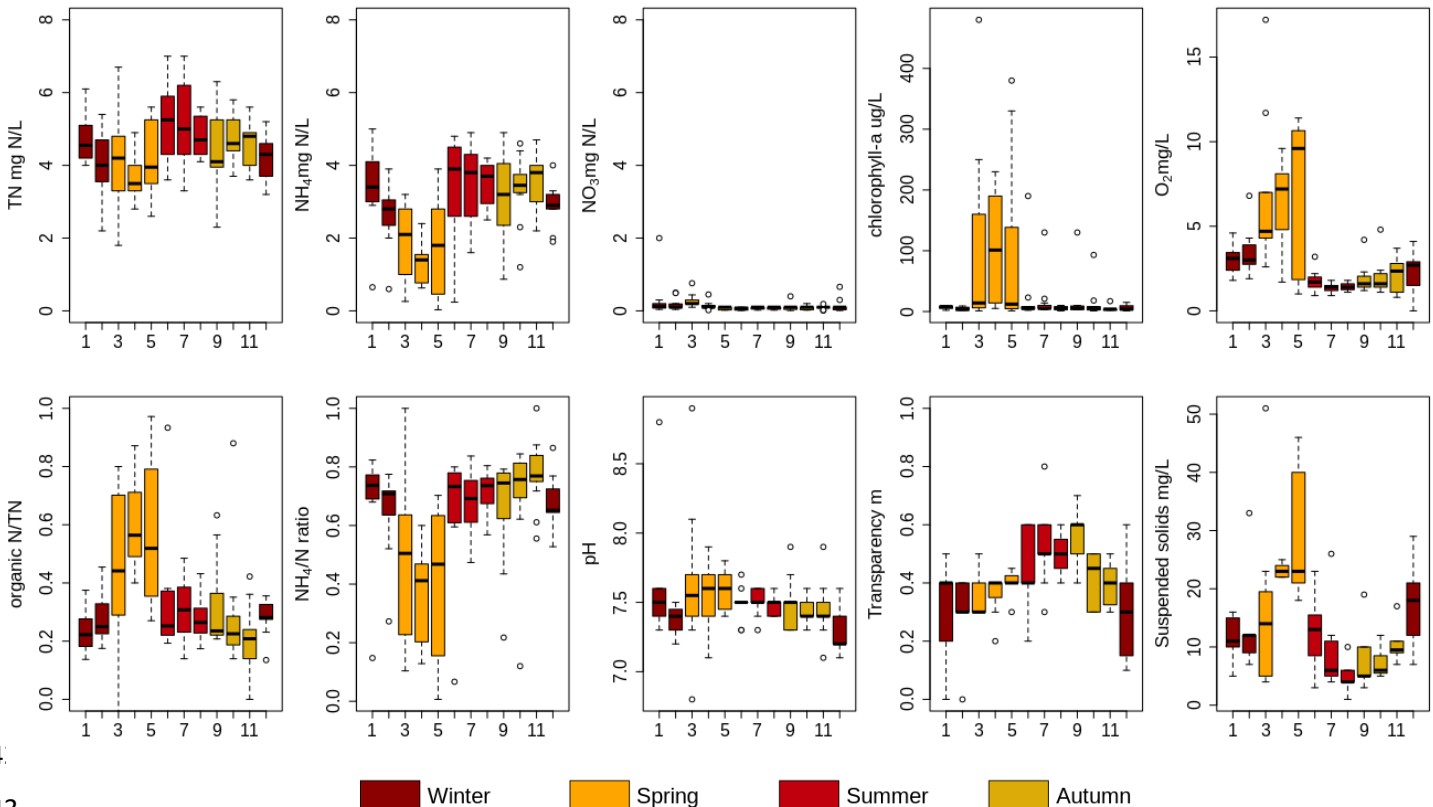



**Figure 6 Monthly measurements of TN, $NH_4$ -N, $NO_3$ - N, chlorophyll-a, $O_2$ organic N/ TN and $NH_4$-N /TN ($NH_4$/N)**
**mass ratio, $pH$ , water transparency, and suspended solids in Geuzenveld from 2007 to 2018. X axis is month. See**
**Table S15-S19 for the correlation tests performed on the dataset.**

The increasing availability of light (and temperature increase) during spring (Figure S6), induces growth of primary producers.
Growth of primary producers results in consumption of ammonium, phosphate and a production of organic-N, chlorophyll-a,
oxygen, and suspended solids, and led to a relatively higher pH because of the uptake of $CO_2$ (Figure 6, Table S16). These
patterns are also clearly reflected in the shift in the NH₄/TN and organic-N/TN ratios during spring (Figure 6). Primary
production occurs both in the water column by phytoplankton as well as by benthic algae. Macrophytes could in principle also
contribute, but they were absent in Geuzenveld. One of the structuring factors governing the relative importance of benthic
and planktonic primary producers is light availability: benthic algae and macrophytes tend to dominate in shallow and clear
waters, while phytoplankton is more likely to dominate in deeper and more turbid waters (Hartwig, 1978; Jäger and Borchardt,
2018; Petranich et al., 2018; Middelburg, 2019). Although our data do not allow conclusive determination whether benthic or
pelagic primary producers dominate, it appears that their relative importance varies with season.

These primary producers also compete for nutrients. Benthic primary producers have direct (macrophytes) or first (benthic algae) access to nutrients that seep up from the subsurface, while planktonic primary producers depend on nutrient supply from surface runoff and nutrients remaining after consumption by benthic primary producers. For example, Henry and Fisher (2003) found that benthic algae can remove up to 80% of nitrogen from an upwelling water source. As we stated above, nutrient-rich groundwater is the major source of N and P to surface waters in polder Geuzenveld. In addition, due to the shallow depth of the ditches, light reaches the bottom with the consequence that benthic algae can proliferate in this polder. These benthic primary producers might utilize the up-flowing nutrients from groundwater and intercept the nutrients from seeping further into the water column (Hansson, 1988; Pasternak et al., 2009). The increasing light availability and thus primary production during spring led to the nearly complete deprivation of $NH_4$ in the water column (Fig.2C).

Following the spring bloom, concentrations of chlorophyll-a (proxy for phytoplankton biomass) and $O_2$ dropped substantially, while $NH_4$ concentrations rapidly recovered to around 4 mg $L^{-1}$ in both the time series (Fig.2C) and the long-term monthly sampling results (Fig.6). Dissolved $O_2$ remained low (close to hypoxia) during the whole summer (below 2 mg $L^{-1}$) (Fig.2E and Fig.6), indicating that oxygen consumption by organic matter degradation and re-oxidation of reduced components from groundwater seepage outcompeted oxygen production from primary production. During summer, suspended solid and chlorophyll-a concentrations were low (Fig.6), indicating low biomass of plankton algae. Suspended solid and phytoplankton dominate light attenuation (Scheffer, 1998; Middelburg, 2019). Consequently, during this period, we observed an abrupt shift of the water regime from a turbid state to completely clear, as reflected in the high transparency from June to September (Fig. 6). The low biomass of phytoplankton might be due to N limitation as nutrients are intercepted by benthic algae at the sediment interface. An alternative explanation is that zooplankton grazing maintained phytoplankton biomass low (Strayer et al., 2008; Genkai-Kato et al., 2012).

Temperature and light reaching the sediment started to fall from September onwards (Figure S6), thereby reducing the intensity of biological activity, including $NH_4$ assimilation. Consequently, $NH_4$ started to behave conservatively again like EC (Fig.2 & Fig.3). The best fit between the modeled and measured $NH_4$ was from the end of November to the beginning of March, i.e during the winter period with lower light levels and shorter day lengths and very low primary production. The absence of primary production during winter, leads to conservative behavior of $NH_4$ governed by the mixing between groundwater and rain water.

**4.3 P binding and turbidity**

Iron chemistry is considered the dominant process governing the P dynamics in shallow groundwater fed ditches (Lijklema, 1994; Smolders et al., 2006; van der Grift et al., 2018). However, primary producers take up P for growth and at the same time release $O_2$ that regulates iron chemistry in lake water column (Table S1-S3, Spear et al., 2007; Zhang and Mei, 2015; Lu et al., 2016). This web of interactions likely controls P dynamics in these ditches.

From spring to autumn, TP concentrations were fluctuating around 0.5 mg $L^{-1}$, and the water had low turbidity (<50 FNU), thus high transparency allowing the growth of benthic algae that produce oxygen. Consequently, when P and Fe rich anoxic groundwater reaches the surface water-sediment interface, Fe oxidized into iron hydroxides in a short time (Van der Grift et al., 2014). P is then sorbed onto those Fe-hydroxides and retained in the sediments. Oxidation of reduced iron consumes $O_2$, contributing to the low $O_2$ conditions of the water column (Fig.2E). Moreover, it leads to the formation of a reddish-brown film of ferric iron (hydrous ferric oxide, Baken et al., 2013; van der Grift et al., 2018) on the bottom of the ditches, which can be seen in summer when the water was transparent. This slimy layer comprising iron hydroxides and benthic microbes can easily be resuspended and therefore act as a source of turbidity following perturbations by pumping, wind, rain or foraging fish, e.g. event 1 (Fig.5). Lu et al (2016) showed that co-precipitation of P with metal oxides was stimulated by periphytic biofilm activity that increased the water pH. Consistently, a relatively higher pH was also observed in our spring monthly samples (Fig.6).

From the late autumn onwards, turbidity and total Fe concentrations substantially increased compared to the rest of the time
(Fig.2, p value < 0.001 for turbidity and = 0.02 for Fe). Turbidity peaked first at 1800 FNU and stayed at a plateau of ~200
NFU during the rest of the cold and wet season. Total Fe in the water column reached to 6 mg $L^{-1}$ from below 1 mg $L^{-1}$. During
this period the water turned brownish and transparency declined (Fig.6). Iron-rich particles are the most likely source of
turbidity in freshwater (Lyvén et al., 2003; Gunnars et al., 2002; and Lofts et al, 2008). The suspension of these brownish iron
colloids was likely stabilised by the presence of the dissolved organic matter (Mosley et al., 2003; Van der Grift et al., 2014),
which (DOC) increased up to 18~33 mg $L^{-1}$ during events (Supplementary information Figure S4). In the late autumn, the
anoxic/oxic interface shifts from the sediment into the water column and so does the locus of colloid formation. The ditch
sediment, which had benthic algae activity releasing $O_2$ during spring and summer, became anoxic in the fall by the upwelling
of the anoxic groundwater. The anoxic seepage occurs year-round, but the production of oxygen by the benthic algae creates
an anoxic-oxic transition at the water-sediment interface, which leads to iron hydroxides precipitation in the slimy layer at the
bottom that disappears after the algae die off. As a consequence, Fe oxidation moved into the water column where the
conditions were relatively oxic (Van der Grift et al., 2014). Nevertheless, there was probably still enough Fe or other mineral
oxides, such as aluminum hydroxide (Kopáček et al., 2005), binding capacity in the sediment for the fixation of P, as P
concentrations remained low during this first turbidity peak. We suggest that the turbidity peak of 1800 FNU is caused by the
mineralisation of the benthic algae once they die off when light and temperature conditions decrease, combined with the shift
of ironhydroxide formation from the sediment-water interface to the water column. The latter process continues through the
whole winter season, until primary production restarts in spring (Figure 7).
During winter, temperatures were below 5°C, pH values were relatively lowered, and TP achieved its peak concentrations
(Fig.2D). During this period, iron reduction in the sediments continued, P bounded to iron oxides gradually got released along
with reduced iron (Li et al., 2016). In the water column, reduced iron was oxidized but much slower than during spring-autumn
due to the lower temperatures (Van der Grift et al., 2014), and dissolved P was incorporated in iron flocs with the result that
particulate P concentrations and turbidity became high (Table S1, $R^2$ for Fe~turbidity 0.81, TP~Fe 0.65; Table S2, Fe~turbidity:
$R^2 = 0.72$, , TP grab~Fe 0.79; Yu et al., 2019).
**4.4 Process synthesis**
With the presence of benthic algae, abundant organic matter and bacteria, the sediment functions as an active environment for
biotic processes (such as primary production and nitrification-denitrification-anammox) and abiotic processes (such as iron
oxidation). Figure 7 shows a conceptual diagram for the N and P dynamics in this lowland urban catchment during the four
seasons which summarizes our hypotheses about the functioning of the system.

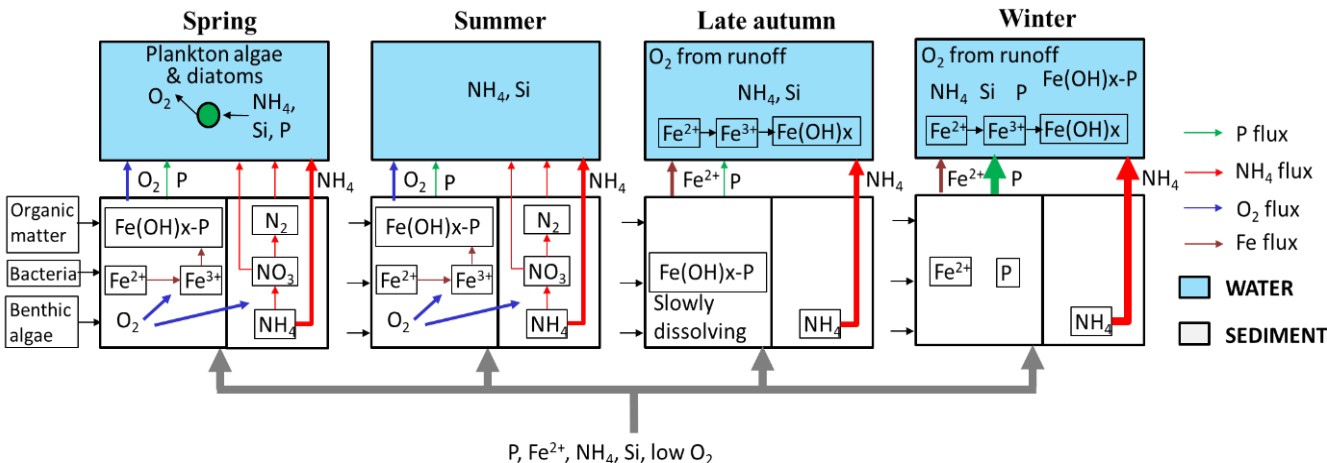


**Figure 7 Schematic representation of N and P dynamics in spring, summer, later autumn and winter. The thickness of the flow lines**
**represents the concentration magnitudes, the thicker the line, the higher the concentrations.**

*Spring:* The improved light (and temperature) conditions stimulated primary production and nutrient uptake (N, P, Si) by phytoplankton and benthic algae. The resulting oxygen production caused oxidation of reduced iron from groundwater and the formation of iron oxides at the sediment surface. P was mostly bounded to this particulate iron instead of being released into the upper water layer. In this period turbidity was relatively low, but suspended solids reached a high concentration due to the phytoplankton.

*Summer:* N and P were still being removed by biological processing, in particular by benthic algae. Phytoplankton biomass decreased because of competition for N or grazing activity. Benthic algae produced $O_2$, which in turn was used to oxidize all reduced iron reaching the sediment-water interface and P was still retained by iron hydroxides in the sediment. The water column was transparent (low TP and phytoplankton biomass) and relatively low in oxygen due to the continuous supply of anoxic groundwater, the mere absence of $O_2$-rich runoff, the oxidation process of Fe(II) and possibly by microbial organic matter decomposition during warm periods with relatively stagnant water.

*Late autumn:* Biological activity declined (colder and less light), and more $NH_4$ reached the water column. Moreover, the redox zone moved from the sediment-water interface into the water column (Van der Grift et al. 2014, 2016); the oxidation of Fe in the water column caused a peak of turbidity. P was still sequestered to minerals in the sediment.

*Winter:* During winter, $NH_4$ and TP showed the highest concentrations because of low biological activity. Iron oxides in the sediment dissolved under reductive and organic matter abundant conditions and released $Fe^{2+}$ and P into the water column increasing P concentrations therein. $NH_4$ and EC dynamics were primarily governed by the conservative mixing between groundwater and precipitation/runoff.

## 4.5 Event scale N and P dynamics

At the event scales, $NH_4$ and EC were reduced by dilution from precipitation/runoff. For P and turbidity there was no clear relation to precipitation events, except for events in late autumn and winter (e.g. Figure 4, event 3). The responses to precipitation and pumping events were different from those reported in the literature. Rozemeijer et al. (2010b) studied an agricultural catchment and found that rainfall events led to $NO_3$ decreases and P increases. Miller et al. (2016) observed $NO_3$ decreases during large discharges in an urban catchment. The lowering of turbidity in our urban catchment during the dilution periods that was associated with the winter events 3 ~~and~~ differs from the observations in literature (van der Grift et al., 2014, Rozemeijer et al., 2010b). In agriculture areas, turbidity usually peaks in response to rainfall events due to erosion and remobilization of sediments. In an urban, paved environment erosion may be limited and runoff water has a low turbidity. Moreover, in the case of turbid pre-event conditions, fresh precipitation water flushes away this turbid water. In addition, Yu et al. (2019) showed that precipitation runoff delivers particles and $O_2$ to the ditches. We suggest that this accelerates the further aggregation of the iron complexes; the resulting larger particles more readily settle to the bottom, causing a reduction of turbidity during the events itself (Fig. 4, EC dilution part of events 3 and 4).

In artificial lowland catchments, water systems are intensively regulated by pumping activity to prevent flood and drought. However, there is a substantial lack of knowledge about the possible consequences of such regulation on aquatic ecology and water quality. Peaks in P and turbidity by the activation of pumps was observed by Van der Grift et al. in their high frequency monitoring campaign in an agriculture lowland polder (Van der Grift et al., 2014 & 2016). This type of event scale dynamics would be easily missed in a daily or lower frequency sampling schedule, especially because pumping occurs almost solely overnight in our regulated catchments. As such, only a sampling schedule with 7 hours intervals (e.g. Neal et al. 2011) or high-frequency monitoring is able to catch the short-term dynamics (Van Geer et al. 2016).

Contrary to the findings of Van der Grift et al. (2014, 2016), the effects of pumping activity on N, P and turbidity dynamics were variable, depending on the season. During the phytoplankton bloom in spring, activation of pumps resulted in flushing and as a result reduced turbidity during the event (Fig. 5 event 4). Consequently, phytoplankton was transferred to the downstream channel and added to the total N pool in that system. In summer (Fig.5 event 1), the dead detritus and the layer of

iron compounds at the sediment surface were easily resuspended and contributed to turbidity peaks at the beginning of the
pumping, but the materials also re-sedimented almost immediately once the flow reached stability. Resuspension also resulted
in an increase of $NH_4$ in the water column which then was being pumped out (Fig.5 event 1). During late autumn, we observed
that the water was highly turbid (see also Yu et al. 2019) which we suggest to be caused by the formation of iron hydroxide
colloids in the water column, which is supported by correlations between Fe-grab and Turbidity ($R^2 = 0.72$, Table S2). We
explain the reduced turbidity after a precipitation event as a result of the activation of the pumps which caused the export of
the turbid water towards the receiving boezem in combination with aggregation of iron hydroxides in the water column and
subsequent settling of the aggregates due to the supply of new $O_2$-rich water (Fig.5 event 2, see also Van der Grift, et al., 2014).
Moreover, $NH_4$ increased again by the pumping activity and was transferred downstream (Fig.5 event 2). The eventual impact
of regulation of the Geuzenveld water system turns the pumping discharge into a point source for nutrients to downstream
water bodies as shown in Figure 8.

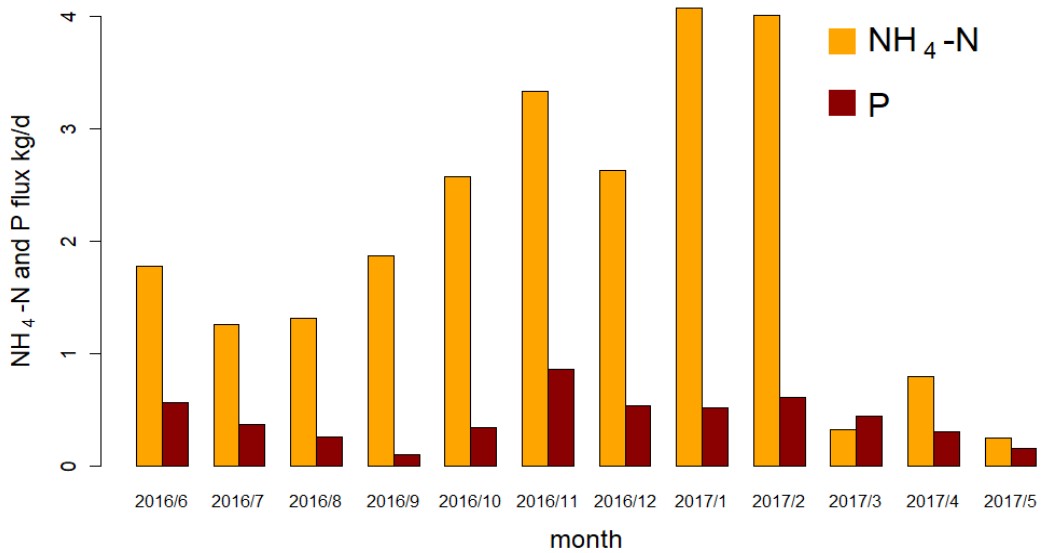

**Figure 8 Average daily $NH_4$-N and P flux (kg per day) in each month in the discharge (calculated from the continuous**
**measurements) of polder Geuzenveld from June 2016 to May 2017.**
Fluxes of N and P were highest during winter (Fig 6). These high fluxes are caused not only by the more frequent pumping
activity, but also by the higher concentration of N and P in the water column in winter. In the time series data, $NH_4$ (the major
form of N), had concentrations above 2.4 mg N $L^{-1}$ (the local environmental quality standard (EQS) for N-total), in all seasons
except spring. $NH_4$ concentrations even reached up to 6.5 mg $L^{-1}$. TP concentrations were constantly higher than 0.15 mg P $L^{-1}$
(the local EQS); during winter it was always over 1 mg P $L^{-1}$. Although the $NH_4$ flux in the discharge was very low in spring
(Fig.8), the actual total N flux might have been much higher, as organic N (phytoplankton) was the major form of TN instead
of $NH_4$ during this period (Fig.6 $NH_4$/N and organic-N/TN). Therefore, even though water authority measures have been
effective in controlling the water quantities in the polder, it had unanticipated impact on nutrients export to the downstream
water bodies. In order to prevent eutrophication in the urban waters, nutrient rich discharge from these areas is exported directly
to the North-Sea Canal and to the North Sea.
**4.6 Implications for urban water management in low lying catchments**
This study demonstrated high frequency monitoring technology to be an effective tool for understanding the complex water
quality dynamics. Investment in high frequency monitoring would greatly benefit the management of urban lowlands with
substantial groundwater seepage by elucidating the principle biogeochemical processes and nutrient temporal patterns for

realizing efficient mitigation and control of eutrophication. For example, redirecting the drain water effluent into constructed wetlands could be considered as a mitigation measure in low lying areas with artificial water systems that resemble the Amsterdam region, e.g. in cities such as New Orleans, Shanghai and Dhaka (Li et al., 2009; Nahar et al., 2014; Jones et al., 2016; Stahl, 2019). Centralizing the treatment of discharge water is also recommended, for instance by harvesting N as phytoplankton from the discharge during spring, or filtrating P at the pumping station during winter. Measures that artificially increase oxygen concentrations in the waters, such as the inlet of oxygen rich water, aeration by fountains or the artificial introduction of grazers or macrophytes may be considered to improve the ecological status of these urban waters. Moreover, aeration of the water in summer and autumn would possibly enhance processes such as coupled-nitrification-denitrification and anammox, eventually converting $NH_4$ to $N_2$, before the water is discharged to downstream waters. Importantly, before the application of any measures or maintenance in urban low-lying catchments, managers should evaluate the potential effects on the biological and chemical resilience, e.g. dredging of a layer with abundant benthic activity might destroy an important buffer to nutrients in growing seasons, especially P.

In this study, we focused on the analysis of the temporal patterns of water composition and on the deduction of the potential biogeochemical processes. Detailed studies about these processes and the biotic communities at the sediment-water interface were outside of the scope of this paper. A comprehensive study on the sediment-water interface would be necessary to further increase our knowledge on the role of the benthic zone in attenuating N and P seeping up from groundwater. Besides, further research would need to consider the optimal physical dimensions of water courses and drain configurations, as to benefit the ecological status of urban waters that are prone to nutrient-rich groundwater seepage.

**5. Conclusions**

This study aimed at improving our understanding of the mechanisms that control the temporal patterns of nutrients and other water quality parameters in an urban catchment. Time series of EC, $NH_4$, TP, and turbidity were obtained by applying a high frequency monitoring technology for one year (May 2016 to July 2016). Observed EC, $NH_4$ and TP could only partly be explained by conservative mixing of groundwater and precipitation components. In particular, N and P fluxes in the shallow ditches were also impacted by biogeochemical processes, such as primary production and iron redox transformations.

(1) $NH_4$, the dominant form of N in surface water, originates primarily from groundwater seepage, and concentrations are lowered by primary producers (phytoplankton and benthic algae) in the growing season. High algal biomass was also clear from high chlorophyll-a and suspended solids in the water column.

(2) TP showed high concentrations in winter, but relatively low concentrations in other seasons. Iron redox chemistry was the principle process controlling the P dynamics in shallow groundwater fed ditches. P dynamics may also have been partly influenced by primary production which consumes P for growth and at the same time produces $O_2$ influencing the redox status in the sediments and in the water column.

(3) High turbidity levels occurred in the late autumn and winter, mostly in the form of iron hydroxides. It resulted from a shift of the anoxic/oxic interface where the formation of iron hydroxides moves from the sediment towards the water column.

(4) Water pumped from the polder to downstream water bodies was rich in $NH_4$ from summer to winter, but rich in organic N in the form of algae during spring. P leaves the polder mainly during the winter season when it is released from the sediment and exported mostly in the form of P sorbed to $Fe(OH)_3$ colloids and as dissolved P.

(5) Precipitation diluted concentrations of most water quality parameters, but delivered $O_2$ to the water column, and in that way indirectly affected P and turbidity by intensifying iron oxidation and precipitation.

(6) Unlike many other natural and artificial catchments, rainfall and pumping events did not increase turbidity or TP concentrations at the short time scale, rather reduced turbidity and TP because of enhanced iron hydroxide precipitation due to oxygen inputs by runoff.

Our understanding of the N and P dynamics in this low-lying urban catchment may contribute to the development of effective water management strategies that reduce eutrophication conditions in both the urban polders and the downstream waters. Drainage of very low-lying areas (for use as residential and/or agricultural areas) not only increases pumping costs, but can also result in difficult to manage water quality conditions. Controlling the source, redirecting and utilizing the drainage water might be strategies to reduce the input of N and P from groundwater into surface water. In addition, we showed that in lowland urban areas with high seepage rates the reactivity of the stream bed sediments largely controls water quality of surface waters and thus should be managed with care when cleaning the surface water systems.

**Acknowledgements**

This work was funded through China scholarship council (no. 201309110088) and supported by Waternet, the Strategic Research Funding of TNO and Deltares. We highly appreciate the help and support from our Waternet co-workers: Eelco Wiebenga, Henk Molenaar, Sonja Viester, Laura Moria, and Frank Smits.

*Code/data availability*: The code scripts and datasets related to this paper are available on request to Liang Yu, contact is xiaobaidrawing@gmail.com.

*Author contribution*: Maarten Ouboter, Joachim Rozemeijer, and Hans Peter Broers funded this research. Hans Peter Broers and Joachim Rozemeijer designed the field work. Liang Yu carried out the field work and the data collection, analysis, visualization, discussion, and the writing of the manuscript, under the supervision of Hans Peter Broers and Joachim Rozemeijer before 2019, Ype van der Velde as the main supervisor since 2019. All the authors participated the discussion of the data analysis results, and helped prepare the manuscript.

*Competing interests*: The authors declare that there is no conflict of interest.

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
