# Peer review of "Drivers of nitrogen and phosphorus dynamics in a groundwater-fed 1"

_Hydrology and Earth System Sciences, 2020_

## Referee Comment (RC1) · Matthias Gassmann (Referee) · 16 Apr 2020

**Review of hess-2020_34**

**'Drivers of nitrogen and phosphorus dynamics in a groundwater-fed urban catchment revealed by high frequency monitoring'**

**by Yu et al. (2020)**

General comments

The topic of the paper in combination with the use of high-frequency nutrient concentration data in a complex hydrological system such as an urban low land polder is an interesting topic well in the scope of HESS. The scientific content could be an interesting addition to the current state of knowledge and is thus worth to be published. Nevertheless, the paper shows some formal and methodological weaknesses, which need to be improved or clarified before publication:

1. The language of the paper is often unprecise (expressions such as 'much higher') and there are many grammatical mistakes (see specific comments). The same expressions are often used shortly after each other, leading to repetitions of the same content (e.g. L 20, 22 and 24: mixing of groundwater and runoff governed water quality). This unnecessarily prolongs the text. A higher scientific precision, a better grammar and a consolidation of the text would add much to its readability and the scientific language.

2. Total phosphorus (TP) and ammonium (NH4) is analysed, but unlike nitrate (NO3) ammonium is not directly a driving factor for eutrophication. NO3 on the other hand is not included in the discussion, mainly because the planed NO3 measurements didn´t work in a proper way. This raises the question why NO3 wasn´t at least monitored by regular grab samples (low frequency). This would have also helped to confirm the conceptual model for nutrition dynamics in low land polders.

3. Though high frequency data were collected, they haven´t been really used in the analysis to precisely describe single events, except for the analysis of single pumping events. There would probably have been the same results when the data resolution would have been one day. Mostly concentrations are discussed in an annual or seasonal context. Please elaborate more on the added value of a 20 min sampling compared to a daily sampling in the discussion. Probably, the focus of the paper should not be too much on high-resolution sampling (e.g. in the title).

4. A statistical analysis of the data is completely missing. All processes seem to be deduced by just visually comparing the graphs, without calculating e.g. correlation coefficients.

5. The authors developed a mixing model to determine which amount of nutrients can be attributed to hydrological mixing and which to biological processes. Latter one is derived by the discrepancy between model results and measured concentration. Electrical conductivity (EC) acts as a conservative tracer in this case. However, the model failed to reproduce EC within the polder after November 2016. This raises the question whether the amount of biological processes for nitrogen and phosphorus can be determined on this basis. This uncertainty should be discussed in more detail.

5. The authors are classifying their study in the context of eutrophication: the findings of this study are meant to help water managers to mitigate eutrophication. However, there are no suggestions how the results of the study can be used to do so.

6. An aim of this study was to analyse and compare annual scale, precipitation events and pumping events (lines 88-90), but this scheme can´t be found in the discussion. The three scales are mixed up rather than to distinguish between the dynamics of the different time scales.

Specific comments

L22: *'through variation of the intensity and duration of the events'* I don't understand the meaning of this sentence.

L23: Is NH4 really the dominant form in surface waters? I know many examples from Europe where nitrate dominates. Furthermore, NH4 gets nitrified to NO3, leading to decrease in NH4 and increase in NO3 concentrations. Maybe the authors mean that NH4 is the dominant form in urban water bodies?

L 24, 25: *'low concentrations during the algae growing season, while concentrations were governed by mixing of groundwater and precipitation inputs in the late autumn and winter.'* This sentences only makes sense, when the authors mention the concentrations in autumn and winter as well.

L26 – 28: The two sentences have nearly the same content: release of reduced iron causes turbidity.

L29, 30: Was organic N measured? A denitrification needs anaerobic conditions, while in spring O2 concentrations were rather high, how does that fit together?

L 41: I would replace *'end up'* by *'reaching'* or something similar

L 45: This sentence belongs to the following paragraph.

L 47: it should be *'in the aquatic environment'* or *'in aquatic environments'*

L 48, 49: molecular nitrogen and phosphate was not mentioned until now, the authors should introduce N2 and PO4 first, like they have done it for nitrate and ammonium

L 49: *'NH4 is the preferred N-form by microbes'.* There are also other microbes which prefer different forms of Nitrogen (like the authors mentioned in the sentence before).

L 50, 51: The content of this sentence is obvious, when there is no NH4 and NO3 the uptake of the substances can´t reach a maximum.

L 53: *'Under aerobic conditions, $NH_4$ can be oxidized to $NO_3$ through nitrification by nitrifying microbes even under cold conditions (below 10 $^{\circ}C$), which is an $O_2$ consuming, acid generating process'* Please revise the sentence structure. It sounds as if the nitrification under cold conditions is O2 consuming.

L 59: *'during events'* what kind of events? The authors should be more precise with their expressions *'during hydrological/precipitation events'*

L60: *'and chemical reactions….'* This part of the sentence doesn't fit substantively to the ones before, which were about transport processes not transformation processes. The latter aspect is discussed in the following paragraph.

L 74: *'N and P dynamics, for instance its response…'* Please replace 'its' by 'the' or 'their'

L 83. Please replace '…insight in…' by '…insight into…'

L 86 – 88: Please replace '*We conducted a one-year high frequency monitoring campaign in 2016-2017, measured parameters EC, NH4, TP, turbidity and water temperature. '* by '*We conducted a one-year high frequency monitoring campaign in 2016-2017. Measured parameters were EC, NH4, TP, turbidity and water temperature.'*

L 97, 98: Do the authors mean that groundwater seeps into the catchment because the water level of the groundwater is higher than the sole of the channels and the drain system?

L 99: *'much higher'* How much is much higher?

L 102: NAP doesn´t need to be explained, naming the abbreviation should be enough.

L 117: The temporal resolution (20 min?) is missing in the description. I could only find it in the abstract.

L135: *'was calibrated'* instead of *'was calibrating'*

L153: What was monitored by Waternet? – *'Waternet has monitored the water quality'?*

L154: *'the frequency became twice…'* Frequency cannot increase by itself: *'frequency was increased….."*

L154: *'were measured in this dataset'* In a data set nothing can me measured. Please be more precise.

L 172: Potential evapotranspiration is a virtual measure derived from meteorological data. It doesn't give an actually evaporated water volume. How is the use of potential evapotranspiration justified? Actual evapotranspiration should rather be used in this case.

L 172: I suppose groundwater seepage S stems from outside of the polder. How are the values of this variable derived? Calibration? And why is it multiplied by the area of the polder? Please clarify these issues.

L 174: Naming the variables in the order of their occurrence in the formulas would be easier to follow.

L 182: *'d(VC)"* is not explained in the text, I guess it is the concentration of the ditch water?

L 185. Is the high salt concentration really the reason for EC being a conservative tracer? Or do you mean that the concentration difference between the two water sources renders EC a useful tracer?

L 193*: 'simulated concentrations … together with their high frequency…'* exchange *'their'* with *'the'*

L201: Rain events are very long (> 1 month), that seems to be more representative for a (sub-)season or similar. A rain event usually is shorter than a few days in central Europe.

L 207, 208: Four times *'and'* in one sentence. Please rephrase.

L 217: Why did the wet season start in October and end in February? Did the authors maybe calculate a cumulative water deficit? According to Figure 2 there has been quite a lot of rain up to the middle of March. Further, a dry and wet season usually refers to a semi-arid climate, which Amsterdam is far from being in. Please think about re-naming the compared time spans and give details on how they were separated from each other.

L222: Please change *'..period that the water temperature…'* to '…during which the water temperature….'

L228 – 229: *'In contrast to the constant water level ranges from surface water regulation regime'* I am not sure about what the authors want to say. Please clarify.

L237: Remove *'…if there was no rain'.*

L 237: *'this duration of the return'* Bad expression

L 241- 244: The authors mention twice, that NH4 deviated from slope of EC.

L 250: The authors refer to excessive precipitation, but unfortunately this is not shown in Figure 2.

L247 – 252: The description of turbidity is rather confusing: 'Turbidity was constantly below 100 FNU' is followed by a peak description of 500 FNU. The authors also miss out, that there are several EC peaks during October. They also repeat the same content *('turbidity stayed around 200 FNU'*) in lines 249 and 251.

L 259: Delete *'when'*

L265 – 274: The authors are writing that concentrations are captured well, but there are discrepancies of more than 50 %. Maybe the authors want to point out, that the dynamics are captured?

L265 – 274. Since water levels were measured: why were the model results of the water levels (L(t)) not compared to measurements?

L 277, 278: What is the criterion for a 'significant dilution event'?

L 287, 288: Why does event 2 follow EC but not event 3? According to figure 3 NH4 concentration seems to increase parallel to EC. The authors attribute the missing dilution only partly to the data gap. I don´t think a statement about the missing part can be made, when there are no data available. Further, the word *'partly'* implies that there was no strong dilution.

L 289: Can you be sure about the dilution? When you compare the high frequency measurements with grab samples (figure 2) 0.35 mg TP/l seems to lie within the uncertainty range of the high frequency measurements. Please discuss a potential sampling uncertainty.

L290: The authors forgot to mention, that TP was also falling again after reaching 0.8 mg/l

L292: There were more small rainfall events during recovery period of event 3 compared to event 4

L310: *'pumping has the least influence on NH4 in winter'* It is difficult for the reader to relate this to figure 5, because the scaling for NH4 concentrations is different for every event

L316: The authors suggest that turbidity is influenced by pre-event conditions, but the reader has no specific information about the pre-event conditions. This point is also not further discussed in the Discussion, where this sentence should be placed anyway.

L 329: *'Runoff in Greuzenveld has waters with EC….'* – Please rephrase.

L332: *'…the mixing model…which revealed close similarity to the measurement'*. This statement is wrong. There is a big discrepancy between model and measurements in the second part.

L341 – 347: Only the discrepancies during winter are discussed, but measurements and the model reach into middle of June.

L 397: NH4 can be consumed by nitrifying bacteria (not by nitrification).

L398, 399: Denitrification and anammox are two different processes and the chemical equation doesn´t fit to neither of them. For anammox NO2 is needed, not NO3. I am also wondering why nitrification and denitrification are not discussed apart from this two sentences. Nitrification is also an oxygen consuming and NH4 reducing process. While denitrification can take place under anoxic conditions.

L 417 Turbidity only increased for a short period (end of October to middle of November)

L418: 'Iron-rich particles are the most likely source of turbidity in freshwater': Concentration of iron particles is high until February. If this is true, why is turbidity low in February? Please clarify.

L 433: *'…turbidity became high'* according to figure 2 turbidity wasn´t high in this time span.

L447: *'relatively low in oxygen (because of warming)'* Additionally, a reason for reduced oxygen might be an increase in O2 consumption by microorganisms.

L 523 – 664: The references are not completely in an alphabetical order and slightly different citation styles were used (e.g. sometimes DOI is written in capital letters, sometimes not)

Figure 1: Readers have to guess the channel after the pumping station is Boezem Haalemmerweg and whether the left drainage system is the secondary water channel which Greuzenveld is connected to. The map above the Google Maps Card with the location is too small and doesn´t help to understand the system. Please provide a better overview map of the study area.

Figure 2: The discrete sampling data points are hard to identify; adding precipitation would help for interpretation of data.

Figure 3: 'measured' and 'modelled' timeseries overlap in the same colour. It is not visible whether 'TP modelled' shows the same peaks like 'TP measured'. Choose different colours.

Figure 4/5: A rearrangement of graphs and scales could add to a better understanding of the figures.

Figure 5: The reader can´t distinguish between day and night time, though the authors discuss this in chapter 3.3.2 based on this figure; while the first block only contains water temperature, the second block contains three measured parameters. There is room for improvement of visibility.

---

## Referee Comment (RC2) · Piet Seuntjens (Referee) · 18 Apr 2020

General

This manuscript describes a high frequent monitoring study of water quality in a groundwater-fed urban ditch. The monitoring allows to elucidate governing processes of water quality and the authors do a good job in trying to explain the observed water quality parameters. The observations are quite specific for the study area at hand given the specific pumping management, and hence extrapolation to other catchments would be less obvious, which limits the generalisation of the results. That is the major draw-back. Sometimes the authors also draw far reaching conclusions which need further

confirmation. Given the fact that the observations are sound and well described, and discussion needs some further confirmation, I rate this publication to be acceptable for publication with minor revisions. The revisions should help to improve the readability and conclusions that can be drawn from this case study.

Specific comments

Figure 1 contains too many features. I cannot read the map of Amsterdam, and the black item above. It seems redundant. I like the figure with the cross sectional view, but some features are unclear and should be redrawn: (1) where does the water from the drain system go to ? does the drain system capture groundwater or seepage water ? What does the green area in the figure represent ? Is seepage vertically oriented towards the bottom of the ditch ? I would expect the ditch captures water from the surroundings.

Figure 4 is too small. What does 1, 2, 3, 4 represent ?

Figure 6 could rainfall and/or EC be added here ?

P7L245 a discrete water sample confirmed the low NH4: only one sample. This seems poor to serve as a confirmation. Did you perform regular grab sampling as to check the online values ? How is the data quality of the online measurements validated ?

P7L270 predicted and observed NH4 concentration generally agree. I would rather say that for both NH4 and P the concentrations are overestimated by the model. This makes sense since you don't take transformation or sink processes into account in the model.

P9L344 the residence time is mentioned here. It would make sense to have the numbers for the residence time of the water in the manuscript. Did you calculate them for the different time periods ?

P10L400 did you measure NO3 in this study? do you have clear evidence for NO3 consuming processes ?

[Figure]

P13L516-517 you state that the reactivity of the streambed sediments largely controls the water quality. Can you be more concrete on : (1) how then exactly the management should take care of this and (2) what type of measurements need to be done in the sediment to better understand the mechanisms in this system and to prove the hypothesis you make in Figure 7 ? You infer the mechanisms in Figure 7 based on surface water (water column) data only, and conclusions may need further elucidation.

————————————————

---

## Referee Comment (RC3) · Anonymous Referee #3 · 22 Apr 2020

Review of hess-2020_34 'Drivers of nitrogen and phosphorus dynamics in a groundwater-fed urban catchment revealed by high frequency monitoring' by Yu et al. (2020)

Context – Goals: This paper presents a 12-month investigation of variations in nutrients and various water quality indicators at the Geuzenveld polder in Amsterdam through, mainly, in situ monitorings of the waters recovered close to a pumping station (allowing to regulate the water levels in the polder). The authors investigated variations among the datasets over 12 months, and tested the incidence of rain events, and pumping events . They've applied a mixing model to explain some of the observed variations.

Several parts of this paper are highly speculative. Correlations between data, and several other issues, would need to be supported by statistical tests. There are no M&M sections on the statistical tests performed; except a presentation of the mixing model. The raw datasets and scripts should be presented in the suppl. materials. Conclusions are not always supported by the presented datasets, and are sometimes highly speculative. They need to be supported by other studies which are not always cited and explained. This paper will require major improvements to meet the scientific quality of HESS papers. Several parts of this paper will need to be re-written and re-considered after a presentation of the statistical tests.

Major comments

1. L86, Please specify the possible "management strategies" and add references

2. Fig. 1; picture is too dark and its resolution is too low; please position the temporary floating platform used for the monitorings on this fig. The Drain 3 sampling point should be indicated; Yu et al., 2019 should be cited in the legend

3. High and low frequency monitorings: How did you compute the confidence intervals / error bars on the monitored values? please clarify these issues.

4. low frequency monitorings - L153 - "monitored at the pumping station" ; please clarify; where were these samples or values collected?

5. There is no section in the materials and methods regarding the statistical analyses of the datasets? please describe the statistical tests that were performed; which statistical packages were used? What was your experimental design regarding these tests?

6. Fig. 2: please indicate in the legend that nutrients and other quality indicators were monitored at the pumping station; variations in the presented datasets should be supported by statistical tests; correlation tests between monitored values should be performed. All raw datasets should be presented in the suppl. materials

7. L219: "The wet season is distinguished by a higher frequency of pumping and lower

water temperatures" ; please do statistical tests to validate these conclusions; when you indicate frequency, do you mean "volume"? please clarify

8. L221-222: "Especially in January and February 2017, there was a considerable period...": please define a "considerable period" by using statistical tests; was that specific of that year?

9. After the pumps stopped, the surface water level recovered faster during the wet season (between October 2016 and 225 March 2017) than during the dry season? Comment: where is this shown? Please clarify

10. L226-227; L229; section 3.1.2: conclusions should be supported by statistical tests and should consider "confidence intervals" of the monitoring tools

11. Please show the position of the piezometer on Fig. 1; its GPS coordinates should be indicated in the materials&methods section

12. L230; please cite a document for the "water level regulation of the boezem Haarlemsmeer"

13. L254: "significantly increased" : which test was performed?

14. L259-263: correlation tests should be done between total Fe values and turbidity

15. L277: there are no red blocks in Fig.1; neither in Fig. 2???

16. Section 3.3.1: datasets should be supported by statistical tests and should consider the confidence intervals of the monitored values

17. L301-304: to be moved in the discussion

18. Section 3.3.2: add statistical tests e. g. L309-310, correlation test between values, validate the seasonal effect, etc

19. L310: "during events 2, 3 and 4, TP and EC are positively correlated"; which test? Please give the p-values, etc

20. Figs. 4; please clarify the legend and relation with Fig. 2. Add statistical tests to define which values are correlated, etc.

21. Please clarify what is correlated or not according to pumping

22. Line 332 - "This presumption is supported by the mixing model result of EC, which revealed close similarity to the measurements".: comment - How did you test this similarity between monitored and modeled values? please give the details of this performance analysis in the result section

23. Too many citations of the figs in the discussion; several issues should be transferred in the result section

24. L333: not clear

25. No cited literature in section 4.1 of the discussion ?

26. Fig. 6 should be presented in the results; the raw datasets should be presented in the suppl. materials. Any information on the confidence intervals for these datasets?

27. L362-364; 380-382, etc: please add statistical tests to support these conclusions

28. L365; 398: qPCR datasets to estimate the population levels of some of the organisms involved in NH4 assimilation, etc, would be interesting to support some of the conclusions reached in this paper

29. L401; Fig. 6 is indicated in relation with NO3 datasets; where are these data?

30. L417 "...significantly..": please support this by a statistical test

31. Scripts and data should be made available in the suppl. materials

32. Section 4.4: speculative section on the importance of biotic processes (benthic algae, bacteria / nitrification-denitrification-anammox) in the variations observed in Fig. 2 and Fig. 6-> there are very few datasets on these issues (only chlorophyll a monitorings); it would have been relevant to add qPCR (variations in total bacterial numbers,

cyanobacteria, denitrifiers, etc) assays to validate the inferences made in Fig. 7. This section need to be supported by more citations of the literature on similar issues.

33. L444-445: "... Phytoplankton biomass decreased because of competition for N or grazing activity..."; this is speculative – there are no data on grazing? At least add a reference on this issue to support this possibility

34. 445-446: not supported by the presented data

35. Fig. 7; please cite in the legend all papers which made possible most of these inferences, and indicate which data presented in this paper added support for the presented scenarios

36. L447- ".. relatively low in oxygen (because of warming)"; please add data which support this effect of warming on oxygen levels (or a citation)

37. L448 – "... Biological activity declined (colder and less light)...": there are no data on these issues; following sentence is also an inference from the literature and not from the presented data

38. L463 – "this accelerates the further aggregation of the iron complexes..." ; this is speculative / not based on the presented data; add arguments (a citation) to support this conclusion or delete

39. L464 – "...The resulting larger particles more readily settle to the bottom.."; comment : no data on this issue; add a citation to support this conclusion or delete

40. L473 +– "... the water was highly turbid because of the formation of iron hydroxide colloids in the water column.."; "..The activation of the pumps caused export of these colloids and particles and thus reduced turbidity ..."– comment : no data on this issue / have you monitored "particle sizes"? please add these datasets; or add a citation to support this conclusion or delete

41. Fig. 8 should be presented in the M&M and result sections. Please show on Fig. 1

where these monitorings were performed

42. Only fig. 7 should be cited in the discussion; other figs should not; all data presentation issues should be moved from the discussion into the result section

43. Discussion is too long; please simplify but avoid over-interpretation of the datasets (conclusions should be strictly based on statistically well supported trends)

44. L499 "..Iron redox chemistry was the dominant process controlling the P dynamics in shallow groundwater fed ditches"; comment: dominant over which other processes??? Please clarify and give arguments / which data demonstrate clearly this relation? Datasets present total Fe values and total-P; have you done correlation tests?

45. L503 – L508 "..mostly in the form of iron hydroxides": comment – did not see any datasets on this issue? Please limit your conclusions to the points that were investigated in the paper

46. L510 – "…by intensifying iron oxidation and precipitation…" : comment – did not see any datasets on this issue? Please limit your conclusions to the points that were investigated in the paper

Minor comments

1. In the introduction, L84: ".. to understand the mechanisms that control the dynamics of N and P in urban delta catchments.."; please clarify by changing "mechanisms" by " the hydrobiogeochemical processes that control…"

2. L115: "During rainfall events, the surface water level will rise faster"; please be more accurate or add a citation on these issues.

3. L17, please put the month before the year

4. Doi numbers have not been indicated for the cited papers

5. L43; why "pivotal"? explain

6. L49; please use another term than "preferred" ; most effectively uses NH4 for protein synthesis

7. Several sentences are too long; please simply at least the following sentences e. g. L36-39; L62-65; 160-163; 356-359

8. L74: "In recently years Âż ; to be changed

9. L75, Please define "high frequency technology"

10. L89-90: last part of this sentence is not needed i. e. "... unraveling the hydrological and the reactive biogeochemical processes that control the nutrient 89 dynamics at these 3 time scales"

11. L118; replace pump by "pumping"

12. L135: "calibrated" instead of calibrating

13. L146: "times"; "lightening -> lightning,

14. L153-154; 164: not clear; to be re-worded

15. L161: shook -> mixed

16. L169: inlets -> inputs

17. L170: outlets -> outputs

18. Fig S1; change valid for validated; please indicate color code in the legend; please perform and indicate the p value for the correlation test. Description of Fig. S1 in this suppl. material should be deleted.

19. L189: please delete "sourced from groundwater"

20. L191-192; to be moved in the results or discussion section

21. L207-212: to be deleted

22. Please avoid citing figs in the discussion

23. Fig. 5; translate the "x" axis

Please also note the supplement to this comment:
https://www.hydrol-earth-syst-sci-discuss.net/hess-2020-34/hess-2020-34-RC3-supplement.pdf

---

## Author Comment (AC1) · 1 Jun 2020

The comment was uploaded in the form of a supplement:
https://www.hydrol-earth-syst-sci-discuss.net/hess-2020-34/hess-2020-34-AC1-
supplement.zip
34, 2020.

---

## Author Comment (AC2) · 1 Jun 2020

The comment was uploaded in the form of a supplement:
https://www.hydrol-earth-syst-sci-discuss.net/hess-2020-34/hess-2020-34-AC2-supplement.zip

---

## Author Comment (AC3) · 1 Jun 2020

The comment was uploaded in the form of a supplement:
https://www.hydrol-earth-syst-sci-discuss.net/hess-2020-34/hess-2020-34-AC3-supplement.zip

---

## Author Comment (AC4) · 3 Jun 2020

The comment was uploaded in the form of a supplement:
https://www.hydrol-earth-syst-sci-discuss.net/hess-2020-34/hess-2020-34-AC4-supplement.zip

---

## Author Comment (AC5) · 3 Jun 2020

The comment was uploaded in the form of a supplement:
https://www.hydrol-earth-syst-sci-discuss.net/hess-2020-34/hess-2020-34-AC5-supplement.zip

---

## Author Comment (AC6) · 3 Jun 2020

The comment was uploaded in the form of a supplement:
https://www.hydrol-earth-syst-sci-discuss.net/hess-2020-34/hess-2020-34-AC6-
supplement.zip
34, 2020.

---

## Author Response (AR1)

**1. Reply to all the reviewers and the Editor**

**Reply to the Editor**

Dear Editor,

Thank you very much for your kind words, your time, and your useful and constructive feedback. The comments from you and the reviewers have been incorporated and this has further improved our manuscript. Below we provide our responses to your comments one by one. We believe that this revision clearer communicates our findings and articulates stronger our conclusion to the audience.

We appreciate the contribution from you and all the reviewers.

Sincerely,

Liang Yu

**Editor decision**

**HESSD-Manuscript "Drivers of nitrogen and phosphorus dynamics in a groundwater-fed urban catchment revealed by high frequency monitoring" (HESS 2020-34).**

**Dear Dr. Liang Yu**

**Thanks for submitting the responses to the three reviews and the extensive material provided. You have properly addressed most of the comments. Nevertheless, there are still a few issues where further revision are needed. I listed them below.**

**1. Reviewer 1, comment 41: Definition of wet and dry periods: You have only partially addressed this comment. It is now clear how you have defined the wet and dry periods in an operational manner. However, as pointed out by Reviewer 1, this operational definition puts the end of the wet season at about mid March 2017 and not simply at the of February. Actually, inspecting the EC data in Fig. 2 nicely illustrates that the operational definitions captures the change points in the temporal evolution.**

**However, the current version mixes an operational, data-driven definition of the period with a calendar-based one. Please rectify this issue such that the periods are delimited in a consistent manner.**

We agree that the period from October to the middle of March captures the temporal evolution of the parameters. Moreover, the water deficit becomes more significant from around March 10, and the pumping frequency is high till the March 15.

Thus we modified the text in response to Reviewer 1:

*"We define wet and dry seasons based on water surplus and deficit. The average net rainfall (the water surplus/deficit in Figure 2) is 1.4 mm/d for the period of 01-10-2016~15-03-2017, and -0.8 mm/d for the rest.*

*Subsequently, we statistically analysed the difference between these two periods for multiple parameters. Table S2 shows the mean of each parameter for the wet and dry seasons. The wet and dry seasons means are significant different for all parameters, but the EC."*

*Table S2 The mean of each parameter, and the significance for the wet and dry seasons*

| | Net rainfall* mm/d | Pump volume* m³/d | Water temperature* °C | EC µs/cm | NH₄* mg N/L | TP * mg P/L | Turbidity* FNU | Fe* mg/L | O₂* mg/L |
|---|---|---|---|---|---|---|---|---|---|
| Wet | 1.4 | 1050 | 6.7 | 1212 | 3.7 | 0.8 | 197 | 3.4 | 4.3 |
| dry | -0.8 | 712 | 17 | 1252 | 3.0 | 0.5 | 15 | 1.5 | 3.3 |

\* $p < 0.05$

**2. Reviewer 2, comment 7: There are several linguistic shortcomings in the suggested text. Please carefully check the language (spell check, grammar).**

After carefully checking the text, we here present an improved version of the added section 4.6:

"**4.6 *Implications for urban water management in low lying catchments***

*This study demonstrated high frequency monitoring technology to be an effective tool for understanding the complex water quality dynamics. Investment in high frequency monitoring would greatly benefit the management of urban lowlands with substantial groundwater seepage by elucidating the principle biogeochemical processes and nutrient temporal patterns for realizing efficient mitigation and control of eutrophication. For example, redirecting the drain water effluent into constructed wetlands could be considered as a mitigation measure in low lying areas with artificial water systems that resemble the Amsterdam region, e.g. in cities such as New Orleans, Shanghai and Dhaka. Centralizing the treatment of discharge water is also recommended, for instance by harvesting N as phytoplankton from the discharge during spring, or filtrating P at the pumping station during winter. Measures that artificially increase oxygen concentrations in the waters, such as the inlet of oxygen rich water, aeration by fountains or the artificial introduction of grazers or macrophytes may be considered to improve the ecological status of these urban waters. Moreover, aeration of the water in summer and autumn would possibly enhance processes such as coupled-nitrification-denitrification and anammox, eventually converting $NH_4$ to $N_2$, before the water is discharged to downstream waters. Importantly, before the application of any measures or maintenance in urban low-lying catchments, managers should evaluate the potential effects on the biological and chemical resilience, e.g. dredging of a layer with abundant benthic activity might destroy an important buffer to nutrients in growing seasons, especially P.*

*In this study, we focused on the analysis of the temporal patterns of water composition and on the deduction of the potential biogeochemical processes. Detailed studies about these processes and the biotic communities at the sediment-water interface were outside of the scope of this paper. A comprehensive study on the sediment-water interface would be necessary to further increase our knowledge on the role of the benthic zone in attenuating N and P seeping up from groundwater. Besides, further research would need to consider the optimal physical dimensions of water courses and drain configurations, as to benefit the ecological status of urban waters that are prone to nutrient-rich groundwater seepage.*"

**3. Reviewer 3, general comment: Reviewer 3 was critical about the lack of statistical analyses. The same opinion was expressed by Reviewer 1 although he did not insist that much on that aspect. In your response, one reads 'We are not sure that statistical testing is the best approach in dealing with a complex dataset with high-frequency data.'. Can you elaborate what you consider a more promising approach, how it is implemented in the manuscript and where you explain this to the reader?**

Following the comments from Review 3 and the Editor, we have performed additional statistical analyses to support our qualitative conclusions, including statistical tests for significant differences among wet and dry seasons (see above), a correlation analysis of all the parameters at the annual scale, comparison between the high frequency and discrete monitoring data, and correlation analysis between mixing model and solute concentrations, correlations between solutes at the rain and pumping event scales. The test results are shown in the supplementary information in the form of a series of coefficients of determination tables. Inclusion of these statistical tests has improved the conclusiveness of our statements and arguments.

**4. Reviewer 3, major comment 3: Is the information provided in the response included in the main text and the figures shown in the SI? If not, please do so.**

Thank you. We now added the following text to section 3.1.2 in the manuscript and the figure in the supplementary information.

"*The correlations coefficients ($R^2$ "Pearson" method used) between the high frequency data and the routine discrete sampling data from the water authority are 0.88 for EC (p-value < 0.05), 0.92 for $NH_4$ (p-value < 0.05), and 0.97 for TP (p-value < 0.05). The scatter plots between the high and low frequency measurements are shown in Figure S6.*"

[Figure]

*Figure S6 Scatter plot of the high frequency measurement vs. the discrete sampling results. The dash line is the 1:1 line.*

**5. Reviewer 3, major comment 7: Please provide more information about your statistical analysis. Was there a significant difference between the two periods? Do the data fall into two clusters? Otherwise the split would be arbitrary. Show data (SI).**

Please refer to our detailed response to comment 1. The two periods were defined based on water excess and differences were tested for significance (see table above).

**6. Reviewer 3, major comment 13: Unless there is a scientific reason not to perform a statistical test, please provide this information. Even if you think that the differences were evident enough, this is no argument not to support this view by a statistical test.**

**Figure S4 is poorly explained. What is exactly depicted? What is an '... increase in variations over the day.' ? One cannot see any sub-daily fluctuations. What are 'difference between daily values' ? One can only guess. Please be more precise and specific.**

We accept the editor and the reviewer's suggestion to increase the strength of the evidence by using a statistical tests instead of the authors' educated opinions. Below we present a statistical test to prove the significant increase of TP from the mid of November till end of February compared to the TP value in the rest of the year.

Null hypothesis: there is no significant difference between the two groups of TP values (group 1: before 15-11-2016 and after 01-03-2017, group 2: the TP value between these two dates).
The null hypothesis is rejected when p < alpha (significance level, take 0.05). *p* is the smaller, the better.
As our data sample is non-normally distributed, we chose the Wilcoxon rank-sum test. The tests were performed in Rstudio (version 3.6.1), wilcox.text() in package "stats". The script and the test result are as below:

Script:
wilcox.test(group1, group2, exact = FALSE, alternative = "greater", conf.int = TRUE, paired = FALSE)
    # noted that, in the script, we chose a one-sided test that set the alternative hypothesis as "the values from group1 are significantly greater than the values from group 2". And, as the sample size is larger than 50, normal approximation was used instead of the Hodges-lehmann estimator to estimate the magnitude of difference between the two data groups.
Test result:
    *p*-value < 2.2e-16

magnitude of difference between the two groups is 0.41 mg P/L

Decision:

*p* value is smaller than alpha, the null hypothesis is rejected. The difference of the two medians from group 1 and group 2 is 0.41 mg P/L. Thus the TP concentration significantly increased starting from the middle of November till the end of February.

And we changed the added text in line 254 into:

*"TP concentrations were significantly higher during the period between 15-11-2016 and 01-03-2017 than the rest of the time (p-value < 0.001 , Figure S4), during which TP fluctuated around 0.5 mg $L^{-1}$, but always below 1 mg $L^{-1}$."*

We replaced Figure S4 with the figure below:

[Figure]

**Figure S4 Statistics of TP concentrations in two periods(group 1: before 15-11-2016 and after 01-03-2017, group 2: the TP value between 15-11-2016 and 01-03-2017). The boxplots include the medians, 20% and 75% percentiles and outliers.**

**7. Reviewer 3, major comment 30: Unless there is a scientific reason not to perform a statistical test, please provide this information. Even if you think that the differences were evident enough, this is no argument not to support this view by a statistical test.**

Similar significance test as above was applied to this comment.

**Turbidity**

Null hypothesis: there is no significant difference between the two groups of turbidity values (group 1: before 01-10-2016 and after 15-11-2016, group 2: the turbidity values between these two dates).

Test result:

p-value < 2.2e-16

magnitude of difference between the two groups is 209 FNU

Decision: *p* value is smaller than alpha, the null hypothesis is rejected. The difference of the two medians from group 1 and group 2 is 209 FNU. Thus the turbidity level significantly increased from October to the middle of November.

**Fe**

Null hypothesis: there is no significant difference between the two groups of Fe values (group 1: before 01-10-2016 and after 01-03-2017, group 2: the Fe values between these two dates).

As our data sample sizes are small (< 20), we chose the Wilcoxon rank-sum test.

**Note that, as the sample size is less than 50, the exact *p* value is able to be calculated, and Hodges-lehmann estimator to estimate the magnitude of difference between the two data groups can be applied.**

Test result:

p-value = 0.02

magnitude of difference between the two groups is 1.1 mg/L

Decision: *p* value is 0.02 < alpha, the null hypothesis is rejected. The difference of the two medians from group 1 and group 2 is 1.1 mg/L. Thus the Fe concentration significantly increased from October to the end of February.

Line 417 is changed into:

*"From the late autumn onwards, turbidity and total Fe concentrations substantially increased compared to the rest of the time (Fig.2, p value < 0.001 for turbidity and = 0.02 for Fe). Turbidity peaked first at 1800 FNU and stayed at a plateau of ~200 NFU during the rest of the cold and wet season. Total Fe in the water column reached to 6 mg/L from below 1 mg/L."*

**Please modify the manuscript as suggested and address the issues listed above.**

We have updated the manuscript according to the responses to both the reviewers and the editor.

**Reply to reviewer 1**

**Review of hess-2020_34**
**'Drivers of nitrogen and phosphorus dynamics in a groundwater-fed urban catchment**
**revealed by high frequency monitoring' by Yu et al. (2020)**

**General comments**

**The topic of the paper in combination with the use of high-frequency nutrient concentration data in a complex hydrological system such as an urban low land polder is an interesting topic well in the scope of HESS. The scientific content could be an interesting addition to the current state of knowledge and is thus worth to be published. Nevertheless, the paper shows some formal and methodological weaknesses, which need to be improved or clarified before publication:**

We thank Professor Matthias Gassmann (Reviewer 1) for his compliments and for his time and valuable review which led to a clearly improved paper.

**1. The language of the paper is often unprecise (expressions such as 'much higher') and there are many grammatical mistakes (see specific comments). The same expressions are often used shortly after each other, leading to repetitions of the same content (e.g. L 20, 22 and 24: mixing of groundwater and runoff governed water quality). This unnecessarily prolongs the text. A higher scientific precision, a better grammar and a consolidation of the text would add much to its readability and the scientific language.**

We agree with the reviewer that the language can be improved. We have worked through the text and updated the language, consolidated the text and eliminated repetitions. Thanks to the reviewer and the many suggested improvements, we think the readability has much improved.

L 20, 22 and 24 are changed accordingly in comment 8.

**2. Total phosphorus (TP) and ammonium ($NH_4$) is analysed, but unlike nitrate ($NO_3$) ammonium is not directly a driving factor for eutrophication.**

There is strong evidence for the role of ammonium in the process of eutrophication and the preference by several forms of phytoplankton over $NO_3$ (Blomqvist et al., 1994[1]; Glibert et al., 2016[2]; Gobler et al., 2016[3]; Andersen et al., 2019[4]). The studies cover both fresh and saline water environment. In our system $NH_4$ is the main form for N (new Figure 6 Response, see below) and Nitrate is only present in very low concentrations. We previously published on our system in Yu et al. (2019)[5] but realize now that it is important information for this paper too. So, we added the graphs of TN and $NO_3$ to the manuscript (see new Figure 6 below).

We made some changes to the text to better refer to the previous study, setting the scene for a reader who is not familiar with the previous paper. For example, we changed the text in the abstract to better explain the situation, and we added to section 2.1:
* * *
[1]Blomqvist P., Pettersson A., and Hyenstrand P.. Ammonium-nitrogen: a key regulatory factor causing dominance of non-nitrogen-fixing cyanobacteria in aquatic systems. ARCHIV FUR HYDROBIOLOGIE, 132(2): 141-164, 1994.

[2]Glibert P.M., Wilkerson F.P., Dugdale R.C., Raven J.A., Dupont C.L., Leavitt P.R., Parker A.E., Burkholder J.M., and Kana T.M.. Pluses and minuses of ammonium and nitrate uptake and assimilation by phytoplankton and implications for productivity and community composition, with emphasis on nitrogen-enriched conditions. Limnology and Oceanography, 61(1): 165-197.

[3]Gobler C.J., Burkholder J.M., Davis T.W., Harke M.J., Johengen T., Stow C.A., and van de Waal D.B.. The dual role of nitrogen supply in controlling the growth and toxicity of cyanobacterial blooms. Harmful Algae, 54: 87-97, 2016.

[4]Andersen I.M., Williamson T.J., Gonzalez M.J., and Vanni A.J. Nitrate, ammonium, and phosphorus drive seasonal nutrient limitation of chlorophytes, cyanobacteria, and diatoms in a hyper-eutrophic reservoir. Limnology and Oceanography, 9999: 1-17, 2019.

[5]Yu L., Rozemeijer J.C., van der Velde Y., van Breukelen B.M., Ouboter M., and Broers H.P.. Urban hydrogeology: Transport routes and mixing of water and solutes in a groundwater influenced urban lowland catchment. Science of the Total Environment, 678: 288-300, 2019.

*"The Geuzenveld study site is part of an urban lowland polder catchment, which is characterized by groundwater seepage that constantly determines the surface water quality, being the main source of solutes in the water system. The groundwater seepage is a continuous source of anoxic, iron and nutrient rich slightly brackish waters. Yu et al. (2019) presented the results of a 10-year monitoring program describing the main processes determining the water quality in the catchments, which is dominated by mixing of runoff water and seepage water. A high-frequency monitoring campaign was set-up to further unravel the temporal pattern on the nutrient N and P, of which N is typically present in the form of $NH_4$ from groundwater."*

Moreover, we improved Figure 1, to better present the study area (see further comments 69).

**$NO_3$ on the other hand is not included in the discussion, mainly because the planned $NO_3$ measurements didn't work in a proper way. This raises the question why $NO_3$ wasn't at least monitored by regular grab samples (low frequency). This would have also helped to confirm the conceptual model for nutrition dynamics in low land polders.**

Indeed, the nitrate was measured in the grab samples and we published about those in Yu et al. 2019. We now realize that we should present this information also in this paper, so we added TN and $NO_3$ boxplots to Figure 6 in the original manuscript. $NO_3$ was measured in a monthly routine from 2006-2016 and in biweeky frequency from 2016 to 2017 by the water authority. Those results showed much lower concentrations of $NO_3$ (median 0.1 mg N/L) than $NH_4$ (median 3 mg N/L) in the water column, indicating the dominance in $NH_4$ over $NO_3$ (see Figure below). Moreover, we included TN and $NO_3$ in the text where we list the parameters measured in the routine monitoring campaign.

New Figure 6 Monthly measurement of TN, NH4 -N, $NO_3$ - N, chlorophyll-a (Chlor), $O_2$, organic N/ TN and $NH_4$-N /TN (NH4/N) mass ratio, pH , water transparency, and suspended solids in Geuzenveld from 2007 to 2018. X axis is month. (The plots of TN, NO3 and organic-N/TN were added to the original Figure in the main manuscript).

**3. Though high frequency data were collected, they haven't been really used in the analysis to precisely describe single events, except for the analysis of single pumping events. There would probably have been the same results when the data resolution would have been one day. Mostly concentrations are discussed in an annual or seasonal context. Please elaborate more on the added value of a 20 min sampling compared to a daily sampling in the discussion. Probably, the focus of the paper should not be too much on high-resolution sampling (e.g. in the title).**

Here we do not completely agree with the reviewer, because we discuss the event scales in detail in the section 2.3.1 and 3.3.2, and Figures 4 and 5 both focus on the event scale (see also reply to comment no.7). In section 4.5 the measured patterns were addressed in detail; we observe large fluctuations within a single day for $NH_4$ (1 mg l$^{-1}$) and P (2 mg l$^{-1}$). The merits of high-frequency sampling are highest for the nutrient concentrations, for which the temporal patterns were not understood using the analysis of biweekly grab samples campaigns (see Yu et al. 2019 for the complete analysis). One of the conclusions is that the responses to precipitation and pumping events were very different from the reported in previous literature, and we gained much understanding about possible processes from the high-resolution dataset that we could not assess otherwise. Moreover, pumping events would be missed completely in a one-day sampling frequency as pumping occurs almost solely overnight. However, we did not specifically answer the question raised by the reviewer about the benefits of our high frequency sampling campaign with a daily sampling schedule. We published about these aspects in earlier papers (e.g. Van Geer et al. 2016, Rozemeijer at al. 2010, papers that we cite in the references section). We added the following text to the discussion (section 4.5):

*"This type of event scale dynamics would be easily missed in a daily or lower frequency sampling schedule, especially because pumping occurs almost solely overnight in our regulated catchments. As such, only a sampling schedule with 7 hours intervals (e.g. Neal et al. 2011)[6] or high-frequency monitoring is able to catch the short-term dynamics (Van Geer et al. 2016, Van der Grift et al. 2016)"*
* * *
[6] Neal, C., Reynolds, B., Norris, D., Kirchner, J. W., Neal, M., Rowland, P., Wickham H., Harman S., Armstrong L., Sleep D., Lawlor, A., Woods C., Williams B., Fry M., Newton G., Wright D.. Three decades of water quality measurements from the Upper Severn experimental catchments at Plynlimon, Wales: an openly accessible data resource for research, modelling, environmental management and education. Hydrological Processes, 25(24), 3818-3830, 2011.

**4. A statistical analysis of the data is completely missing. All processes seem to be deduced by just visually comparing the graphs, without calculating e.g. correlation coefficients.**

We intentionally pursued understanding about the hydrological and hydrochemical temporal patterns through studying the high-frequency dataset at 3 time scales, as much is to be learned from these patterns. We did not report about the statistical patterns in order to keep the manuscript concise. As both reviewer 1 and 3 requested for a more thorough statistical treatment of our data, we added this information in the Supporting Information and summarized the results by mentioning correlation coefficients and p-values in the main text, where appropriate. We refer to the response to reviewer 3 for more details.

**5. The authors developed a mixing model to determine which amount of nutrients can be attributed to hydrological mixing and which to biological processes. Latter one is derived by the discrepancy between model results and measured concentration. Electrical conductivity (EC) acts as a conservative tracer in this case. However, the model failed to reproduce EC within the polder after November 2016. This raises the question whether the amount of biological processes for nitrogen and phosphorus can be determined on this basis. This uncertainty should be discussed in more detail.**

The model is meant for illustrating an ideal situation that the ditch water is a pure mixing between groundwater and rainwater, without any physi-bio-chemical disturbances. The good fitness of the general pattern and in the first 6 months between the modeled and measured data of EC indicates that the hydrological process we assumed are convincing. Even though the modelled EC is lower than the measurements from mid-November onwards, the dynamics and amplitudes of the temporal pattern remain consistent with the period before Nov 2016, from which we deduced that the patterns is governed by the hydrological mixing process. During the later period, the deviation between the model and the measurement stays constant, which seems to be induced by underestimating the groundwater contribution other than by some biogeochemical processes. We added explanation for the deviation. If $NH_4$ and TP patterns would be governed by the hydrological mixing, the dynamics and amplitudes of these measurement should also be stationary in time. However, $NH_4$ and TP measured data drifted drastically away. There are no other processes other than biogeochemical processes that can explain those discrepancies. As such we are confident with our inference on the biogeochemical processes.

To clarify the point, we added to line 267: *"After that, the conservative mixing approach underestimated the EC but the main patterns were still reproduced; as groundwater is the only contributor to the high EC due to the seepage of quite mineralized, slightly brackish water, the model must underestimate the seepage flux from November 2016 on".* Further changes to the text with additional discussion of the patterns are introduced in the replies to comments 50 and 60.

**6. The authors are classifying their study in the context of eutrophication: the findings of this study are meant to help water managers to mitigate eutrophication. However, there are no suggestions how the results of the study can be used to do so.**

*We agree with the reviewer that the part of providing practical suggestions for water management in this paper could be improved. We added a section (section 4.6 Implications for urban water management in low lying catchments) on this topic. It reads:*

"This study demonstrated high frequency monitoring technology to be an effective tool for understanding the complex water quality dynamics. Investment in high frequency monitoring would greatly benefit the management of urban lowlands with substantial groundwater seepage by elucidating the determining biogeochemical processes and nutrient temporal patterns for realizing efficient mitigation and control of eutrophication. For example, a direct treatment of the drain water applying constructed wet lands could be considered as a mitigation measure in low lying areas with artificial water systems that resemble the Amsterdam region, e.g. in cities such as New Orleans, Shanghai and Dhaka. Centralizing the treatment of discharge water is also recommended, for instance by harvesting N as phytoplankton from the discharge during spring, or filtrating P at the pumping station during winter. Measures that artificially increase oxygen concentrations in the waters, such as the inlet of oxygen rich water, aeration by fountains or the artificial introduction of grazers or macrophytes may be considered to improve the ecological status of these urban waters. Moreover, aeration of the water in summer and autumn would possibly enhance processes such as nitrification and anammox, eventually converting $NH_4$ to $N_2$, before the water is discharged to downstream waters. Importantly, before the application of any measures or maintenance in urban low lying catchments, managers should evaluate the potential effects on the biological and chemical resilience of the ecosystem communities, e.g. dredging of a layer with abundant benthic activity might destroy an important buffer to nutrients in growing seasons, especially P."

**7. An aim of this study was to analyse and compare annual scale, precipitation events and pumping events (lines 88-90), but this scheme can't be found in the discussion. The three scales are mixed up rather than to distinguish between the dynamics of the different time scales.**

As suggested by the reviewer, we also first wrote a time-scale based discussion structure, but discussing each of the solutes for each timescale became long, repetitive and tedious. Therefore, we found a solute based discussion structure easier to follow. As the solute behavior increases in complexity (i.e. the number of processes that affect it), we feel that this is the natural structure that best fits our manuscript. This structure allows us to focus on the drivers, more than on what occurs during each timescale. In order to avoid false expectations we removed suggestions that we would compare the 3 time scales, as we analyze them complementary instead. We realize that most of our conclusions deal with the seasonal scale as those are eventually most relevant for fluxes of nutrients leaving the catchment, but we tried to make the conclusions more balanced also making reference to shorter time scales. For example, we added in the conclusion section: "*Unlike many other natural and artificial catchments, rainfall and pumping events did not increase turbidity or TP concentrations at the short time scale, rather reduced turbidity and TP because of iron hydroxide precipitation and removal of phytoplankton from the catchment.*"

**Specific comments**

**8. L22: 'through variation of the intensity and duration of the events' I don't understand the meaning of this sentence.**

We removed the sentence and reworded the preceding sentence into:

"*Mixing of upwelling groundwater and runoff from precipitation on pavements and roofs was the dominant hydrological process and governed the temporal pattern of the EC, while N and P fluxes from the polder were also significantly regulated by primary production and iron transformations*".

**9. L23: Is NH4 really the dominant form in surface waters? I know many examples from Europe where nitrate dominates. Furthermore, NH4 gets nitrified to NO3, leading to decrease in NH4 and increase in NO3 concentrations. Maybe the authors mean that NH4 is the dominant form in urban water bodies?**

We agree that the statement is not precise enough. Indeed, in many surface waters, $NO_3$ is the dominant form. However, ammonium is the main N-form in the anoxic groundwater seepage that has passed organic rich (peat) layers in the subsurface. As a consequence, ammonium also dominates in the low lying polder catchments, like Geuzenveld, that receive this seepage water. What we mean is that $NH_4$ is the dominant form of N in Geuzenveld, which is fed by anoxic, old groundwater sourced from the organic matter abundant subsurface. So we clarified our message which was too generally stated. We now mention the source of $NH_4$ which was elaborated in previous papers (Yu et al. 2018, 2019). It now reads:

"*In our groundwater-seepage controlled catchment, $NH_4$ appeared to be the dominant form of N with surface water concentrations in the range of 2-6 mg N/L which stems from production in a organic -rich subsurface. The concentrations of ammonium in the surface water were governed by mixing of groundwater and runoff water in autumn and winter and showed reduced concentrations up to 0.1 mg N/L during the algae growing season in spring*".

We further made the nitrate measurements available (see reply comment 2) which gives evidence for the dominance of $NH_4$ to $NO_3$ for the readers.

**10. L 24, 25: 'low concentrations during the algae growing season, while concentrations were governed by mixing of groundwater and precipitation inputs in the late autumn and winter.' This sentences only makes sense, when the authors mention the concentrations in autumn and winter as well.**

Agreed. We added the concentration ranges, following the reply on comment 9.

**11. L26 – 28: The two sentences have nearly the same content: release of reduced iron causes turbidity.**

Agreed. We deleted "*Rapid Fe²⁺ oxidation in the water column is the major cause of turbidity.*" And formulated new sentences to clarify what we meant. We emphasize the different position of the iron oxides (water columns versus sediments).

*"Total P and turbidity were high during winter, due to the release of reduced iron and P from anoxic sediment to the water column, where $Fe^{2+}$ was rapidly oxidised into iron oxides which contributed to turbidity. In the other seasons, P is retained in the sediment by precipitation of iron oxides".*

Moreover, for consistency reasons, we added the concentration ranges for P as well, following the reply on comment 9.

**12. L 29, 30: Was organic N measured? A denitrification needs anaerobic conditions, while in spring O2 concentrations were rather high, how does that fit together?**

In our dataset, organic N was not directly measured, but Kjeldahl-N was during the biweekly grab sampling campaigns (Kjeldahl nitrogen is the sum of the ammonium nitrogen and organic nitrogen). By subtracting NH₄-N from Kjeldahl-N we came up with an estimate of organic N. In order to better clarify the nitrogen species in the surface water, we extended Figure 6 of the manuscript to include NO₃, TN and organic-N/TN ratio (see the attached new Figure 6 in the system). Clearly the proportion of organic N over total N is increased in spring, wheareas NH₄/TN decreased. In Spring, organic-N occupied more than 50% of total in spring. This was implicitly evaluated in the original paper to describe the organic-N pattern, but we now choose to present this explicitly.

As phytoplankton produces oxygen this gives rise to increased oxygen concentration in the water column, and also increase pH due to the uptake of CO₂ from the water. As we discussed in the rest of the paper, the ditches were fed by anoxic groundwater constantly over the year, which created a sediment with low oxygen level, and oxygen was mainly supplied by runoff, by phytoplankton growth and growth of benthic algae.

We made the following changes to the manuscript:
- We added NO₃, TN and Kjeldahl-N to Figure 6
- Mentioning TN and Kjeldahl-N in the list of low-frequency parameters
- Section 2.2.2. added: "Organic-N was estimated by subtracting NH₄-N from Kjeldahl-N.
- Section 4.2 Line 362-364 changed into: *"Growth of primary producers results in a consumption of ammonium, phosphate and a production of organic-N, chlorophyll, oxygen, and suspended solids, and led to a relatively higher pH because of the uptake of CO₂ (Figure 6). This patterns is also clearly reflected in the shift in the NH₄/TN and organic-N/TN ratios during spring (Figure 6)"*
- Section 4.5: *"Fig.6 NH₄/N and organic-N/TN"*

**13. L 41: I would replace 'end up' by 'reaching' or something similar**

Agreed. Changed accordingly, replaced "*end up*" by "*reaching*"

**14. L 45: This sentence belongs to the following paragraph.**

Agreed. Moved "*Nutrients dynamics are governed by biological, chemical, and physical processes and their interactions.*" to Line 47 before "*Assimilation….*"

**15. L 47: it should be 'in the aquatic environment' or 'in aquatic environments'**

Agreed. Added "*the*" before "*aquatic environment*"

**16. L 48, 49: molecular nitrogen and phosphate was not mentioned until now, the authors should introduce N2 and PO4 first, like they have done it for nitrate and ammonium**

We added in Section 1: *"Recently, groundwater has been identified as another important source of N and P in cities situated in low-lying deltas, where dissolved NH₄ and PO₄ in groundwater seep up into urban surface water (Yu et al, 2018 & 2019)".*

**17. L 49: 'NH4 is the preferred N-form by microbes'. There are also other microbes which prefer different forms of Nitrogen (like the authors mentioned in the sentence before).**

Agreed. The reference we referred to was done in European estuaries, and their conclusions were made for the tidal estuary environment.

We added "*in some cases like in estuaries*" after "*by microbes*"

**18. L 50, 51: The content of this sentence is obvious, when there is no NH4 and NO3 the uptake of the substances can't reach a maximum.**

See previous comment.

**19. L 53: 'Under aerobic conditions, NH4 can be oxidized to NO3 through nitrification by nitrifying microbes even under cold conditions (below 10 °C ), which is an O2 consuming, acid generating process' Please revise the sentence structure. It sounds as if the nitrification under cold conditions is O2 consuming.**

Agreed. Changed into: "*Under aerobic conditions, NH$_4$ can be oxidized to NO$_3$ through nitrification by nitrifying microbes, which is an O$_2$ consuming and acid generating process. Nitrification even occurs under cold conditions (below 10 °C)*"

**20. L 59: 'during events' what kind of events? The authors should be more precise with their expressions 'during hydrological/precipitation events'**

*Agreed. The reference is about heavy precipitation events. So, we added "precipitation" before "events".*

**21. L 60: 'and chemical reactions....' This part of the sentence doesn't fit substantively to the ones before, which were about transport processes not transformation processes. The latter aspect is discussed in the following paragraph.**

Agreed. Deleted ", and chemical reactions such as mineral precipitation with associated P incorporation cause removal from water column (Rozemeijer et al., 2010a; Van der Grift et al., 2014; Yu et al., 2019)"

Line 67, replaced *"Griffioen, 2006; van der Grift, 2014"* by *"Griffioen, 2006; Rozemeijer et al., 2010a; van der Grift, 2014; Yu et al., 2019"*

**22. L 74: 'N and P dynamics, for instance its response...' Please replace 'its' by 'the' or 'their'**

Agreed and done

**23. L 83. Please replace '...insight in...' by '...insight into...'**

Agreed and done.

**24. L 86 – 88: Please replace 'We conducted a one-year high frequency monitoring campaign in 2016-2017, measured parameters EC, NH4, TP, turbidity and water temperature. ' by 'We conducted a one-year high frequency monitoring campaign in 2016-2017. Measured parameters were EC, NH4, TP, turbidity and water temperature.'**

Agreed and done.

**25. L 97, 98: Do the authors mean that groundwater seeps into the catchment because the water level of the groundwater is higher than the sole of the channels and the drain system?**

The groundwater head is higher than the water level in the channels and the level of the drains. We changed the text to clarify this.

It now reads: "*Geuzenveld is a groundwater fed catchment due to the constantly higher groundwater head (-2.5 ~ -3 m NAP) in the main aquifer relative to the surface water level in the polder ditches (~ -4.25 m NAP) (Fig.2). To keep the foundations of the building dry, there is a groundwater drainage system placed under an artificial sandy layer, right on top of a natural clay layer. The drain elevations range from -4.84 to -4.61 m NAP, which is below the phreatic groundwater level throughout the year, making sure that groundwater seepage either discharges through the drains or the ditches*"

**26. L 99: 'much higher' How much is much higher?**

Agreed. Changed accordingly, see comment 25.

**27. L 102: NAP doesn't need to be explained, naming the abbreviation should be enough.**

Agreed. Deleted *"(NAP: Normalized Amsterdam Peil, a known international standard conforming to mean sea level)"*

**28. L 117: The temporal resolution (20 min?) is missing in the description. I could only find it in the abstract.**

Agreed. Line 124 added *"The monitoring frequencies were set to 20 mins, 10 mins, 5 mins, 5 mins and 5 mins interval for TP, NH$_4$-N, turbidity, EC and water temperature, respectively."* at the end.

**29. L 135: 'was calibrated' instead of 'was calibrating'**

Agreed. Changed accordingly.

**30. L 153: What was monitored by Waternet? – 'Waternet has monitored the water quality'?**

Agreed. Added "*the water quality*" after "*Since 2006, Waternet has monitored*"

**31. L 154: 'the frequency became twice...' Frequency cannot increase by itself: 'frequency was increased....."**

Agreed. Changed "*became*" into "*was increased to*"

**32. L 154: 'were measured in this dataset' In a data set nothing can me measured. Please be more precise.**

Agreed, reworded.
Deleted *"Many parameters were measured in this dataset, but for this research"*.
Added *"parameters from the routine monitoring campaign" after " We selected the following..."*

**33. L 172: Potential evapotranspiration is a virtual measure derived from meteorological data. It doesn't give an actually evaporated water volume. How is the use of potential evapotranspiration justified? Actual evapotranspiration should rather be used in this case.**

We did not have measurement for the actual evapotranspiration. So, we used the potential evapotranspiration instead. Potential evapotranspiration was downloaded from the meteorological station (2 km away from the study area). The results were derived from Makkink calculation as noted in the downloaded file. Given the year-round seepage conditions throughout the polder, combined with an artificially drained subsurface, we assumed that potential evapotranspiration is close to the actual evapotranpiration (no water shortages occur here).

We added: *"Given the year-round seepage conditions throughout the polder, combined with an artificially drained subsurface, we assumed that actual evapotranspiration is close to the actual evapotranpiration as no water shortages occur in our situation".*

**34. L 172: I suppose groundwater seepage S stems from outside of the polder. How are the values of this variable derived? Calibration? And why is it multiplied by the area of the polder? Please clarify these issues.**

We did not have measurement for the seepage neither within nor outside of the polder. In this study, we used the difference between groundwater head in the first aquifer and the surface water level (Figure 2) to estimate a range of the seepage. The actual number of 1.5 mm per day was chosen based on the behavior of the mixing model (Figure 3) and was calibrated against the water level changes observed near the pumping station.

As we assumed a homogeneous distribution of the seepage within the polder, the calculation of the flow rate of groundwater was multiplied by the area of the polder instead of by the area of the ditches. We think it is a convincing assumption as the drain system underground is effectively collecting and transporting seepage from everywhere of the polder to the ditches.

*We added: "In this study, we used the difference between groundwater head in the first aquifer and the surface water level (Figure 2) to estimate a range of the seepage. The actual number of 1.5 mm per day was chosen based on the behavior of the mixing model and calibrated using the measured water levels (Figure 3)."*

**35. L 174: Naming the variables in the order of their occurrence in the formulas would be easier to follow.**

Agreed.

Changed Line 174-176 from "*L is surface water level in the ditches, V is total water volume in the ditches, P is precipitation, S is a constant seepage, E is potential evapotranspiration, A polder is area of the polder, A ditch the area of the ditches in the polder. Water level L determines the activation of pumping activity. Pump(t) is water volume being pumped out with maximum capacity 216 $m^3 h^{-1}$.*" into "***V** is total water volume in the ditches, **P** is precipitation, **S** is a constant seepage, **E** is potential evapotranspiration, **$A_{polder}$** is area of the polder, **Pump(t)** is water volume being pumped out with maximum capacity 216 $m^3 h^{-1}$, **$A_{ditch}$** the area of the ditches in the polder. **L** is surface water level in the ditches. Water level **L** determines the activation of pumping activity.*"

Changed Line 183-184 from "*C(t) is solute concentration at time t, Cgw is the average groundwater concentration, Cp is the average concentration in runoff, V is the ditch water volume given by equation (1).*" into "***V** is the ditch water volume given by equation (1), **C(t)** is solute concentration at time t, **$C_{gw}$** is the average groundwater concentration, **$C_p$** is the average concentration in runoff.*"

**36. L 182: 'd(VC)" is not explained in the text, I guess it is the concentration of the ditch water?**

Thank the reviewer for his comment.

"d(VC)/dt" is a expression for the change of solute mass in an unit time. Both "V" and "C" were explained in the text.

**37. L 185. Is the high salt concentration really the reason for EC being a conservative tracer? Or do you mean that the concentration difference between the two water sources renders EC a useful tracer?**

Agreed. A high salt concentration does not render EC directly as a useful tracer. Indeed the high salt concentration difference between groundwater and rain makes EC behave as a valuable conservative tracer.

We rewrote the text as follows: *"In our study area, the EC is a useful water quality parameter for describing the mixing processes between groundwater and runoff water, as the EC represents the end members of the mixing: groundwater with an high EC (1750 µS/cm) and runoff water (100 µS/cm) with a low EC (see also Yu et al., 2019). Moreover, we assume that EC behaving as a conservative tracer as the EC is highly correlated with the Cl concentration ($R^2 = 0.71$, p-value < 0.05) and the temporal patterns of EC and Cl are very similar (see supplement Figure S1)".*

**38. L 193: 'simulated concentrations ... together with their high frequency...' exchange 'their' with 'the'**

Agreed. Changed *"their"* into *"the"*.

**39. L201: Rain events are very long (> 1 month), that seems to be more representative for a (sub-)season or similar. A rain event usually is shorter than a few days in central Europe.**

Thanks to the reviewer, it is a very good question.

The actual precipitation did not last for a month. In this paper, we defined a "rain event" based on the dilution pattern of EC (from EC started to be diluted until it recovered to the original level before the event). And we tried to cover the four seasons. This way of defining events is more helpful for gaining the main messages instead of lost in details. We think that the original text covers this aspect sufficiently.

**40. L 207, 208: Four times 'and' in one sentence. Please rephrase.**

Following suggestions of all reviewers to remove redundancy in the text, we skipped this introduction sentence.

**41. L 217: Why did the wet season start in October and end in February? Did the authors maybe calculate a cumulative water deficit? According to Figure 2 there has been quite a lot of rain up to the middle of**

**March. Further, a dry and wet season usually refers to a semi-arid climate, which Amsterdam is far from being in. Please think about re-naming the compared time spans and give details on how they were separated from each other.**

We better defined our wet and dry season in the text. The reviewer is right that this is not typically a semi-arid climate wet season, but we propose to keep the term and better define it instead, as winter and summer seasons would lead to further confusion. Indeed, we estimated the cumulative water deficit based on the amount of pumping. As indicated in the text, not so much the precipitation sum, but rather the frequency of pumping shows the "wet"and "dry"seasons most clearly. In accordance with the reply to reviewer 3, we quantified the pumping volumes over the wet and dry period.

We added: *"We defined the wet season based on the absence of a water deficit, which corresponds with the period of higher frequency pumping. This period is correspondingly characterized by the higher intensity of the water level fluctuations and covers the period October 2016 until the end of February in 2017 (Fig 2A and 2B). Typically, the dry season showed a higher water deficit, indicating water loss due to evapotranspiration under warmer conditions. The wet season is distinguished by higher average daily pumping volumes and lower water temperatures (Fig.2B) than the ones of the rest of the year (wet season: 997 m³/d, dry season: 787 m³/d).*

**42. L222: Please change '..period that the water temperature...' to '...during which the water temperature....'**

Agreed. Changed accordingly.

**43. L228 – 229: 'In contrast to the constant water level ranges from surface water regulation regime' I am not sure about what the authors want to say. Please clarify.**

Agreed.

Line 227, added *"(light blue)"* after *"Fig.2A"*.
Line 228, replaced *"water level ranges from surface water regulation regime, ..."* with *"surface water level (Fig.2A, dark blue), ...".*

**44. L237: Remove '...if there was no rain'.**

Agreed. Changed accordingly.

**45. L 237: 'this duration of the return' Bad expression**

Agreed. Changed *"This duration of the return to pre-event EC values"* into *"The duration of this process".*

**46. L 241- 244: The authors mention twice, that NH4 deviated from slope of EC.**

Agreed. Changed into "NH$_4$ decreased from around 4 mg L$^{-1}$ to around 2 mg L$^{-1}$ between the middle of June to the end of August 2016 and reached down to almost 0 mg L$^{-1}$in the second period. Whereas..."

**47. L 250: The authors refer to excessive precipitation, but unfortunately this is not shown in Figure 2.**

Added *"and a large pumping volume"* after *"correspond to excessive precipitation".*

The precipitation event coincides with a high pumping volume, so we added: *"and a high pumping volume"* to help find the peak we refer to.

**48. L247 – 252: The description of turbidity is rather confusing: 'Turbidity was constantly below 100 FNU' is followed by a peak description of 500 FNU. The authors also miss out, that there are several EC peaks during October. They also repeat the same content ('turbidity stayed around 200 FNU') in lines 249 and 251.**

We clarified and simplified the text:

*"Turbidity stayed below 60 FNU during the dry season until October 2016 and substantially increased after a first rain event to 500 FNU (more details refer to Figure S2 in supplementary information). A drop to about 200 FNU occurred right after*

*this first peak, which seemed to correspond to excessive precipitation and a large pumping volume (Fig.2B). Soon after, turbidity went up again and peaked at 1800 FNU. Turbidity levelled of towards values around 200 FNU for the rest of the wet season, and dropped below 60 FNU from April 2017 on".*

**49. L 259: Delete 'when'**

Agreed. Changed accordingly

**50. L265 – 274: The authors are writing that concentrations are captured well, but there are discrepancies of more than 50 %. Maybe the authors want to point out, that the dynamics are captured?**

Agreed, we actually mean that the dynamics and amplitudes are well captured, but not always the absolute values. We rephrased the complete paragraph to better describe our observations and to start the hypotheses which were further explored in the Discussion section.

*"A simple fixed-end-member mixing model was used to reconstruct the conservative mixing of EC, $NH_4$, and TP. The simulated and the measured EC, $NH_4$, and TP are plotted in Figure 3. By comparing the model results with the high frequency measurements, potential processes that might deprive or enrich nutrients relative to the conservative mixing process along the flow routes were inferred from the discrepancies between the modeled and the measured data. Figure 3(A) shows that the predicted and observed EC dynamics agree reasonably well from May to November 2016. After that, the conservative mixing approach underestimated the EC but the main dynamics and amplitude were still reproduced; as groundwater is the only contributor to the high EC due to the seepage of quite mineralized, slightly brackish water, the model must underestimate the seepage flux from November 2016 on. Overall, the observed dynamics of EC are consistent with mixing of high EC seepage water with low EC runoff water.*
*The dynamics of measured $NH_4$ concentrations show resemblance to the model results, especially during the wet season. Clearly, $NH_4$ is diluted during the rain events and a gradual increase of $NH_4$ starts after each rain even during the wet season showing slopes that resemble the model reconstruction. Over the whole period, measured $NH_4$ concentrations are overestimated by the model, indicating that some $NH_4$ is probably lost due to non-conservative processes. This is especially true for the spring season of 2017, where $NH_4$ concentrations must be controlled by other processes. Concentrations of TP are generally far below the conservative model reconstruction, except between the end of November and the beginning of March. During this particular period the minimum measured TP concentrations are captured nicely by the conservative model, however distinct peaks up to 3 mg L-1 are not captured by the model and must have different physical or chemical processes determining them".*

**51. L265 – 274. Since water levels were measured: why were the model results of the water levels (L(t)) not compared to measurements?**

The reviewer is acknowledged for this question. We did compare the measurement to the modeled results, in order to calibrate the groundwater seepage rate for the conservative model. We added the comparison in the supplementary information, but kept it out of the manuscript itself, in order to keep it concise.

*We added: "and calibrated using the measured water levels (Figure S6)."*

**52. L 277, 278: What is the criterion for a 'significant dilution event'?**

Agreed, we added our criterion. Line 201 added *"((EC value reduced by over 35%))"* after *"dilution extent of EC".*

**53. L 287, 288: Why does event 2 follow EC but not event 3? According to figure 3 NH4 concentration seems to increase parallel to EC. The authors attribute the missing dilution only partly to the data gap. I don't think a statement about the missing part can be made, when there are no data available. Further, the word 'partly' implies that there was no strong dilution.**

Agree, the use of "partly" is confusing. Our meaning is that we cannot tell whether NH4 followed EC in event 3 and 4 like in event 1 and 4, that is because we don't have data (data gaps) of $NH_4$ in event 3 and 4.

Line 286-287, rephrased into "The dilution patterns of the $NH_4$ in events 1 and 2 were similar to those of EC. Due to the data gaps of $NH_4$ in event 3 and 4 we cannot describe the pattern of $NH_4$ in these two events."

**54. L 289: Can you be sure about the dilution? When you compare the high frequency measurements with grab samples (figure 2) 0.35 mg TP/l seems to lie within the uncertainty range of the high frequency measurements. Please discuss a potential sampling uncertainty.**

Thanks reviewer for his comments, which made us check our data again. We are quite sure about the dilution patterns for TP during the dry period, as the available grab samples confirm the high frequency results. Some more details are provided below.

We added: *"This pattern is nicely reflected in the available grab samples of that event period, confirming the measurement uncertainty is limited. The response during period 2 is unclear because of too many data gaps, but in general TP show a dilution patterns during rainfall events in the dry and warm season"*

Dilution:

    To answer the questions, we plotted Event 1 with the TP grab sampling data (red dot) as below. It shows that TP started to decline at the same time when EC started to drop. It was due to the precipitation between June 20 and 24. For both EC and TP, we are sure that the pattern above is dilution in the summer. However, redox reactions (iron oxidation) have to be taken into consideration of the decreasing pattern in the late autumn and winter.

[Figure]

Sampling uncertainty or measurement uncertainty:

Table below reveals that the grab sampling results fit well with the high frequency monitoring time series. The sampling uncertainties are low. It confirms the reliability of the high frequency data, as well as the dilution pattern we proposed above.

| Date | TP grab mg P/L | TP HQ (mg P/L) | | Uncertainty |
| --- | --- | --- | --- | --- |
| | | range | mean | |
| 10-06-2016 | 0.70 | 0.58-0.69 | 0.63 | 0.01-0.11 |
| 17-06-2016 | 0.66 | 0.60-0.66 | 0.63 | 0-0.06 |
| 27-06-2016 | 0.79 | 0.71-0.76 | 0.74 | 0.03-0.08 |
| 11-07-2016 | 0.52 | 0.51-0.54 | 0.52 | 0.01-0.03 |

**55. L290: The authors forgot to mention, that TP was also falling again after reaching 0.8 mg/l**

The reviewer is referred to the figure above. That the TP fell again might be due to the occurrence of precipitation that day (June 27). However the variation (0.05 mg P/L) is insignificant.

**56. L292: There were more small rainfall events during recovery period of event 3 compared to event 4**

Agreed. What we wanted to express is the occurrence of the high concentration of TP when rainfall was absent, regardless of the dilution or recovery period.

Line 291-293, changed into "*In events 3 and 4, rainfall events are less intensive but last longer. TP concentrations increased up to 3 mg L$^{-1}$ when rainfall was absent. No such pattern occurred to TP in the beginning of spring (event 4).* ".

**57. L310: 'pumping has the least influence on NH4 in winter' It is difficult for the reader to relate this to figure 5, because the scaling for NH4 concentrations is different for every event**

Agreed. But the trend of NH$_4$ cannot be seen if all the four events are at the same scale. So, we decided to add detail information to elaborate the concentration variations during the pumping events. This can give readers more precise information.

We changed the text into "*The concentrations of NH4 were disturbed the least (event 1: 0.38 mg N L$^{-1}$, event 2: 1.02 mg N L$^{-1}$, event 3: 0.15 mg N L$^{-1}$ , event 4: 0.76 mg N L$^{-1}$) by pumping events in winter (event 3).*"

**58. L316: The authors suggest that turbidity is influenced by pre-event conditions, but the reader has no specific information about the pre-event conditions. This point is also not further discussed in the Discussion, where this sentence should be placed anyway.**

We skipped this sentence.

**59. L 329: 'Runoff in Geuzenveld has waters with EC....' – Please rephrase.**

Agreed.
Deleted "waters with an "
Replaced "low compared to" by "lower than"
Added "EC" after "...the groundwater"

**60. L332: '...the mixing model...which revealed close similarity to the measurement'. This statement is wrong. There is a big discrepancy between model and measurements in the second part.**

We rephrased the text to resemble the right observation of the reviewer. We believe that the discrepancy is due to a shift during a specific short period, and adapted the text to reflect that position.

*"This presumption is supported by the agreement between modelled and measured EC dynamics for the period between May to November 2016. Precipitation events diluted the EC values at the pumping station, and the magnitude of dilution depended on the intensity of precipitation; heavy rainfall resulted in low EC values (Fig.2D and Fig.4). In periods with absence of rainfall, the EC values follow a recovery curve that resembles a linearly mixed reservoir with concentrations increasing to values that approach the EC of the continuous groundwater supply of around 1500 μS/cm. After November 2016, the conservative mixing approach underestimated the EC but the main dynamics were still reproduced and the amplitude of the EC dynamics remains similar to the model results, except for the short period Nov 20th to Dec 1st, 2016. Starting around Nov 20th, the EC started to increase relative to the dry season before. It coincides with an intensive pumping event after the first intensive rainfall event that happened after a prolonged period of water deficit. This may be related with a first flush from the drain system that starts to be activated more strongly, thus removing clogged material and lowering the overall resistance of the drain system for shallow and deep groundwater flow. It suggests that this triggered the inflow of somewhat more mineralized groundwater relative to the period before, creating a shift in the EC towards ~250 uS/cm higher values that continued during the remainder of the monitoring campaign. It appeared that it raised the EC, but did not change the amplitude or dynamics of the EC during the remainder of that period (Fig 2 and 3, Table S6). An alternative reason for the higher EC starting from November, 2016 on, would be the application of road salts during the winter period. Although freezing conditions occurred from November onwards, we did not find any evidence for the effects of road salts, as the chloride concentrations in the grab samples only showed two higher measurements, one in December 2016 and one in January 2017 (see Supplement, Figure S2.) So, overall, the observed dynamics of EC are consistent with mixing of high EC seepage water with low EC runoff water."*

**61. L341 – 347: Only the discrepancies during winter are discussed, but measurements and the model reach into middle of June.**

Agreed. See reply to comment 60.

**62. L 397: NH4 can be consumed by nitrifying bacteria (not by nitrification).**

Rephrased as: *"Apart from primary production, $NH_4$ can be oxidized to NO3 in the process of microbial nitrification (Zhou et al., 2015)".*

**63. L 398, 399: Denitrification and anammox are two different processes and the chemical equation doesn't fit to neither of them. For anammox NO2 is needed, not NO3. I am also wondering why nitrification and denitrification are not discussed apart from this two sentences. Nitrification is also an oxygen consuming and NH4 reducing process. While denitrification can take place under anoxic conditions.**

We skipped the reference as the nitrate concentrations are typically very low in this seepage catchment (see Figure 6). Nitrate and $NO_2$ might be intermediate species in the primary production or chemical reactions, but do not affect the main patterns of ammonium uptake in spring and the transfer to organic-N as was discussed here.

**64. L 417 Turbidity only increased for a short period (end of October to middle of November)**

From the end of October until the end of February, Turbidity was most of the time above 200 FNU which can be considered as high level compare to the dry season (typically below 50 FNU).

We adapted the text to clarify this: *"From the late autumn onwards, turbidity and total Fe concentrations significantly increased, peaking first to 1800 FNU and staying at a plateau of ~200 NFU during the rest of the cold and wet season (Fig.2). During this period the water turned brownish and transparency declined (Fig.6)"*

**65. L 418: 'Iron-rich particles are the most likely source of turbidity in freshwater': Concentration of iron particles is high until February. If this is true, why is turbidity low in February? Please clarify.**

We clarified that turbidity values of ~200 FNU are elevated relative to the spring and summer seasons, but this was probably missed because it is less prominent than the 1800 FNU peak in late autumn. Actually, the 1800 FNU peak started a new situation with elevated turbidity, and later increasing TP concentrations once the ironhydroxide layer at the sediment-interface became completely dissolved (see Figure 7). Complementary to the addition of the text of comment 64, we added the next sentence halfway the section to further explain our hypothesis:

*"We suggest that the turbidity peak of 1800 FNU is caused by the mineralization of the benthic algae once they die off when light and temperature conditions decrease, combined with the shift of ironhydroxide formation from the sediment-water interface to the water column. The latter process continues through the whole winter season, until primary production restarts in spring (Figure 7)"*

**66. L 433: '...turbidity became high' according to figure 2 turbidity wasn't high in this time span.**

See reply to previous comments. Turbidity was most of the time above 200 FNU which can be considered as high level compare to other time (mostly below 50 FNU), please refer to the text in line 247-252.

**67. L 447: 'relatively low in oxygen (because of warming) ' Additionally, a reason for reduced oxygen might be an increase in O2 consumption by microorganisms.**

We change the text to also mention this possibility: *"and relatively low in oxygen due to the continuous supply of anoxic groundwater, the mere absence of $O_2$ -rich runoff, the oxidation process of Fe(II) and possibly by microbial organic matter decomposition during warm periods with relatively stagnant water."*

**68. L 523 – 664: The references are not completely in an alphabetical order and slightly different citation styles were used (e.g. sometimes DOI is written in capital letters, sometimes not)**

Agreed. Changed accordingly.

**69. Figure 1: Readers have to guess the channel after the pumping station is Boezem Haalemmerweg and whether the left drainage system is the secondary water channel which Greuzenveld is connected to. The map above the Google Maps Card with the location is too small and doesn't help to understand the system. Please provide a better overview map of the study area.**

We provided a better location map to help understand the system, which shows the position of the polder, the pumping station, main flow direction and overall setting within Amsterdam. It is uploaded in the "Fig.1" in the system.

**70. Figure 2: The discrete sampling data points are hard to identify;**

Agreed. Changed accordingly as below.

We found the detailed precipitation data is more useful when interpret the data in the event scales. For the annual scale, we think showing the water surplus and deficiency (precipitation minus potential evapotranspiration) gives a better representation, especially to derive the wet and dry periods which are based on water deficits and not precipitation alone.

**71. Figure 3: 'measured' and 'modelled' timeseries overlap in the same colour. It is not visible whether 'TP modelled' shows the same peaks like 'TP measured'. Choose different colours.**

We used green color to identify the measured TP, making it consistent with Figure 4. We now use consistent colors throughout the graphics.

**72. Figure 4/5: A rearrangement of graphs and scales could add to a better understanding of the figures.**

We were aware of the problem of the scales. For those parameters with small variations, we plot them in the same scales for all events, such as for water temperature, rain, surface water level, EC and NH4. But the ranges of TP and turbidity are beyond the possibility for plotting all the events in the same scales without losing the patterns. The arrangement of the parameters in Figure 4 and 5 is determined by the similarity of their behavior. For instance, EC shows more similar pattern with NH4 than with TP or turbidity, same for the TP and turbidity group. Besides, those groups are consistent with the groups in Figure 2. So, we intend to keep the Figures as they are, except that we will combine water temperature and precipitation rate in the first row of figures 4 and 5, as suggested in comment 73. This will avoid having 3 y-scales in row 2 and will help better identify the main patterns.

**73. Figure 5: The reader can't distinguish between day and night time, though the authors discuss this in chapter 3.3.2 based on this figure; while the first block only contains water temperature, the second block contains three measured parameters. There is room for improvement of visibility.**

Agreed. We define day from 7 am to 8 pm in autumn and winter, and 6 am to 10 pm in spring and summer. We will indeed combine water temperate and precipitation rate in the first row of figures 4 and 5. This will avoid having 3 y-scales in row 2 and will help better identify the main patterns.

**Reply to reviewer 2**
**This manuscript describes a high frequent monitoring study of water quality in a groundwater-fed urban ditch. The monitoring allows to elucidate governing processes of water quality and the authors do a good job in trying to explain the observed water quality parameters. The observations are quite specific for the study area at hand given the specific pumping management, and hence extrapolation to other catchments would be less obvious, which limits the generalisation of the results. That is the major draw-back. Sometimes the authors also draw far reaching conclusions which need further confirmation. Given the fact that the observations are sound and well described, and discussion needs some further confirmation, I rate this publication to be acceptable for publication with minor revisions. The revisions should help to improve the readability and conclusions that can be drawn from this case study.**

We thank Professor Piet Seuntjens (Reviewer 2) for his compliment and for his time and valuable review which led to a clearly improved paper. We are glad that the reviewer recognizes our efforts to understand the nutrient dynamics in this complex system, which involved combining and integrating a complex dataset.

We agree that some results and interpretation are highly site specific, given the local conditions with high seepage rates and the human controlled water level regime. However, as we stated in the discussion (?) we also see that these types of low-lying, artificial polder catchments are becoming a more common phenomenon, mainly in subsiding delta-cities. Therefore, the main message of the hard to manage groundwater impact on the water quality is also important outside our pilot site, as we described in the discussion section.

**Specific comments**

**1. Figure 1 contains too many features. I cannot read the map of Amsterdam, and the black item above. It seems redundant. I like the figure with the cross sectional view, but some features are unclear and should be redrawn: (1) where does the water from the drain system go to ? does the drain system capture groundwater or seepage water? What does the green area in the figure represent ? Is seepage vertically oriented towards the bottom of the ditch ? I would expect the ditch captures water from the surroundings.**

Agreed, see comment 69 of reviewer 1. We provided a new Figure 1 in the attachment.

**2. Figure 4 is too small. What does 1, 2, 3, 4 represent ?**

Following the suggestion of reviewer 1, comment 72, we updated the figure for better visualization uploaded in the system.

The Caption will now read: "*Selected precipitation events 1, 2, 3 and 4 showing dilution and peaks of water quality parameters, with hourly precipitation (mm/h) and hourly pumping activity (m/h). Note that different scales of TP and turbidity were used to reveal the dynamics*"

**3. Figure 6 could rainfall and/or EC be added here ?**

We considered this, but have presented those graphics in a previous paper already (Yu et al., 2019). We have added $NO_3$, TN and organic-N/TN ratios to this figure to better explain the N dynamics. The figure would be overloaded adding extra parameters. As is visible from the graphic below, plotting the EC would not do a lot to better explain the dynamics, as the complete EC continuous times series is already available in the manuscript.

[Figure]

**4. P7L245 a discrete water sample confirmed the low NH4: only one sample. This seems poor to serve as a confirmation. Did you perform regular grab sampling as to check the online values ? How is the data quality of the online measurements validated ?**

We agree that one single measurement is not convincing, but we mentioned it to suggest that this may reveal a similar pattern in 2016 when the high-frequency measurements had not started. We further clarified the text to reflect this:

*"A similar pattern of dilution and recovery is also visible for NH₄, especially for the period August 2016 – March 2017, where NH₄ shows a very similar response as EC, although with somewhat larger day to day fluctuations. However, a contrasting pattern without NH₄ recovery occurred twice: from the middle of June to the end of August 2016 and from the middle of March to the middle of May 2017. During these periods, concentrations of NH4 were considerably lower and deviated from the slope of the EC pattern. NH₄ decreased from around 4 mg L⁻¹ to around 2 mg L⁻¹ between the middle of June to the end of August 2016, but the continuous NH4 measurements are not supported by the grab samples which follow the EC pattern more closely. During the second period from March to the middle of May the deviation from the recovery curved is more pronounced, and NH₄ concentrations dropped to almost 0 mg L-1 and started recovery from the beginning of May. This pattern is fully supported by the available grab samples. During the same period in 2016 the high-frequency monitoring had not yet started, a single NH₄ grab measurement is available for the 2nd of May, that seems to reveal a similar pattern in the spring of 2016".*

Moreover, we discuss grab sampling results over the long-term dataset of 2007-2018 in Section 4.2, Figure 6, which we use to discuss the results of NH₄ and the other N species. That discussion confirms the value of the single measurement in May 2016 and the NH₄ high-frequency pattern for March-May 2017.

**5. P7L270 predicted and observed NH4 concentration generally agree. I would rather say that for both NH4 and P the concentrations are overestimated by the model. This makes sense since you don't take transformation or sink processes into account in the model.**

Based on the comments of both reviewer 1 and 2 we changed our line of reasoning into:

*"The dynamics of measured NH₄ concentrations show resemblance to the model results, especially during the wet season. Clearly, NH₄ is diluted during the rain events and a gradual increase of NH₄ starts after each rain even during the wet season showing slopes that resemble the model reconstruction. Over the whole period, measured NH₄ concentrations are overestimated by the model, indicating that some NH₄ is probably lost due to non-conservative processes. This is especially true for the spring season of 2017, where NH₄ concentrations must be controlled by other processes. Concentrations of TP are generally far below the conservative model reconstruction, except between the end of November and the beginning of March. During this particular period the minimum measured TP concentrations are captured nicely by the conservative model, however distinct peaks up to 3 mg L⁻¹ are not captured by the model and must have different physical or chemical processes determining them".*

In the discussion part of the paper, we relate this to the mobilization of P that was once sorbed and fixated in the sediments during the dry season.

**6. P9L344 the residence time is mentioned here. It would make sense to have the numbers for the residence time of the water in the manuscript. Did you calculate them for the different time periods ?**

We calculated the residence time in four seasons which is surface water volume divided by flux (using daily average pumping flux in our case). It shows that the residence time in spring is 11.4 days, summer is 10.2 days, autumn is 10.2 days and in winter is 9.6 days which is the shortest.

**7. P10L400 did you measure NO3 in this study? do you have clear evidence for NO3 consuming processes ?**

This was dealt with under replies to comments 2 and 9 of reviewer 1.

**8. P13L516-517 you state that the reactivity of the streambed sediments largely controls the water quality. Can you be more concrete on : (1) how then exactly the management should take care of this and (2) what type of measurements need to be done in the sediment to better understand the mechanisms in this system and to prove the hypothesis you make in Figure 7 ? You infer the mechanisms in Figure 7 based on surface water (water column) data only, and conclusions may need further elucidation.**

Agreed. As all the reviewers commented on the lack of management strategies, we decided to add an brief section in the discussion part of the paper to discuss management and monitoring implications.

**(1) how then exactly the management should take care of this**

*"4.6 Implications for urban water management in low lying catchments*

*This study demonstrated high frequency monitoring technology to be an effective tool for understanding the complex water quality dynamics. Investment in high frequency monitoring would greatly benefit the management of urban lowlands with substantial groundwater seepage by elucidating the determining biogeochemical processes and nutrient temporal patterns for realizing efficient mitigation and control of eutrophication. For example, a direct treatment of the drain water applying constructed wet lands could be considered as a mitigation measure in low lying areas with artificial water systems that resemble the Amsterdam region, e.g. in cities such as New Orleans, Shanghai and Dhaka. Centralizing the treatment of discharge water is also recommended, for instance by harvesting N as phytoplankton from the discharge during spring, or filtrating P at the pumping station during winter. Measures that artificially increase oxygen concentrations in the waters, such as the inlet of oxygen rich water, aeration by fountains or the artificial introduction of grazers or macrophytes may be considered to improve the ecological status of these urban waters. Moreover, aeration of the water in summer and autumn would possibly enhance processes such as nitrification and anammox, eventually converting $NH_4$ to $N_2$, before the water is discharged to downstream waters. Importantly, before the application of any measures or maintenance in urban low lying catchments, managers should evaluate the potential effects on the biological and chemical resilience of the ecosystem communities, e.g. dredging of a layer with abundant benthic activity might destroy an important buffer to nutrients in growing seasons, especially P."*

**(2) what type of measurements need to be done in the sediment to better understand the mechanisms in this system and to prove the hypothesis you make in Figure 7 ?**

We agree with the reviewer that some important hypotheses we drew in this paper need to be proved in future studies. The necessary researches need to be done are elucidated as below (added to the new section 4.6):

*"In this study, we concentrated on the analysis of the temporal patterns of water composition and on the deduction of the potential biogeochemical processes. Detailed studies about these processes and the biotic communities at the sediment-water interface were outside the scope of this paper. But testing the hypotheses with a comprehensive study on the sediment-water interface would be required to further increase our knowledge on the role of the benthic zone in attenuating N and P seeping up from groundwater. Besides, further research would need to consider the optimal physical dimensions of water courses and drain configurations, as to benefit the ecological status of urban waters that are prone to nutrient-rich groundwater seepage."*

**Reply to reviewer 3**

**Review of hess-2020_34 'Drivers of nitrogen and phosphorus dynamics in a groundwater-fed urban catchment revealed by high frequency monitoring' by Yu et al. (2020)**

**Context – Goals**

**This paper presents a 12-month investigation of variations in nutrients and various water quality indicators at the Geuzenveld polder in Amsterdam through, mainly, in situ monitorings of the waters recovered close to a pumping station (allowing to regulate the water levels in the polder). The authors investigated variations among the datasets over 12 months, and tested the incidence of rain events, and pumping events. They've applied a mixing model to explain some of the observed variations. Several parts of this paper are highly speculative. Correlations between data, and several other issues, would need to be supported by statistical tests. There are no M&M sections on the statistical tests performed; except a presentation of the mixing model. The raw datasets and scripts should be presented in the suppl. materials. Conclusions are not always supported by the presented datasets, and are sometimes highly speculative. They need to be supported by other studies which are not always cited and explained. This paper will require major improvements to meet the scientific quality of HESS papers. Several parts of this paper will need to be re-written and re-considered after a presentation of the statistical tests.**

We thank reviewer 3 for the time and effort put into reviewing our paper. The main point the reviewer 3 makes is to add more evidences to the patterns that we described qualitatively, using statistical summaries and tests. Our paper aims to improve understanding of the processes in a lowland urban water system which is fed by groundwater, and we have published previous papers about the regional patterns and the local water system, which included statistical approaches , including correlation analysis, linear regression and PCA analysis (Yu et al. 2018, Yu et al. 2019). For the present paper, we choose to focus on the temporal patterns and dynamics, using an end-member mixing model to infer the hydrological and biogeochemical processes, and developing hypotheses and suggestions for the main processes. The merits of the study are the prolonged and detailed, unique dataset that is provided, which allows for an understanding of processes at time scales that were never achieved in a similar urban catchment system. This approach was well acknowledged by reviewers 1 and 2 and their comments greatly helped to improve the reporting about the main results. We are not sure that statistical testing is the best approach in dealing with a complex dataset with high-frequency data, but evaluating the comments of reviewer 3, we choose to provide statistical inferences of all major graphs that we present and discuss in the paper. This way, the main message of the paper could further be strengthened. We choose to present the statistical outcomes in the Supplementary information, but mention the statistical summaries, including tests for correlation and correlation coefficients in the text, when we made a statement about our data. We believe this sharpens our results and discussion, for which we acknowledge and thank reviewer 3. Although we could not confirm all hypothesis that we posed in the paper, for example due to a lack of detailed ecosystems measurements of aquatic and benthic communities, we believe that the study helps to further study similar systems, eventually unraveling all processes and adding further experimental proof.

**Major comments**

**1. L86, Please specify the possible "management strategies" and add references**

Agreed. We inserted a section management strategies and added some references.

*"A deep understanding of the hydrobiogeochemical processes that control water quality dynamic would be a great asset for controlling eutrophication and improving aquatic ecological status (Fletcher et al., 2015[7]; Díaz et al., 2016[8]; Eggimann et al., 2017[9]; Nizzoli et al., 2020[10])."*

**2. Fig. 1; picture is too dark and its resolution is too low; please position the temporary floating platform used for the monitorings on this fig. The Drain 3 sampling point should be indicated; Yu et al., 2019 should be cited in the legend**

We provided a new Figure 1 as requested by all reviewers.

**3. High and low frequency monitorings: How did you compute the confidence intervals / error bars on the monitored values? please clarify these issues.**

We did not compute confidence intervals. Scatter plots of grab samples against high-frequency measurements are now provided to give an impression of measurement uncertainty. The correlations coefficients ($R^2$, "Pearson" method used) between the high frequency data and the routine discrete sampling data from Waternet are 0.88 for EC ($p$-value < 0.05), 0.92 for $NH_4$ ($p$-value < 0.05), and 0.97 for TP ($p$-value < 0.05).

[Figure]

**4. low frequency monitorings - L153 - "monitored at the pumping station" ; please clarify; where were these samples or values collected?**

Figure 1 gives now more details about the monitoring location.

[7]Fletcher T.D., Shuster W., Hunt W.F., Ashley R., Butler D., Arther S., Trowsdale S., Barraud S., Semadeni-Davies A., Bertrand-Krajewski J.L., Mikkelsen P.S., Rivard G., Uhl M., Dagenais D., and Viklander M.. SUDS, LID, BMPs, WSUD and more – The evolution and application of terminology surrounding urban drainage. Urban Water Journal, 12(7): 525-542, 2015.

[8]Díaz P., Stanek P., Frantzeskaki N., and Yeh D.H.. Shifting paradigms, changing waters: Transitioning to integrated urban water management in the coastal city of Dunedin, USA. Sustainable Cities and Society. 26: 555-567, 2016.

[9]Eggimann S., Mutzner L., Wani O., Schneider M.Y., Spuhler D., de Vitry M.M., Beutler P., and Maurer M.. The Potential of Knowing More: A Review of Data-Driven Urban Water Management. Environmental Science & Technology, 51: 2538-2553, 2017.

[10]Nizzoli D., Welsh D.T., and Viaroli P.. Denitrification and benthic metabolism in lowland pit lakes: The role of trophic conditions. Science of the Total Environment, 703: 134804, 2020.

**5. There is no section in the materials and methods regarding the statistical analyses of the datasets? please describe the statistical tests that were performed; which statistical packages were used? What was your experimental design regarding these tests?**

In the methods section, we now describe the correlation analysis (method is "Pearson") that was performed for all time series graphs provided in the paper. In order to confirm the qualitative statements about the temporal patterns that we describe at the 3 time scales, we choose 4-days averages for the seasonal time scale and the precipitation event time scale, and hourly for the pumping event time scale. These time intervals comply with the time scales over which the results were described and the interpretation was made. All the correlation coefficient tables were included in the the Supplementary information which is uploaded as well in the HESS system.

**6. Fig. 2: please indicate in the legend that nutrients and other quality indicators were monitored at the pumping station; variations in the presented datasets should be supported by statistical tests; correlation tests between monitored values should be performed. All raw datasets should be presented in the suppl. Materials**

We have indicated in Figure 1 where the monitoring location is situated. Correlation tables are now available for all graphics presented in the paper in the Supplement. All raw data will be made available through the Data Repository of the VU-university which will be accessible once the paper is accepted for publication.

**7. L219: "The wet season is distinguished by a higher frequency of pumping and lower water temperatures" ; please do statistical tests to validate these conclusions; when you indicate frequency, do you mean "volume"? please clarify**

We performed a statistical analysis to distinguish the wet and dry season, as below:

*"The wet season is distinguished by a higher average daily pumping volumes and lower water temperatures (Fig.2B) than the ones of the rest of the year (wet season: 997 m³/d, dry season: 787 m³/d)"*

**8. L221-222: "Especially in January and February 2017, there was a considerable period...": please define a "considerable period" by using statistical tests; was that specific of that year?**

Agreed. **Changed** "there was a considerable period that the water temperature was below 3 °C." **into** "during which the water temperature dropped to below 3 °C.". We don't think that all aspects in the paper need to be validated statistically; this information is easily extracted from the graphs itself.

It is not specific of the year. It is normal that water temperature goes below 3 °C in winter in Geuzenveld, see figure below (monthly average temperature from 2006 to 2018 measured by Waternet).

[Figure]

However, these are monthly measurement. The water temperature is below 3°C more often. The air temperature measured at a meteo station near the study area is shown as below.

[Figure]

**9. After the pumps stopped, the surface water level recovered faster during the wet season (between October 2016 and 225 March 2017) than during the dry season? Comment: where is this shown? Please clarify**

This is shown in Figure 2B as indicated.

**10. L226-227; L229; section 3.1.2: conclusions should be supported by statistical tests and should consider "confidence intervals" of the monitoring tools**

We now provided correlation coefficient tables in the Supplement that quantify the evaluation in the text. Where appropriate, the correlation coefficients and p-values ($H_0$ hypothesis: there is no correlation) are now given at the evaluation in the text.

**11. Please show the position of the piezometer on Fig. 1; its GPS coordinates should be indicated in the materials&methods section**

The position is indicated in Figure 1 and the GPS coordinates are added to the Data Repository.

**12. L230; please cite a document for the "water level regulation of the boezem Haarlemsmeer"**

We referred to: https://www.rijnland.net/actueel/water-en-weer/waterpeil

**13. L254: "significantly increased" : which test was performed?**

There was no test performed. We changed the wording "significantly" into "substantially". It now reads:

*"Before the middle of November 2016 and after March 2017, TP fluctuated around 0.5 mg L$^{-1}$, but always below 1 mg L$^{-1}$. TP concentrations substantially increased starting from the middle of November as did the variation over the day (Figure 2). "*

The Figure below illustrates the increase in variations over the day. We do not think any further statistical testing is necessary to confirm this result. For Figure S4, we calculated the difference between daily values (an average of

[Figure]

the hourly data) in the time series of TP, the differences are shown in the figure below (added to the supplementary information):

**Figure S4 Difference between daily monitoring values (an average of the hourly data) in the time series of TP**

The changes of the daily average concentration of TP are more substantial (the differences are in the range of -0.49 ~ 0.73 mg/L) from the mid November, 2016 to March, 2017 than the rest of the year (the differences are in the range of  -0.2 ~ 0.21 mg/L).

**14. L259-263: correlation tests should be done between total Fe values and turbidity**

Correlation test between total Fe and turbidity is 0.74, $p = 0.0003$. The scatter plot is shown as below.
Line 261, added "and $R^2 = 0.74$, $p < 0.001$" after "(Fig.2D"

[Figure]

**15. L277: there are no red blocks in Fig.1; neither in Fig. 2???**

Agreed. This is a typing mistake.
Line 276-277, **replaced** *"Fig.1 (red blocks)"* **by** *"Fig.2 (4 pink shades)"*.
Line 680, **replaced** *"red blocks"* **by** *"The 4 pink shades"*

**16. Section 3.3.1: datasets should be supported by statistical tests and should consider the confidence intervals of the monitored values.**

We now provide correlation tables in the Supplement that quantify the evaluation in the text. Where appropriate, the correlation coefficients and $p$-values ($H_0$ hypothesis: there is no correlation) are now given at the evaluation in the text.

**17. L301-304: to be moved in the discussion**

Agreed.
Line 303, **added** "and turbidity" **after** "Peaks in P".
Line 304, **added** "2014 & " **after** "Van der Grift et al.,"
Line 301-304, **moved** *"In artificial lowland catchments, water systems are intensively regulated by pumping activity to prevent flood and drought. However, there is a substantial lack of knowledge about the possible consequences of such regulation on aquatic ecology and water quality. Peaks in P and turbidity by the activation of pumps was observed by Van der Grift in his high frequency monitoring campaign in an agriculture lowland polder (Van der Grift et al., 2014 & 2016)."* **to line 466.**
Deleted line 466 *"Van der Grift et al. (2014) studied agricultural areas and observed that P and turbidity were significantly increased by pumping events."*

**18. Section 3.3.2: add statistical tests e. g. L309-310, correlation test between values, validate the seasonal effect, etc**

We now provide correlation tables in the Supplement that quantify the evaluation of the seasonal effect in the text. Correlation coefficients and $p$-values ($H_0$ hypothesis: there is no correlation) are now given at the evaluation in the text.

**19. L310: "during events 2, 3 and 4, TP and EC are positively correlated"; which test? Please give the p-values, etc**

We now provide correlation tables for Figure 4 in the Supplement (Table 7-10) that quantify the evaluation in the text. We provided the correlation coefficients and $p$-values ($H_0$ hypothesis: there is no correlation) for TP and EC at this time scale.

**20. Figs. 4; please clarify the legend and relation with Fig. 2. Add statistical tests to define which values are correlated, etc.**

We now provide correlation tables for all parameters in Figure 2 in the Supplement (Table 1-3), quantifying the evaluation in the text. Where appropriate, the correlation coefficients and $p$-values ($H_0$ hypothesis: there is no correlation) are now given at the evaluation in the text.

**21. Please clarify what is correlated or not according to pumping**

See comment 20.

**22. Line 332 - " This presumption is supported by the mixing model result of EC, which revealed close similarity to the measurements".: comment - How did you test this similarity between monitored and modeled values? please give the details of this performance analysis in the result section**

We now provide correlation tables for all parameters in Figure 3 in the Supplement (Table 4-6), quantifying the evaluation in the text. Where appropriate, the correlation coefficients and $p$-values ($H_0$ hypothesis: there is no correlation) are now given at the evaluation in the text.

**23. Too many citations of the figs in the discussion; several issues should be transferred in the result section**

Here reviewer 3 deviates form reviewers 1 and 2. We choose to keep the current division of figures and text for the results and discussion sections.

**24. L333: not clear**

We rephrased this paragraph according to comment 60 by reviewer 1.

**25. No cited literature in section 4.1 of the discussion ?**

We now refer to Yu et al., 2019 and Walsh et al., 2005.

**26. Fig. 6 should be presented in the results; the raw datasets should be presented in the suppl. materials. Any information on the confidence intervals for these datasets?**

We choose to present this Figure in the Discussion part of the paper, as it relates to another dataset over a much longer period, which is used to check our hypotheses and give further proof as would be expected in a Discussion section. Our paper focuses on the new results of the 2016-2017 high-frequency time series and the intensified grab sampling during that campaign. We will provide these raw data of Figure 6 in the Data Repository.

**27. L362-364; 380-382, etc: please add statistical tests to support these conclusions**

We indeed now provide correlation coefficient tables for all parameters in Figure 6 in the Supplement (15-18), quantifying the evaluation in the text. Where appropriate, the correlation coefficients and $p$-values ($H_0$ hypothesis: there is no correlation) are now given at the evaluation in the text.

**28. L365; 398: qPCR datasets to estimate the population levels of some of the organisms involved in NH4 assimilation, etc, would be interesting to support some of the conclusions reached in this paper**

We agree with the reviewer but this is far beyond the scope of the present paper. It is definitely one of the ways forward, testing, validating or falsifying hypotheses made in this paper.

**29. L401; Fig. 6 is indicated in relation with NO3 datasets; where are these data?**

See reply to comments 2 and 9 of reviewer 1. $NO_3$ is now integrally part of Figure 6 and the discussion.

Following the suggestion of reviewer 1 we deleted the lines: *"Apart from primary production, $NH_4$ can also be consumed through nitrification, i.e. oxidation of $NH_4$ to $NO_3$ by microbes (Zhou et al., 2015). The produced $NO_3$ can be taken up by primary producers and by microbes reducing it to dinitrogen gas (denitrification and anammox ($NO_3 + NH_4 \rightarrow N_2 + H_2O$);*

*Thamdrup and Dalsgaard, 2002; Kuenen, 2008). These $NO_3$ consuming processes were very active as $NO_3$ concentration were sometimes high (e.g. 50 mg $L^{-1}$) in street runoff samples (Yu et al., 2019), but low in surface waters (Fig. 6). "*

**30. L417 "...significantly..": please support this by a statistical test**

We changed "significantly" into "importantly". The pattern is clear and does not need further testing.

**31. Scripts and data should be made available in the suppl. Materials**

We prepared all our data to make it available on VU-university repository once the paper is accepted for publication. It includes the complete raw dataset, data processing and analysis scripts, as well as the data visualization scripts.

**32. Section 4.4: speculative section on the importance of biotic processes (benthic algae, bacteria / nitrification-denitrification-anammox) in the variations observed in Fig. 2 and Fig. 6-> there are very few datasets on these issues (only chlorophyll a monitorings); it would have been relevant to add qPCR (variations in total bacterial numbers, cyanobacteria, denitrifiers, etc) assays to validate the inferences made in Fig. 7. This section need to be supported by more citations of the literature on similar issues.**

We consider this outside the scope of our paper. The paper poses hypotheses based on the water quality campaign and addresses where abiotic, hydrological processes can no longer explain the nutrients behavior. Some of the hypotheses, especially the ones about biotic processes definitely need further elaboration and field experiments. Still, we think that the presented water quality parameters and time series allow us to make inferences about these processes, citing our previous papers and relevant literature.

We added the following sentences to the new section 4.6 to make this point more clear in the text:

*"In this study, we concentrated on the analysis of the temporal patterns of water composition and on the deduction of the potential biogeochemical processes. Detailed studies about these processes and the biotic communities at the sediment-water interface were outside the scope of this paper. But testing the hypotheses with a comprehensive study on the sediment-water interface would be required to further increase our knowledge on the role of the benthic zone in attenuating N and P seeping up from groundwater. Besides, further research would need to consider the optimal physical dimensions of water courses and drain configurations, as to benefit the ecological status of urban waters that are prone to nutrient-rich groundwater seepage."*

**33. L444-445: "... Phytoplankton biomass decreased because of competition for N or grazing activity...";; this is speculative – there are no data on grazing? At least add a reference on this issue to support this possibility**

We changed the text to emphasize that Figure 7 is meant to summarize our hypotheses, which are not proven, but suggested based on the available data.

"*Figure 7 shows a conceptual diagram for the N and P dynamics in this lowland urban catchment during the four seasons which summarizes our hypotheses about the functioning of the system*"

**34. 445-446: not supported by the presented data**

See reply to comment 33

**35. Fig. 7; please cite in the legend all papers which made possible most of these inferences, and indicate which data presented in this paper added support for the presented scenarios**

We prefer to mention the papers in the main text as we did.

**36. L447- ".. relatively low in oxygen (because of warming)"; please add data which support this effect of warming on oxygen levels (or a citation)**

Agreed. We performed a correlation analysis of the parameters. Oxygen has negative correlation (p <0.05, "pearson") with water temperature in a statistical test. However, there might be more reasons contributed to the low oxygen level, such as oxygen consuming process such as denitrification and organic matter (dead algae) decomposition (references). We adapted the text to read: *"and relatively low in oxygen due to the continuous supply of anoxic groundwater, the mere absence of $O_2$ -rich runoff, the oxidation process of Fe(II) and possibly by microbial organic matter decomposition during warm periods with relatively stagnant water*)".

**37. L448 – "... Biological activity declined (colder and less light)...": there are no data on these issues; following sentence is also an inference from the literature and not from the presented data**

Primary production is determined by sunshine and temperature (reference). The trend of the temperature is shown in Figure 2 in the paper. The trend of sunshine duration is shown as below, which is added to the supplementary information (Figure S5). The sunshine duration starts to decline from autumn. The lowest period is from November to February.

The following sentence *"Moreover, the redox zone moved from the sediment-water interface into the water column"* is indeed an inference from literature. However, it is the common situation in the Netherlands. We now refer to Van der Grift et al. 2014 & 2016 (see the original manuscript).

[Figure]

**Figure S5 Sunshine duration (hours per day in each month)**

**38. L463 – "this accelerates the further aggregation of the iron complexes..." ; this is speculative / not based on the presented data; add arguments (a citation) to support this conclusion or delete**

We adapted the wording to emphasize our inference:

*"Yu et al. (2019) showed that precipitation runoff delivers particles and $O_2$ to the ditches. We suggest that this accelerates the further aggregation of the iron complexes; the resulting larger particles more readily settle to the bottom, causing a reduction of turbidity during events (Fig. 4)."*

**39. L464 – "...The resulting larger particles more readily settle to the bottom..."; comment : no data on this issue; add a citation to support this conclusion or delete**

See reply to comment 38

**40. L473 +– "... the water was highly turbid because of the formation of iron hydroxide colloids in the water column.."; "..The activation of the pumps caused export of these colloids and particles and thus reduced turbidity ..."– comment : no data on this issue / have you monitored "particle sizes"? please add these datasets; or add a citation to support this conclusion or delete**

We rephrased the text as our original statement was too bold: *"We explain the reduced turbidity after a precipitation event as a result of the activation of the pumps which caused the export of the turbid water towards the receiving boezem in combination with aggregation of ironhydroxides in the water column and subsequent settling of the aggregates due to the supply of new $O_2$-rich water (Fig.5 event 2, see also Van der Grift 2014)".*

**41. Fig. 8 should be presented in the M&M and result sections. Please show on Fig. 1 where these monitorings were performed**

The monitoring was performed at the monitoring location as depicted in Figure 1. This was indicated in the Figure caption. We choose to present this in the discussion part of the paper as it deals with the consequences of the work, translating it into fluxes and loads which are relevant for water management, which is dealt with in the subsequent section 4.6.

**42. Only fig. 7 should be cited in the discussion; other figs should not; all data presentation issues should be moved from the discussion into the result section**

We choose to keep the original structure, as we think it improves the readability of the work and was well received by reviewers 1 and 2.

**43. Discussion is too long; please simplify but avoid over-interpretation of the datasets (conclusions should be strictly based on statistically well supported trends)**

The objective of our paper is not only to present our measured datasets and derive statistically underpinned conclusions. As our datasets are complex by nature, with many interactions and feedbacks between the measured and unmeasured parameters, we also set out to form new hypotheses about the driving processes that can explain the observed abiotic water quality behavior. Further research indeed needs to confirm or falsify these hypotheses. We believe that reviewer 3 underestimates the value of our hypotheses in such complex "natural" systems outside a laboratory setting where we cannot control all aspects. For example, based on this research we are now keen on initiating a follow up research where we will set out to measure the redox-profile in the ditch sediment and how this is affected by bentic algae, as this became an important hypothesis to explain observer P and $NH_4$. We therefore believe, that the ideas and hypotheses about the driving mechanisms in such a complex dynamic system are of even more value (to us) than the statistically significant relations between measured parameters. We did perform the test and correlation analysis where we could, which indeed helped to strengthen our inferences from the visual inspection of the temporal patterns. Though we believe that the length and structure of the discussion is in correspondence with the data that we provide and for which we pose the hypothesis of the functioning of the water system.

**44. L499 "..Iron redox chemistry was the dominant process controlling the P dynamics in shallow groundwater fed ditches"; comment: dominant over which other processes??? Please clarify and give arguments / which data demonstrate clearly this relation? Datasets present total Fe values and total-P; have you done correlation tests?**

We **replaced** "dominant" **with** "determining". We provided the correlation tests which confirmed the relation between Fe and P (Supplementary information Table 1, $R^2 = 0.65$, $p = 0.002$). Moreover, the Fe and P data from grab sampling were presented already in Yu et al. 2019 and further evidence stems from Van der Grift et al. 2014&2018 which we cited in the respective sections. We did not want to include references in the conclusion section.

**45. L503 – L508 "..mostly in the form of iron hydroxides": comment – did not see any datasets on this issue? Please limit your conclusions to the points that were investigated in the paper**

We also use the work on the same catchment that was referred to extensively in the paper (Yu et al. 2019). We did not want to include references in the conclusion section.

**46. L510 – "...by intensifying iron oxidation and precipitation..." : comment – did not see any datasets on this issue? Please limit your conclusions to the points that were** investigated **in the paper**

See comment 45.

**Minor comments**

**1. In the introduction, L84: ".. to understand the mechanisms that control the dynamics of N and P in urban delta catchments.."; please clarify by changing "mechanisms" by " the hydrobiogeochemical processes that control..."**

Agreed, see Major comment 1 (Reviewer 3).

**2. L115: "During rainfall events, the surface water level will rise faster"; please be more accurate or add a citation on these issues.**

Agreed.
changed " will rise" into "rises"
added "(Fig.2A)" after "faster"

**3. L17, please put the month before the year**

Agreed.
moved "March" before "2016"
moved "June" before "2017"

**4. Doi numbers have not been indicated for the cited papers**

Our experience is that this will be done automatically by HESS.

**5. L43; why "pivotal"? Explain**

"pivotal" means "of crucial importance in relation to the development or success of something else". And researches (e.g. Nyenje, et al., 2010; Toor et al., Paerl et al., 2016; 2017; Le Moal et al., 2019) have reported the necessity of developing the understanding on nutrients dynamics for eutrophication alleviation.

**6. L49; please use another term than "preferred" ; most effectively uses NH4 for protein synthesis**

Not agreed.

**7. Several sentences are too long; please simply at least the following sentences e. g. L36-39; L62-65; 160-163; 356-359**

Agreed.
Line 37: **added** "*. The identified sources of nutrients are* " **after** "*P*"
Line 37: deleted "*in cities with combined drainage systems*"
Line 64: **replaced** "2019)," **by** "*. The retained P are* "
Line 160-163: **replaced** "*To release all Fe that may have sorbed or precipitated during storage, we added 1 or 0.5 ml HCl in the water samples to dissolve eventual flocks, homogenized the samples in an ultrasonic bath for 24h, shook again to break down all the flocks, sampled 10 mL of the water with pipet into a Teflon bottle, added 3.2 mL HCl : HNO 3 3:1 for extraction, and subsequently put them in a stove at 90 °C for 24 hours.*" **by** "*To release all Fe that may have sorbed or precipitated during storage, we added 1 or 0.5 ml HCl in the water samples to dissolve eventual flocks. Then the samples were homogenized in an ultrasonic bath for 24h, mixed again to break down all the flocks. For extraction of all the Fe, transferred 10 mL of the homogenized sample into a Teflon bottle, added 3.2 mL HCl : HNO$_3$ 3:1 , and stored in a stove at 90 °C for 24 hours.*"
Line 356-359: **replaced** "*While NH$_4$ dynamics during winter can be explained by mixing, this is not the case during spring and summer because biological processes are then overruling physical mixing. This resulted in much lower measured NH$_4$ concentrations than calculated by our conservative mixing model during the growing season, benthic and planktonic primary producers (e.g. phytoplankton) assimilate nutrients and are an important factor controlling nutrient dynamics in rivers, lakes, streams (Hansson, 1988; Jäger et al., 2017).*" **by** "*NH$_4$ dynamics during winter can be explained by mixing. However, during spring and summer  biological processes are then overruling the mixing process. It resulted in lower measured NH$_4$ concentrations than the modeled during this period. Studies have shown that benthic and planktonic primary producers (e.g. phytoplankton) assimilate nutrients and are an important factor controlling nutrient dynamics in rivers, lakes, streams (Hansson, 1988; Jäger et al., 2017).*"

**8. L74: "In recently years » ; to be changed**

Agreed, **changed** "*In recently years*" **into** "*In the past few years*".

**9. L75, Please define "high frequency technology"**

Agreed and done.
Line 75, added "*the development of new sensors and sampling technologies allow us to get data with substantially shorter intervals. In this paper, the high frequency monitoring technology is referred as automatic monitoring program with sampling and analyzing frequencies that are sufficient for obtaining detail water quality variation information. *" after "*In the past few years*".
**Changed** "*, high frequency*" **into** "*High frequency*".

**10. L89-90: last part of this sentence is not needed i. e. "... unraveling the hydrological and the reactive biogeochemical processes that control the nutrient 89 dynamics at these 3 time scales"**

Agreed. Deleted "*, unraveling the hydrological and the reactive biogeochemical processes that control the nutrient dynamics at these 3 time scales*".

**11. L118; replace pump by "pumping"**

Agreed. **Replaced** *"pump"* **by** *"pumping"*.

**12. L135: "calibrated" instead of calibrating**

Agreed. **Replaced** *"calibrating"* **by** *"calibrated"*.

**13. L146: "times"; "lightening -> lightning,**

Agreed. **Replaced** *"time"* **by** *"times"*, **replaced** *"lightening"* **by** *"lightning"*.

**14. L153-154; 164: not clear; to be re-worded**

Line 153-154: *"Since 2006, Waternet has monitored the water quality with a frequency of 12 times per year by sampling at the pumping station of Geuzenveld. Between 2016 and 2017, the sampling frequency was increased to twice per month."*

Line 164: Clarified as: *"Then the samples were homogenized in an ultrasonic bath for 24h, mixed again to break down all the flocks. For extraction of all the Fe, we transferred 10 mL of the homogenized sample into a Teflon bottle, added 3.2 mL HCl : HNO₃ 3:1 , and stored in a stove at 90 °C for 24 hours. The final solutions were analyzed by ICP-AES. Blanks were included and treated identical to samples."*

**15. L161: shook -> mixed**

Agreed. **Replaced** *"shook"* **by** *"mixed"*.

**16. L169: inlets -> inputs**

Agreed. **Replaced** *"inlets"* **by** *"inputs"*

**17. L170: outlets -> outputs**

Agreed. **Replaced** *"outlets"* **by** *"outputs"*

**18. Fig S1; change valid for validated; please indicate color code in the legend; please perform and indicate the p value for the correlation test. Description of Fig. S1 in this suppl. material should be deleted.**

Agreed.
**Changed** *"valid"* into *"validated"*.
Replaced the figure by the one below:

[Figure]

**Added** *"(R² = 0.71, p-value < 0.05)"* **after** *"EC and Cl"*
Deleted *"In the study area, groundwater is the water resource with the highest Cl concentration, and contributes most of EC. Thus, road salt was presumably the contributor to the relatively higher EC from the continuous measurement during winter. Cl subsequently will be expected to be significantly elevated during winter as the same time of the rise of EC. However, neither significant rise of EC nor Cl was observed in the discrete sampling data as shown in the figure."*

**19. L189: please delete "sourced from groundwater"**

Agreed. Deleted *"sourced from groundwater"* in line 189.

**20. L191-192; to be moved in the results or discussion section**

Agreed. **Move line 191-192** *"By comparing the modeled EC, NH 4 -N and TP with high frequency measurements, potential processes that might deprive or enrich nutrients along the flow routes were inferred from the discrepancies between the modeled and the measured data."* **to line 266 after** *"...plotted in Figure 3."*
Deleted *"EC, NH₄-N and TP"*.

**21. L207-212: to be deleted**

Agreed. Deleted line 207-201.

**22. Please avoid citing figs in the discussion**

We did not think this is sensible, as it would reduce readability of the paper.

**23. Fig. 5; translate the "x" axis**

Agreed, changed accordingly.

**2. List of all relevant changes**

**Changes to the figures:**
We have remade Figure 1.
We have remade Figure 2.
We have remade Figure 3.
We have remade Figure 4.
We have remade Figure 5.
We have remade Figure 6.

**Changes to the manuscript:**

**L15 added** "(N)" **after** "nitrogen"**, added** "(P)" **after** "phosphorus"

**L17 moved** "March" **before** "2016" **and moved** "June" **before** "2017"

**L19 changed** "discusses" **into** "discussed"

**L20-21 reworded the preceding sentence into:**
"Mixing of upwelling groundwater (main source of N and P) and runoff from precipitation on pavements and roofs was the dominant hydrological process governing the temporal pattern of the EC, while N and P fluxes from the polder were also regulated by primary production and iron transformations."

**L22-25 replaced by:**
"In our groundwater-seepage controlled catchment, $NH_4$ appeared to be the dominant form of N with surface water concentrations in the range of 2-6 mg N/L, which stems from production in an organic-rich subsurface. The concentrations of $NH_4$ in the surface water were governed by the mixing process in autumn and winter and were reduced down to 0.1 mg N/L during the algae growing season in spring."

**L26 added** "concentrations of" **before** "chlorophyll-a"

**L26-28 replaced** "Total P…iron oxides." **by**
"Total P and turbidity were high during winter (range 0.5-2.5 mg P/L and 200-1800 FNU, respectively) due to the release of P and reduced iron from anoxic sediment to the water column, where $Fe^{2+}$ was rapidly oxidised and precipitated as iron oxides which contributed to turbidity. In the other seasons, P is retained in the sediment by sorption to precipitated iron oxides."

**L29 replaced** "downstream water bodies" **by** "receiving waters"**; replaced** "as" **by** "in the form of"

**L31 replaced** ", it is possible to formulate" **by** "we suggested"

**L33 changed** "can" **into** "may"**; deleted** "situation"

**L37 added** ". The identified sources of nutrients are " **after** "P"**; deleted** "in cities with combined drainage systems"

**L40 after "low-lying deltas" added:**
"where dissolved $NH_4$ and $PO_4$ in groundwater seep up into urban surface water"

**L41 replaced** 'end up' **by** 'reaching'

**L45 moved** "Nutrients dynamics are governed by biological, chemical, and physical processes and their interactions. " **to L47 before** "Assimilation…."

**L47 added** "the" **before** "aquatic environment"

**L49 added** "in some cases like in estuaries" **after** "by microbes"

**L53 changed** 'Under aerobic conditions, NH4 can be oxidized to NO3 through nitrification by nitrifying microbes even under cold conditions (below 10 °C ), which is an O2 consuming, acid generating process' **into:**
"Under aerobic conditions, NH4 can be oxidized to NO3 through nitrification by nitrifying microbes, which is an O2 consuming and acid generating process. Nitrification even occurs under cold conditions (below 10 °C)"

**L59 deleted** "the" **before** "dilution"; **added** "precipitation" **before** "events".

**L60 deleted** ", and chemical reactions such as mineral precipitation with associated P incorporation cause removal from water column (Rozemeijer et al., 2010a; Van der Grift et al., 2014; Yu et al., 2019)"

**L64 replaced** "2019)," **by** ". The retained P are "

**L66 changed** "environment" **into** "environments"

**L67 replaced** "Griffioen, 2006; van der Grift, 2014" **by** "Griffioen, 2006; Rozemeijer et al., 2010a; Van der Grift, 2014; Yu et al., 2019"

**L74 replaced** 'its' **by** 'their'; **changed** ", land use, etc" **into** "and land use"

**L74-75 replaced** "In recently years, high" **by**
"In the past few years, the development of new sensors and sampling technologies allow us to obtain data with substantially shorter intervals. In this paper, the high frequency monitoring technology is referred to as an automatic monitoring program with sampling and analyzing frequencies that are sufficient for obtaining detailed water quality variation information. High"

**L77 changed** "quantitively" **into** "quantitatively"

**L83 replaced** "...insight in..." **by** "...insight into..."; **added** "and fate in urban delta catchments affected by groundwater" **after** "transport"

**L84-86 replaced** "The goal of this…downstream waters." **by**
"A deep understanding of the water quality dynamic drivers would be a great asset for controlling eutrophication and improving aquatic ecological status (Fletcher et al., 2015; Díaz et al., 2016; Eggimann et al., 2017; Nizzoli et al., 2020)."

**L86-90 replaced** 'We conducted…3 time scales' **by** 'We conducted a one-year high frequency monitoring campaign in 2016-2017. Measured parameters were EC, NH4, TP, turbidity and water temperature. The temporal patterns of these parameters were studied at three time scales: the annual scale, rain event scale, and pumping event scale.

**L93 added:**
"The Geuzenveld study site is part of an urban lowland polder catchment, which is characterized by groundwater seepage that constantly determines the surface water quality, being the main source of solutes in the water system. The groundwater seepage is a continuous source of anoxic, iron and nutrient rich slightly brackish waters. Yu et al. 2019) presented the results of a 10 year monitoring program describing the main processes determining the water quality in the catchments, which isdominated by mixing of runoff water and seepage water. A high-frequency monitoring campaign was set-up to further unravel the temporal pattern on the nutrient N and P, of which N is typically present in the form of NH4 from groundwater."

**L94 changed** "neighbourhoods" **into** "neighborhoods"

**L97-103 replaced** "Because Geuzenveld…." **by**
"Geuzenveld is a groundwater fed catchment due to the constantly higher groundwater head (-2.5 ~ -3 m NAP) in the main aquifer relative to the surface water level in the polder ditches (~ -4.25 m NAP). (Fig.2). To keep the foundations of the building dry, there is a groundwater drainage system placed under an artificial sandy layer, right on top of a natural clay layer. The drain elevations range from -4.84 to -4.61 m NAP , which is below the phreatic groundwater level throughout the year, making sure that groundwater seepage either discharges through the drains or the ditches."

**L102 deleted** "(NAP: Normalized Amsterdam Peil, a known international standard conforming to mean sea level)"

**L115 changed** " will rise" **into** "rises"**, added** "(Fig.2A)" **after** "faster"

**L118 replaced** "pump" **by** "pumping".

**L124 at the end added** "The monitoring frequencies were set to 20 mins, 10 mins, 5 mins, 5 mins and 5 mins interval for TP, NH4-N, turbidity, EC and water temperature, respectively."

**L135: changed** 'was calibrating' **into** 'was calibrated'

**L146 replaced** "time" **by** "times", **replaced** "lightening" **by** "lightning".

**L151 replaced** "in" **by** "of"

**L153 added** "by sampling" **after** "per year"

**L153 added** "the water quality" **after** "Since 2006, Waternet has monitored"

**L154 changed** "became" **into** "was increased to"

**L154**
**deleted** "Many parameters were measured in this dataset, but for this research".
**added** "parameters from the routine monitoring campaign" **after** " We selected the following..."

**L156 added:**
"$NO_3$, TN, Kjeldahl-N"

**L157 added** "Organic-N was estimated by subtracting $NH_4$-N from Kjeldahl-N.**"; replaced** "chlorophyII" **by** "chlorophyll"

**L160-163 replaced** "To release all Fe … for 24 hours." **by** "To release all Fe that may have sorbed or precipitated during storage, we added 1 or 0.5 ml HCl in the water samples to dissolve eventual flocks. Then the samples were homogenized in an ultrasonic bath for 24h, mixed again to break down all the flocks. For extraction of all the Fe, transferred 10 mL of the homogenized sample into a Teflon bottle, added 3.2 mL HCl : $HNO_3$ 3:1 , and stored in a stove at 90 °C for 24 hours."

**L164 Clarified as** "During late autumn, we observed that the water was highly turbid (see also Yu et al. 2019) which we suggest to be caused by the formation of iron hydroxide colloids in the water column, which is supported by correlations between Fe-grab and Turbidity ($R^2$= 0.72, Table S2). We explain the reduced turbidity after a precipitation event as a result of the activation of the pumps which caused the export of the turbid water towards the receiving boezem in combination with aggregation of iron hydroxides in the water column and subsequent settling of the aggregates due to the supply of new $O_2$-rich water (Fig.5 event 2, see also Van der Grift, et al., 2014)."

**L161 replaced** "shook" **by** "mixed".

**L166-171 replaced by** "A correlation analysis between the high frequency and discrete monitoring data was applied to illustrate the reliability of the high frequency time series. Furthermore, the time series data were analysed at 3 time scales: annual scale, rainfall events (several days) and single pumping events (several hours). The relationships among the monitored parameters was explored by testing their correlations at each time scale. At the annual scale, a correlation analysis was applied to the complete time period and the wet and dry periods (definition in section 3.1.1). To discern the hydrological and chemical/biological attributes to the observed dynamics, a linear mixing model was introduced at the annual scale, assuming precipitation and groundwater seepage are the only water inputs, pumping and evapotranspiration are the only outputs, and pumping activity is the only way solutes leave the water system. In this model, we assumed a constant seepage rate. Accordingly, surface water level was calculated from:"

**L174-176 changed** "L is surface water … maximum capacity 216 m3h-1." **into** "V is total water volume in the ditches, P is precipitation, S is a constant seepage, E is potential evapotranspiration, Apolder is area of the polder, Pump(t) is water volume being pumped out with maximum capacity 216 m3 h-1, Aditch the area of the ditches in the polder. L is surface water level in the ditches. Water level L determines the activation of pumping activity. "

**L179 added** "Given the year-round seepage conditions throughout the polder, combined with an artificially drained subsurface, we assumed that the potential evapotranspiration was close to the actual evapotranspiration as no water shortages occur in our situation. In this study, we used the difference between groundwater head in the first aquifer and the surface water level (Figure 2A) to estimate a range of the seepage. The actual number of 2 mm per day was chosen based on the behavior of the mixing model and calibrated using the measured surface water levels (Figure S1)."

**L183-184 changed from** "C(t) is solute concentration…by equation (1)." **into** "V is the ditch water volume given by equation (1), C(t) is solute concentration at time t, Cgw is the average groundwater concentration, Cp is the average concentration in runoff."

**L183 at the beginning added** "V is the ditch water volume given by equation (1),"

**L185-187 changed** "Due to…2019)" **into** "In our study area, the EC is a useful water quality parameter for describing the mixing processes between groundwater and runoff water, as the EC represents the end members of the mixing: groundwater with an high EC (1750 µS/cm) and runoff water (100 µS/cm) with a low EC (see also Yu et al., 2019). Moreover, we assume that EC behaving as a conservative tracer as the EC is highly correlated with the Cl concentration ($R^2$ = 0.71, $p$-value < 0.05) and the temporal patterns of EC and Cl are very similar (see supplement Figure S2)."

**L188 replaced** "final chosen" **by** "calibrated"; **changed** "1.5" **into** "2"

**L189 deleted** "sourced from groundwater"

**L190 replaced** "above gave us" **by** "provided"

**L191-192 deleted** "By comparing the modeled EC, NH 4 -N and TP with high frequency measurements, potential processes that might deprive or enrich nutrients along the flow routes were inferred from the discrepancies between the modeled and the measured data."

**L193 changed** "their" **into** "the"; **added** "A comparison between the modeled and the measured results was performed by using correlation analysis." **after** "high frequency measured time series"

**L198 changed** "01-01-2016" **into** "06-2015"; **changed** "results were" **into** "model was"

**L199 deleted** "very"; **added** "the" **after** "to"

**L201 added** "((EC value reduced by over 35%))" **after** "dilution extent of EC".

**L205 added** "Correlation analysis was as well applied to each event at the corresponding two time scales, averaging over whole days for precipitation events and over hours for pumping events. Data processing and analyzing were performed using Rstudio (R version 4.0.2) and time series package "xts"."

**L207-212 deleted**

**L216 added** "light" **before** "blue"

**L217 changed** "pink color" **into** "dark blue"

**L217-219 replaced** "The wet season was from October … water temperatures (Fig.2B)." **by**
"We defined the wet and dry seasons based on water surplus and deficit. The average net rainfall (the water surplus/deficit in Figure 2) is 1.4 mm/d for the period of 01-10-2016~15-03-2017, and -0.8 mm/d for the rest. Subsequently, we statistically analysed the difference between these two periods for multiple parameters. Table 2

shows the mean of each parameter for the wet and dry seasons. The wet and dry seasons means are significant different for all parameters, but the EC.

**Table 2 The mean of each parameter, and the significance for the wet and dry seasons**

| | Net rainfall* mm/d | Pump volume* $m^3$/d | Water temperature* °C | EC µs/cm | $NH_4$* mg N/L | TP * mg P/L | Turbidity* FNU | Fe* mg/L | $O_2$* mg/L |
|---|---|---|---|---|---|---|---|---|---|
| Wet | 1.4 | 1050 | 6.7 | 1212 | 3.7 | 0.8 | 197 | 3.4 | 4.3 |
| dry | -0.8 | 712 | 17 | 1252 | 3.0 | 0.5 | 15 | 1.5 | 3.3 |

* $p < 0.05$
"

**L219 replaced** "Water temperature ranged from 2 to 26°C**.**" **by** "Over the whole monitoring period, the water temperature ranged between 2 to 26 °C.**"**

**L222 changed** '..that the water temperature was below...' **into** '...during which the water temperature dropped to....'

**L227 changed** "shallow**"** **into** "phreatic**"**

**L227 added** "(light blue)" **after** "Fig.2A".

**L228 added** "(Figure 1, 52°22'46.0"N 4°47'15.6"E**"** **after** "outside of the polder**"** ; **replaced** "water level ranges from surface water regulation regime, ..." **with** "surface water level (Fig.2A, dark blue), ...".

**L229-230 replaced** "might have been caused by**"** **by** "is related to**"; replaced** "Haarlemsmeer**"** **by** "Haarlemmerweg**"; replaced** ". Groundwater**"** **by** "(https://www.rijnland.net/actueel/water-en-weer/waterpeil). Phreatic water**"**

**L234 added** "The coefficients of determination ($R^2$ "Pearson" method used) between the high frequency data and the routine discrete sampling data from the water authority are 0.88 for EC (*p*-value < 0.05), 0.92 for $NH_4$ (*p*-value < 0.05), and 0.97 for TP (*p*-value < 0.05). The scatter plots between the high and low frequency measurements are shown in Figure S7."

**L234 replaced** "are" **by** "were"

**L235 changed** "feed" **into** "fed"

**L237 removed** '...if there was no rain'.

**L237 changed** "This duration of the return to pre-event EC values" **into** "The duration of this process".

**L238- 246 changed** "A similar pattern… time of the year" **into**
"A similar pattern of dilution and recovery is also visible for $NH_4$, especially for the period August 2016 – March 2017, where $NH_4$ shows a very similar response as EC (Table S2, wet season, $R^2 = 0.73$ ), although with somewhat larger day to day fluctuations. However, a contrasting pattern without $NH_4$ recovery occurred twice: from the middle of June to the end of August 2016 and from the middle of March to the middle of May 2017. During these periods, concentrations of $NH_4$ were considerably lower and deviated from the slope of the EC pattern. NH4 decreased from around 4 mg L-1 to around 2 mg L-1 between the middle of June to the end of August 2016, but the continuous NH4 measurements are not supported by the discrete samples which follow the EC pattern more closely. During the second period from March to the middle of May the deviation from the recovery pattern is more pronounced, and NH4 concentrations dropped to almost 0 mg L-1 and started recovering from the beginning of May. This pattern is fully supported by the available discrete samples. During the same period in 2016 the high-frequency monitoring had not yet started, a single NH4 discrete measurement is available for the 2nd of May, that seems to reveal a similar pattern in the spring of 2016."

**L247 – 252 replaced by**
"Both TP and turbidity showed contrasting patterns during the wet and dry seasons (Fig. 2D). Turbidity stayed below 60 FNU during the dry season until October and rapidly increased after a first rain event to 500 FNU (more

details refer to Figure S3 in supplementary information). A drop to about 200 FNU occurred right after this first peak, which seemed to correspond to excessive precipitation and a large pumping volume (Fig.2B). Soon after, turbidity went up again and peaked at 1800 FNU. Turbidity leveled off towards values around 200 FNU for the rest of the wet season and dropped below 60 FNU from April 2017 onwards. "

**L253-254 replaced by**
"TP concentrations were significantly higher during the period between 15-11-2016 and 01-03-2017 than the rest of the time (p-value < 0.001, Figure S5), during which TP fluctuated around 0.5 mg $L^{-1}$, but always below 1 mg $L^{-1}$."

**L254 added** "(Table S2, $R^2$ = -0.68)" **after** "low temperatures"

**L258 added** "high frequency" **after** "values from the"; **added** ", Table S1, $R^2$ = 0.88" **after** "Fig.2D"

**L259 delete** 'when'

**L261 Added** "and Table S2, $R^2$ = 0.72" **after** "Fig.2D"; **added** "(a negative correlation between temperature and Fe is shown in Table S1)" **after** "2 mg $L^{-1}$"

**L 267 replaced** "After that, the conservative mixing approach underestimated EC but the main patterns were still reproduced. Accordingly," **by:**
"After that, the conservative mixing approach underestimated the EC but the main dynamics and the amplitudes were still reproduced (Table S6, $R^2$ = 0.82); as groundwater is the only contributor to the high EC due to the seepage of quite mineralized, slightly brackish water, the model must underestimate the seepage flux from November 20$^{th}$, 2016 on. Overall, "

**L265 – 274 changed into:**
"A simple fixed-end-member mixing model was used to reconstruct the conservative mixing of EC, NH$_4$, and TP. The simulated and the measured EC, NH$_4$, and TP are plotted in Figure 3. The correlations between the modeled and measured results are shown in the supplementary information (Table S4-S6). Potential processes that might deprive or enrich nutrients relative to the conservative mixing process along the flow routes were inferred from the discrepancies between the modeled and the measured data. Figure 3(A) and Table S5 show that the predicted and observed EC dynamics agree reasonably well from May to November 20$^{th}$, 2016 ($R^2$ = 0.91). After that, the conservative mixing approach underestimated the EC but the main dynamics and the amplitudes were still reproduced (Table S6, $R^2$ = 0.82); as groundwater is the only contributor to the high EC due to the seepage of quite mineralized, slightly brackish water, the model must underestimate the seepage flux from November 20$^{th}$, 2016 on. Overall, the observed dynamics of EC are consistent with mixing of high EC seepage water with low EC runoff water (coefficient of determination between the modeled and measured EC is 0.65 over the complete period, Table S4).
The dynamics of measured NH$_4$ concentrations show close resemblance to the model results, especially during the wet season (01-10-2016~15-03-2017). Clearly, NH$_4$ is diluted during the rain events and a gradual increase of NH$_4$ starts after each rain event during the wet season showing slopes that resemble the model reconstruction. Over the whole period, measured NH$_4$ concentrations were overestimated by the model, indicating that some NH$_4$ is probably lost due to non-conservative processes. This is especially true for the spring season of 2017, where NH$_4$ concentrations must be controlled by additional processes. Concentrations of TP are generally far below the conservative model reconstruction, except between the end of November and the beginning of March. During this particular period the minimum measured TP concentrations are captured nicely by the conservative model, however distinct peaks up to 3 mg $L^{-1}$ are not captured by the model and must have different physical or chemical processes determining them. "

**L276-277, replaced** "Fig.1 (red blocks)" **by** "Fig.2 (4 pink shades)".

**L277 replaced** "significant" **by** "clear"

**L283 added** "dry season" **after** "rainfall during"

**L 286-287, rephrased into** "The dilution patterns of the NH4 in events 1 and 2 were similar to those of EC. Due to the data gaps of NH4 in event 3 and 4 we cannot describe the pattern of NH4 in these two events."

**L286-288 replaced** "The dilution pattern… as stated before"
**into**
"The dilution patterns of the $NH_4$ in events 1 and 2 were similar to those of EC ($R^2$ = 0.86 and 0.83, respectively, Table S7 & S8) and show resemblance for event 3 ($R^2$ = 0.75, Table S10). Moreover, a direct negative correlation between $NH_4$ and rain intensity supports this dilution effect for event 2. Due to the data gaps of $NH_4$ in event 4 we cannot completely describe the pattern of $NH_4$ for this one, but it corresponds with that start of reduced $NH_4$ which was described in sections 3.1 and 3.2"

**L280-293, changed into** "The response of TP was generally not related to the intensity of rainfall and pumping, except for event 3 during the wet period. Dilution effects, as were observed for $NH_4$, were not observed for TP for events 1, 2 and 4. During the wet season event 3, TP concentrations show negative correlations with precipitation and pumping intensity ($R^2$ = -0.79 and -0.59, respectively, Table S9) and correspond with decreasing turbidity. Event 4 marks the transition between the wet and dry season and the drop in TP coincides with the drop in $NH_4$, independently from individual rain storms during the dry season."

**L296-299 replaced by** "Turbidity is more variable and has higher variance for wet season events 3 and 4, which corresponds with the findings of the annual scale analysis (section 3.1.2). During event 3, turbidity varied between 100 and 500 FNU. Although clear relations exist between Fe, TP and turbidity, all higher during the wet season (Figure 2, Table S2), these are not clearly reflected at the scale of individual precipitation events. Simultaneous peaks of TP and turbidity occur that are not easily related to the weather conditions in November and December but TP and turbidity show contrasting signals at the start of the event. The turbidity clearly decreases during rain storm event 3 and at the start of event 4. This change is not reflected by the correlation at the total event scale (Tables S9 and S10) but obvious when studying only the time scale of the decreasing limb of the EC dilution. Event 4 coincides with the transition to the spring season in 2017, showing decreasing EC, TP and turbidity in the last rains of the wet season and a strong decrease of $NH_4$ and increase of turbidity when conditions dried up and temperatures rose."

**L301-304, moved** "In artificial lowland catchments, water systems are intensively regulated by pumping activity to prevent flood and drought. However, there is a substantial lack of knowledge about the possible consequences of such regulation on aquatic ecology and water quality. Peaks in P and turbidity by the activation of pumps was observed by Van der Grift in his high frequency monitoring campaign in an agriculture lowland polder (Van der Grift et al., 2014 & 2016)." **to L466.**

**L304-306 deleted** "In this study… respond to the pumping activity"

**L308-317 changed into**
"While the effects of pumping on EC are rather small, TP, NH4 and turbidity are all affected by pumping. The effects of pumping appear to be different for events in different seasons; turbidity for example increases during pumping in July and December but decreases in May. The increase during the December pumping is especially marked (R2 Pumping intensity versus Turbidity = 0.77, Table S13). TP decreases during pumping in July (R2 = -0.67) and October and increases in May (R2 = 0.6). Event 2 seems to have started a major drop in turbidity (more than 1000 FNU) that continued some time after pumping. "

**L320 added** "eventually" **after** "in order to"

**L327-353 basing on the comments from the Reviewers and the Editor, there is a major change to section 4.1. Now it is rephrased and adapted into** "In a highly manipulated low-lying urban catchment like Geuzenveld, mixing between rainwater and groundwater in the ditches is fast due to the high fraction of impervious area and the installation of both a rainwater and a groundwater drainage system that transport these contrasting water types efficiently to the ditches (Yu et al., 2019; Walsh et al., 2005). Runoff in Geuzenveld has EC of about 166 µS/cm (Yu et al., 2019), which is lower than the groundwater EC (1746 µS/cm on average). As a relatively conservative water quality parameter (Figure S2), mixing between rainwater and groundwater should be the main process for EC. This presumption is supported by the agreement between modelled and measured EC dynamics for the period between May to November 2016. Precipitation events diluted the EC values at the pumping station, and the magnitude of dilution depended on the intensity of precipitation; heavy rainfall resulted in low EC values (Fig.2D and Fig.4). In periods with absence of rainfall, the EC values follow a recovery curve that resembles a linearly mixed reservoir with concentrations increasing to values that approach the EC of the continuous groundwater supply of around 1500 µS/cm. After November 2016, the conservative mixing approach underestimated the EC but the main dynamics were still reproduced and the amplitude of the EC dynamics remains similar to the model

results, except for the short period Nov 20th- to Dec 1st, 2016. Starting around Nov 20th, the EC started to increase relative to the dry season before. It coincides with an intensive pumping event after the first intensive rainfall event that happened after a prolonged period of cumulative water deficit. This may be related with a first flush from the drain system that starts to be activated more strongly, thus removing clogged material and lowering the overall resistance of the drain system for shallow and deep groundwater inflow (van der Velde et al., 2010). It suggests that this triggered the inflow of somewhat more mineralized groundwater relative to the period before, creating a shift in the EC towards ~250 µS/cm higher values that continued during the remainder of the monitoring campaign. It appeared that it raised the EC, but did not change the amplitude or dynamics of the EC during the remainder of that period Fig 2 and 3, Table S6). An alternative reason for the higher EC starting from November, 2016 on, would be the application of road salts during the winter period. Although freezing conditions occurred from November onwards, we did not find any evidence for the prolonged effects of road salts, as the chloride concentrations in the grab samples only showed two higher measurements, one in December 2016 and one in January 2017 (see Supplement, Figure S2.) So, overall, the observed dynamics of EC are consistent with mixing of high EC seepage water with low EC runoff water.

During winter, mixing can also explain the dynamics of $NH_4$ and TP (Fig.3). Compared with groundwater, which carries around 8 mg $L^{-1}$ $NH_4$ and 1.6 mg $L^{-1}$ TP, rain and runoff have much lower nutrient concentrations, which makes groundwater the main nutrients source (Yu et al., 2019). Nutrients derived from groundwater mix with rainwater in the ditches through direct seepage and the efficient groundwater drainage systems. Clearly, $NH_4$ is diluted during the rain events and a gradual increase of $NH_4$ starts after each rain event during the wet season showing slopes that resemble the model reconstruction. Over the whole period, measured $NH_4$ concentrations are overestimated by the model, indicating that some $NH_4$ is probably lost to transformation processes. This is especially true in the spring season of 2017, where $NH_4$ concentrations must be controlled by other processes. Concentrations of TP are generally far below the conservative model reconstruction, except between the end of November and the beginning of March. During this particular period the minimum measured TP concentrations are captured nicely by the conservative model, however distinct peaks up to 3 mg $L^{-1}$ are not captured by the model and must have different physical or chemical processes determining them. While the mixing process can explain part of the dynamics of $NH_4$ and TP in the wet season, the mixing assumption cannot explain the behavior of $NH_4$ and TP during other seasons, when $NH_4$ and TP measured time series drift far below from the conservative mixing model pattern because of biological and chemical processes.**"**

**L356-359 replaced** "While $NH_4$ … Jäger et al., 2017)." **by** "$NH_4$ dynamics during winter can be explained by mixing. However, biological processes are overruling the mixing process during spring and summer. It resulted in lower measured $NH_4$ concentrations than modeled during this period. Studies have shown that benthic and planktonic primary producers (e.g. phytoplankton) assimilate nutrients and are an important factor controlling nutrient dynamics in rivers, lakes, and streams (Hansson, 1988; Jäger et al., 2017)."

**L360 added** "**, Table S3" after "Fig.2 and 3**"; added "**as" before "summerized**"; added "**and Table S15-S19" after "Figure 6"**

**L362-364 changed into:**
"Growth of primary producers results in a consumption of ammonium, phosphate and a production of organic-N, chlorophyll, oxygen, and suspended solids, and led to a relatively higher pH because of the uptake of CO2 (Figure 6). This patterns is also clearly reflected in the shift in the NH4/TN and organic-N/TN ratios during spring (Figure 6)"

**L377 deleted** "significantly"

**L380 replaced** "chlorophyII" **by** "chlorophyll"

**L391 added** "(Figure S6)" **after** "onwards"

**L397-401 deleted**

**L405 added** "Table S1-S3," **after** "water column ("

**L407 added** "(< 50 FNU)" **after** "low turbidity"

**L410 added** "the" **before** "sediments"

**L417-418 replaced** "From the late autumn…(Fig.6)" **by**
"From the late autumn onwards, turbidity and total Fe concentrations substantially increased compared to the rest of the time (Fig.2, p value < 0.001 for turbidity and = 0.02 for Fe). Turbidity peaked first at 1800 FNU and stayed at a plateau of ~200 NFU during the rest of the cold and wet season. Total Fe in the water column reached to 6 mg $L^{-1}$ from below 1 mg $L^{-1}$. During this period the water turned brownish and transparency declined (Fig.6)."

**L421 replaced** "S3" **by** "S4"

**L426 added** "the" **before** "water column"

**L428: at the end added**
"We suggest that the turbidity peak of 1800 FNU is caused by the mineralisation of the benthic algae once they die off when light and temperature conditions decrease, combined with the shift of ironhydroxide formation from the sediment-water interface to the water column. The latter process continues through the whole winter season, until primary production restarts in spring (Figure 7)."

**L432 added** "in iron flocs" **after** "was incorporated"

**L433 added** "Table S1, $R^2$ for Fe~turbidity 0.81, TP~Fe 0.65; Table S2, Fe~turbidity: $R^2$ = 0.72, , TP grab~Fe 0.79;" **before** "Yu et al"

**L438 added** "which summarizes our hypotheses about the functioning of the system." **at the end**

**L444 replaced** "uptaken" **by** "removed"; **added** "by" **before** "benthic"

**L447 replaced** "because of warming." **by**
"due to the continuous supply of anoxic groundwater, the mere absence of $O_2$-rich runoff, the oxidation process of Fe(II) and possibly by microbial organic matter decomposition during warm periods with relatively stagnant water."

**L449 added** "(Van der Grift et al. 2014, 2016)" **after** "water column"

**L450 replaced** "bounded by mineral compounds" **by** "sequestered to minerals"

**L452 added** "increasing P concentrations therein" **after** "water column"

**L455-456 replaced** "At … and winter" **by** "At the event scales, $NH_4$ and EC were reduced by dilution from precipitation/runoff. For P and turbidity there was no clear relation to precipitation events, except for events in late autumn and winter (e.g. Figure 4, event 3)."

**L456 deleted** "very"

**L459 deleted** "response to events in"; **added** "during the dilution periods that was associated with the winter events 3 and" **after** "our urban catchment"

**L462-464 changed** "In addition…(Fig.4)" **into** "In addition, Yu et al. (2019) showed that precipitation runoff delivers particles and $O_2$ to the ditches. We suggest that this accelerates the further aggregation of the iron complexes; the resulting larger particles more readily settle to the bottom, causing a reduction of turbidity during the events itself (Fig. 4, EC dilution part of events 3 and 4)."

**L466 replaced** "Van der Grift et al. (2014) studied agricultural areas and observed that P and turbidity were significantly increased by pumping events. However, in our study, " **by**
"This type of event scale dynamics would be easily missed in a daily or lower frequency sampling schedule, especially because pumping occurs almost solely overnight in our regulated catchments. As such, only a sampling schedule with 7 hours intervals (e.g. Neal et al. 2011) or high-frequency monitoring is able to catch the short-term dynamics (Van Geer et al. 2016).
Contrary to the findings of Van der Grift et al. (2014, 2016),"

**L472 changed** "a significant" **into** "an"

**L473-476 changed** "During the late autumn…the colloids." **into**
"During late autumn, we observed that the water was highly turbid (see also Yu et al. 2019) which we suggest to be caused by the formation of iron hydroxide colloids in the water column, which is supported by correlations between Fe-grab and Turbidity ($R^2$= 0.72, Table S2). We explain the reduced turbidity after a precipitation event as a result of the activation of the pumps which caused the export of the turbid water towards the receiving boezem in combination with aggregation of iron hydroxides in the water column and subsequent settling of the aggregates due to the supply of new $O_2$-rich water (Fig.5 event 2, see also Van der Grift, et al., 2014)."

**L484 replaced** "be" **by** "have been"

**L485 added** "and organic-N/TN" **after** "Fig.6 $NH_4$/N"

**L491 deleted** "at three different time scales: annual scale, rain event and pumping event scale"

**L498 replaced** "chlorophyII" **by** "chlorophyll"

**L499 added** "relatively" **after** "but"

**L500 replaced** "dominant" **with** "principle"

**L503 deleted** "is"

**L510 added:**

[revised manuscript text omitted]

---

## Editor Decision (ED1)

Editor decision

HESSD-Manuscript **"Drivers of nitrogen and phosphorus dynamics in a groundwater-fed urban catchment revealed by high frequency monitoring"** (HESS 2020-34).

Dear Dr. Liang Yu

Thanks for submitting the responses to the three reviews and the extensive material provided. You have properly addressed most of the comments. Nevertheless, there are still a few issues where further revision are needed. I listed them below.

**Reviewer 1, comment 41: Definition of wet and dry periods:** You have only partially addressed this comment. It is now clear how you have defined the wet and dry periods in an operational manner. However, as pointed out by Reviewer 1, this operational definition puts the end of the wet season at about mid March 2017 and not simply at the of February. Actually, inspecting the EC data in Fig. 2 nicely illustrates that the operational definitions captures the change points in the temporal evolution. However, the current version mixes an operational, data-driven definition of the period with a calendar-based one. Please rectify this issue such that the periods are delimited in a consistent manner.

**Reviewer 2, comment 7:** There are several linguistic shortcomings in the suggested text. Please carefully check the language (spell check, grammar).

**Reviewer 3, general comment:** Reviewer 3 was critical about the lack of statistical analyses. The same opinion was expressed by Reviewer 1 although he did not insist that much on that aspect. In your response, one reads '*We are not sure that statistical testing is the best approach in dealing with a complex dataset with high-frequency data.*'. Can you elaborate what you consider a more promising approach, how it is implemented in the manuscript and where you explain this to the reader?

**Reviewer 3, major comment 3:** Is the information provided in the response included in the main text and the figures shown in the SI? If not, please do so.

**Reviewer 3, major comment 7:** Please provide more information about your statistical analysis. Was there a significant difference between the two periods? Do the data fall into two clusters? Otherwise the split would be arbitrary. Show data (SI).

**Reviewer 3, major comment 13:** Unless there is a scientific reason not to perform a statistical test, please provide this information. Even if you think that the differences were evident enough, this is no argument not to

support this view by a statistical test.

Figure S4 is poorly explained. What is exactly depicted? What is an '... *increase in variations over the day.*'? One cannot see any sub-daily fluctuations. What are '*difference between daily values*'? One can only guess. Please be more precise and specific.

**Reviewer 3, major comment 30:** Unless there is a scientific reason not to perform a statistical test, please provide this information. Even if you think that the differences were evident enough, this is no argument not to support this view by a statistical test.

Please modify the manuscript as suggested and address the issues listed above.

Sincerely

Christian Stamm

---

## Author Response (AR2)

**Response to Reviewer 1**

**General comments**

**I would like to thank the authors for all the hard work they put into the revision of this manuscript, which has been improved much. My questions and comments have largely been answered satisfactorily. A list of some minor suggestions can be found in the specific comments. Once they are answered, I recommend publication of the article.**

We appreciate the reviewer for his confirmation and his contribution for improving the manuscript in the previous and the current revision.

**Specific comments**

**1. L99: A bracket is missing before '2019'**

**Agreed. Added** *"("* **before** *"2019".*

**2. L100: change 'catchments' to 'catchment'**

**Agreed. Changed** *"catchments"* **into** *"catchment".*

**3. L102: change '…pattern on the nutrient N…' to '…patterns of the nutrients N…'**

**Agreed. Changed** *"pattern on the nutrient"* **into** *"patterns on the nutrients".*

**4. L103: change 'since 1990s…' to 'Since the 1990s…'**

**Agreed. Added** *"the"* **before** *"1990s".*

**5. L107: Please expand 'NAP' once. The explanation given in the first version is not necessary.**

**Agreed. Added** *", NAP: Normalized Amsterdam Peil"* **after** *"NAP".*

**6. L108: Please remove the full stop between the brackets.**

**Agreed. Removed** *"."* **between** *"NAP)"* **and** *"(Fig.2".*

**7. L203: In the answers to the comments a value of 1.5 mm was given for the seepage. In the manuscript, a value of 2.0 mm was given. Which one is true?**

We apologize for this discrepancy. Indeed '1.5 mm per day' was typed in our previous response to the reviewer, see Reply to Reviewer 1 comment 34. However, '2 mm per day' as presented in the manuscript was the value taken in our calculation based on the behavior of the mixing model and calibration using the measured water levels (Figure 3).

**8. L211: Please correct the formatting of the text after the variable.**

**Changed accordingly.**

**9. L246: change 'significant' to 'significantly'**

**Agreed. Changed** *"significant"* **into** *"significantly".*

**10. L271: Change 'The coefficients of determination ($R^2$ "Pearson method used)…' to 'Pearson's coefficient of determination ($R^2$)…'**

**Agreed. Replaced** *"coefficients of determination ($R^2$ "Pearson method used)"* **by** *"Pearson's coefficients of determination ($R^2$)".*

**11. L325: It is no wonder the model cannot capture the short-term peaks of TP, since the model runs with a daily time step while the measurements were hourly. It would be better to aggregate the hourly measurements to daily averages and compare them to the model results for all parameters. Which of the sub-daily values were used for model comparison, anyway? Noon time? Midnight?**

The time step of our model is as well an hour, which is the same as the measurements (The measurements are aggregated from 5 min (EC), 10 min ($NH_4$), and 20 min (TP) intervals to an hour). It means we compared two datasets both with hourly interval.

The time for the sub-daily values used for model comparison were set at 10:00 am, as that usually is the time the samples were measured in the field or collected and preserved for later analysis in the lab.

**12. L378-384, Figure 4 and 5: Please add the unit to the turbidity axis, the SW level, water temperature and EC.**

Agreed. Figure 4 and 5 were changed accordingly. Please see the manuscript for the new version of these two figures.

**13. L412: A bracket is missing before 'Fig 2' and a full stop after 'Fig'.**
**Changed accordingly.**

**14. L393-431: There are still a lot of repetitions in chapter 4.1, e.g. the sentence in lines 426-428 was already used in lines 324-326. Please remove the repetitions and consolidate the text.**
**Agreed.**
**Line 324-326 changed into** *"During this particular period the minimum measured TP concentrations are captured nicely by the conservative model, but the distinct peaks up to 3 mg $L^{-1}$ are not. "*
*Section 4.1 now reads as below:*
*"4.1 Hydrological mixing between groundwater and rainfall*
*In a highly manipulated low-lying urban catchment like Geuzenveld, mixing between rainwater and groundwater in the ditches is fast due to the high fraction of impervious area and the installation of both a rainwater and a groundwater drainage system that transport these contrasting water types efficiently to the ditches (Yu et al., 2019; Walsh et al., 2005). Runoff in Geuzenveld has EC of about 166 μS/cm (Yu et al., 2019), which is lower than the groundwater EC (1746 μS/cm on average). As a relatively conservative water quality parameter (Figure S2), mixing between rainwater and groundwater should be the main process for EC. This presumption is supported by the agreement between the modelled and the measured EC dynamics for the period between May to November 2016. Precipitation events diluted the EC values at the pumping station, and the magnitude of dilution depended on the intensity of precipitation; heavy rainfall resulted in low EC values (Fig.2D and Fig.4). In periods with the absence of rainfall, the EC values follow a recovery curve that resembles a linearly mixed reservoir with concentrations increasing to values that approach the EC of the continuous groundwater supply of around 1500 μS/cm. After November 2016, EC was underestimated by the model. The sudden increase of the measured EC around Nov 20th coincides with an intensive pumping event after the first intensive rainfall that happened after a prolonged period of cumulative water deficit. This may be related with a first flush from the drain system that starts to be activated more strongly, thus removing clogged material and lowering the overall resistance of the drain system for shallow and deep groundwater inflow (van der Velde et al., 2010). It suggests that this triggered the*

*inflow of somewhat more mineralized groundwater relative to the period before, creating a shift in the EC towards ~250 µS/cm higher values that continued during the remainder of the monitoring campaign. It appeared that it raised the EC, but did not change its amplitude or dynamics during the remainder of that period (Fig. 2 and 3, Table S6). The elevated EC may alternatively due to the application of road salts in winter which starts from November. But we did not find any evidence for the prolonged effects of road salts, as the chloride concentrations in the grab samples only showed two higher measurements, one in December 2016 and one in January 2017 (see Supplement, Figure S2).*

*The mixing process can explain part of the dynamics of $NH_4$ and TP in the wet season, but insufficient for explaining the dynamics during the dry season due to the presence of biological and chemical processes. Compared with groundwater, which carries around 8 mg $L^{-1}$ $NH_4$ and 1.6 mg $L^{-1}$ TP, rain and runoff have much lower nutrient concentrations, which makes groundwater the main nutrients source (Yu et al., 2019). Nutrients derived from groundwater mix with rainwater in the ditches through direct seepage and the efficient groundwater drainage systems. Clearly, $NH_4$ is diluted during the rain events and a gradual increase of $NH_4$ starts after each rain event during the wet season showing slopes that resemble the model reconstruction. The overestimation of the modeled $NH_4$ in general indicates a probable lost to transformation processes, especially in the spring of 2017. Concentrations of TP are also generally far below the conservative model reconstruction. The distinct peaks up to 3 mg $L^{-1}$ are not captured by the model and must be determined by different physical or chemical processes.* **"**

**15. L553-555: There is something grammatically wrong in this sentence.**
**Agreed. Deleted "***and***", added "***the***" before "***observations***"**

**16. L604: '…New Orleans, Shanghai and Dhaka.' Are there any references for this?**
**We added the following references after** *"…New Orleans, Shanghai and Dhaka"* :
*" (Li et al., 2009; Nahar et al., 2014; Jones et al., 2016; Stahl, 2019)"*

**The full information of the references was added in the reference list as below:**

*Jones C.E., An K., Blom R.G., Kent J.D., Ivins E.R., and Bekaert D.. Anthropogenic and geologic influences on subsidence in the vicinity of New Orleans, Louisiana. JGR Solid Earth, 121(5): 3867-3887, 2016.*

*Li X., Chen M., and Anderson B.C.. Design and performance of a water quality treatment wetland in a public park in Shanghai, China. Ecological Engineering, 35: 18-24, 2009.*

*Nahar, M.S., Zhang, J., Ueda, A. et al. Investigation of severe water problem in urban areas of a developing country: the case of Dhaka, Bangladesh. Environmental Geochemistry and Health, 36: 1079-1094, 2014.*

*Stahl M.O.. Groundwater pumping is a significant unrecognized contributor to global anthropogenic element cycles. Groundwater. 57(3) : 455-464, 2019.*

**15. Figure S3: The caption states 'Hourly time series of TP and turbidity…' but only turbidity is shown in the graph.**
**Agreed. Deleted "***TP and***".**

**Responses to Reviewer 3**

**Revision of this paper was deeply performed by the authors. Conclusions are now well-supported by statistical tests, and the modifications performed on the text clarified all issues raised in the prevous evaluation report. I only have minor comments which are only made to help improving the presentation and reading of this work.**

We appreciate the confirmation from the reviewer, as well as his/her previous contribution into improving the manuscript.

**1. L25: replace "algae" by "algal"**
**Agreed. Replaced** *"algae"* **by** *"algal"***.**

**2. L33: add a comma before "we"**
**Agreed. Added ","** **before "***we***"**

**3. L51: not clear ; to be re-worded: "NH4 is the preferred N-form by microbes in some cases..." ?**
**Agreed.**
**Added** *"In estuaries,"* **before** *"NH$_4$ is the preferred..".*
**Changed** *"by"* **into** *"for".*
**Deleted** *"in some cases like in estuaries".*

**4. L55: is a climate-active gas; not clear, to be re-worded**
**Agree.**
**Replaced** *", or denitrified"* **by** *"It may also be denitrified".*
**Added** *"under such condition"* **after** *"N$_2$O".*

**5. L99: to be revised**
**Agreed. Changed** *"The groundwater seepage is a continuous source of anoxic, iron and nutrient rich slightly brackish waters."* **I**nto *"The groundwater seepage is a continuous source of slightly brackish, anoxic, and iron and nutrient rich water.".*

**6. L245: add, .."after seasons, ...and statistical tests."**
**Agreed. Added** *"and their significance test results"* **after** *"..seasons".*

**7. L246: should read "significantly"**
**Agreed. Changed "***significant***" into "***significantly***".**

**8. Table 2: please indicate in the footnote the type of statistical tests performed**
**Agreed. Added footnote after "***$p < 0.05$***":** *"Wilcoxon rank-sum test. The tests were performed in Rstudio (version 3.6.1), wilcox.text() in package "stats"."*

**9. For legends of Fig. 2 & 4 – a link with the "connected" supplementary tables or figs would facilitate a quick reading / e. g. "see Table S1, 2, 3 for the correlation tests performed on the datasets, and ...." Etc**
**Agreed.**
**Added "***See Table S1-S3 for the correlation tests performed on the dataset***" at the end of the caption of Figure 2.**

Added "*See Table S4-S6 for the correlation tests performed on the dataset*" **at the end of the caption of Figure 3.**

Added "*See Table S7-S10 for the correlation tests performed on the dataset*" **at the end of the caption of Figure 4.**

Added "*See Table S11-S14 for the correlation tests performed on the dataset*" **at the end of the caption of Figure 5.**

Added "*See Table S15-S19 for the correlation tests performed on the dataset*" **at the end of the caption of Figure 6.**

**10. L390: add: "… we start with the presentation of the ….**
**Agreed. Added "***the presentation of***" after "***…we start***".**

**11. L413: to be revised; "… November, 2016 on, » ??**
**Agreed.**
**Line 412, added** "(" **before** "Fig.2"**.**
**Line 413, replaced** *"An alternative reason for the higher EC starting from November, 2016 on, would be the application of road salts during the winter period."* **by** *"The elevated EC may alternatively due to the application of road salts in winter which starts from November."*

---

## Author Response (AR3)

**Comments from the Editor**

**There is just one single question that needs clarification. In response to Reviewer 1, you write on bullet point 11: "It means we compared two datasets both with hourly interval. The time for the sub-daily values used for model comparison were set at 10:00 am, as that usually is the time the samples were measured in the field or collected and preserved for later analysis in the lab. "**

**I was confused now: if you use hourly data, why do you use sub-daily data at one specific time? Can you clarify this (add also a sentence in the manuscript if appropriate).**

**Response to the editor**

We understand the Editor's confusion as the first paragraph of our previous response refers to the high-frequency measurements, and the second paragraph refers to the grab samples.

We now change the response into:

The time step of our model is as well an hour, which is the same as presented time series of the _high-frequency_ measurements. The presented _high frequency measurements_ were aggregated from 5 min (EC), 10 min ($NH_4$), and 20 min (TP) intervals into an hourly interval. It means we compared the two datasets with both hourly interval.

We now clarified this in the text at Line 220:

_"The model provided a tool to simulate hourly concentration dynamics under the assumption that EC, $NH_4$ and TP were conservative."_

What we meant to say about the grab samples was:

_The comparison between the hourly model, high-frequency and the grab sampling results was based on their concentrations at 10:00 am,_ as that usually was the time the _grab_ samples were measured in the field or collected and preserved for later analysis in the lab."

We changed the text to avoid this confusion, including a previous response that we gave to reviewer 3 in the first round of our responses to the reviewers (July 30th 2020):

[revised manuscript text omitted]